# Cardiotoxicity Induced by Anticancer Therapies: A Call for Integrated Cardio-Oncology Practice

**DOI:** 10.3390/ph18091399

**Published:** 2025-09-17

**Authors:** Giuliana Ciappina, Luigi Colarusso, Enrica Maiorana, Alessandro Ottaiano, Tindara Franchina, Antonio Picone, Gaetano Facchini, Chiara Barraco, Antonio Ieni, Maurizio Cusmà Piccione, Concetta Zito, Massimiliano Berretta

**Affiliations:** 1Section of Experimental Medicine, Department of Medical Sciences, University of Ferrara, 44121 Ferrara, Italy; giuliana.ciappina@unife.it; 2Department of Clinical and Experimental Medicine, University of Messina, 98122 Messina, Italy; dott.luigicolarusso@gmail.com (L.C.); maurizio.cusma@gmail.com (M.C.P.); concetta.zito@unime.it (C.Z.); 3School of Specialization in Medical Oncology, Department of Human Pathology “G. Barresi”, University of Messina, 98122 Messina, Italy; enrica.maiorana@studenti.unime.it; 4Division of Innovative Therapies for Abdominal Metastases, Istituto Nazionale Tumori IRCCS Fondazione G. Pascale, 80131 Naples, Italy; a.ottaiano@istitutotumori.na.it; 5Department of Human Pathology in Adult and Developmental Age ‘Gaetano Barresi’, University of Messina, 98122 Messina, Italy; tfranchina@unime.it (T.F.); aieni@unime.it (A.I.); 6Division of Medical Oncology, AOU “G. Martino” Hospital, University of Messina, 98125 Messina, Italy; antonio.picone@polime.it; 7Oncology Complex Unit, Santa Maria delle Grazie Hospital, Azienda Sanitaria Locale Napoli 2 Nord, 80078 Pozzuoli, Italy; gaetano.facchini@aslnapoli2nord.it (G.F.); chiara.barraco@aslnapoli2nord.it (C.B.)

**Keywords:** cardiotoxicity, anticancer drugs, targeted therapy, chemotherapy-induced cardiotoxicity, immunotherapy, cardio-oncology

## Abstract

The introduction of novel oncologic therapies, including targeted agents, immunotherapies, and antibody–drug conjugates, has transformed the therapeutic landscape of cancer care. This evolution has resulted in a dual clinical scenario; while survival outcomes have markedly improved, leading to a growing population of long-term cancer survivors, an increasing incidence of previously unrecognized treatment-related toxicities has emerged. Among these, cardiovascular adverse events represent some of the most prevalent and clinically significant complications observed in both conventional chemotherapy and modern therapeutic regimens. Cardiotoxicity has become a major concern, with the potential to adversely affect not only cardiovascular health but also the continuity and efficacy of oncologic treatments, thereby impacting overall survival. This opinion paper synthesizes current evidence, identifies critical gaps in knowledge, and advocates for a multidisciplinary, evidence-based framework to guide the prevention, early detection, and optimal management of cardiotoxicity associated with anticancer therapies.

## 1. Introduction

Oncological therapeutic tools are constantly evolving, as demonstrated by advancements in targeted therapy, immunotherapy, and antibody–drug conjugates (ADCs), which have significantly improved prognostic outcomes for cancer patients [1]. Despite the substantial improvement in survival rates brought by modern cancer therapies, these treatments are frequently accompanied by a wide range of adverse effects, whose nature, timing, and severity are influenced both by the therapeutic strategy employed and patient-specific factors [2]. Cardiovascular diseases (CVDs) represent some of the most important treatment-related toxicities, raising increasing concern due to their potential to cause premature morbidity and mortality in cancer survivors [3]. Although cancer recurrence or progression remains the primary cause of death in this population, CVDs consistently rank as the second most common cause [4]. While genetic and environmental factors may also influence cardiovascular risk, multiple epidemiological studies comparing cancer survivors with age- and sex-matched general populations indicate that prior cancer therapies, including chemotherapy, radiotherapy, and targeted agents, along with disease- and treatment-associated factors, play a substantial role in this increased incidence [5]. Moreover, epidemiological evidence indicates that cancer survivors exhibit an elevated risk of CVD, with cardiovascular mortality exceeding cancer-specific mortality in select tumor subtypes. An analysis of 3,234,256 cancer survivors from the SEER database, encompassing 28 different tumor types, revealed that 38% of deaths were attributed to cancer, while 11.3% resulted from CVDs, with 76.3% of these cardiovascular deaths caused by heart disease. The highest incidence of cardiovascular mortality was observed among patients with cancers of the urinary bladder (19.4%), larynx (17.3%), prostate (16.6%), uterus (15.6%), colorectum (13.7%), and breast (11.7%) [6]. Therefore, timely identification and effective management of cardiotoxicity are crucial for improving both survival outcomes and the quality of life in cancer patients [7]. The cardiotoxicity spectrum includes a wide range of CVDs; cancer treatment can be associated with life-threatening arrhythmia, ischemia, infarction, and damage to the cardiac valves, conduction system, or pericardium [8]. These frequent cardiovascular side effects may necessitate altering, pausing, or stopping life-saving cancer treatments, which risks reducing their effectiveness. Moreover, they can profoundly affect patients’ quality of life and overall survival, independent of the cancer prognosis [7]. Managing patients with coexisting cancer and cardiovascular disease effectively demands specialized knowledge and a coordinated multidisciplinary approach, factors that have prompted the emergence of the field of cardio-oncology [9]. Notably, the 2022 European Society of Cardiology (ESC) Guidelines on Cardio-Oncology formally introduced the concept of cancer-therapy-related cardiotoxicity, detailing specific entities including cancer-therapy-related cardiac dysfunction (CTRCD), immune checkpoint inhibitor (ICI)-associated myocarditis, hypertension, arrhythmias, and vascular toxicities [10]. However, the limitations in accurately predicting the long-term cardiovascular effects of cancer therapies contribute to both underdiagnosis and overdiagnosis of cardiovascular disease. This diagnostic uncertainty can lead either to missed opportunities for preventive intervention or the unnecessary discontinuation of potentially life-saving cancer treatments [11,12]. Despite growing interest in the field of cardio-oncology in recent years, numerous aspects of cancer-drug-induced cardiovascular disease remain incompletely understood and warrant further investigation [13]. This opinion paper aims to present a comprehensive overview of cardiotoxicity related to anticancer therapies, underscoring its increasing clinical relevance amid rising cancer survival rates. It advocates for a coordinated, multidisciplinary model of care and seeks to elucidate the current challenges in the early detection, risk assessment, prevention, and management of cardiotoxicity.

## 2. Therapeutic Agents Associated with Cardiotoxicity

Cancer-therapy-related cardiotoxicity includes various conditions that may be induced by different classes of agents, such as chemotherapeutic drugs like anthracyclines, proteasome inhibitors, and microtubule inhibitors; targeted therapies (e.g., anti-HER2 drugs, anti-angiogenic agents); immunotherapy; and newer drugs, such as ADCs. The mechanisms underlying cardiotoxicity and the associated clinical manifestations are typically characteristic of each drug class, although in some cases they may appear similar. Identifying the specific type of cardiotoxicity related to the pharmacological class is essential for the accurate therapeutic management of patients [14].

### 2.1. Chemotherapy-Induced Cardiotoxicity

#### 2.1.1. Anthracyclines

Anthracyclines, such as doxorubicin, daunorubicin, and epirubicin, are potent chemotherapeutic agents derived from Streptomyces species. Their anticancer activity is mediated by multiple mechanisms, including intercalation between DNA base pairs, which disrupts DNA replication and transcription; inhibition of topoisomerase II, preventing the religation of DNA double-strand breaks; and generation of reactive oxygen species, leading to oxidative damage of DNA, proteins, and lipids [15]. These drugs have been foundational in treating a diverse array of malignancies, ranging from hematologic cancers like leukemia and lymphomas to various solid tumors, including those of the breast, stomach, uterus, ovary, bladder, and lung. Despite their efficacy, the clinical application of anthracyclines is significantly constrained by their well-documented cardiotoxic potential, which increases with cumulative dosage. Indeed, the risk of anthracycline-induced heart failure is estimated at 3–5% with a cumulative dose of 400 mg/m^2^ and rises dramatically to 18–48% at 700 mg/m^2^, eventually leading to progressive cardiomyopathy and overt heart failure [16].

Given the indication of anthracyclines across a wide range of tumor types, the administration of these agents ranks among the most prevalent in oncology practice. Annually, approximately one million patients in North America receive anthracycline-based regimens. Even with a relatively low incidence of severe cardiac complications (up to 5% in high-risk patients), the absolute number of affected individuals remains substantial, given the large population of patients treated with anthracyclines worldwide [17]. In patients without predisposing factors, doxorubicin is typically well tolerated up to cumulative doses of 300 mg/m^2^, with heart failure occurring in less than 2% of cases. However, retrospective analyses reveal a dose-related increase in cardiotoxic events, reaching 3–5% at 400 mg/m^2^, 7–26% at 550 mg/m^2^, and up to 48% at 700 mg/m^2^, which has led to the discontinuation of higher-dose regimens. Nonetheless, the challenge of managing both subacute and late-onset cardiotoxicity persists [7]. Second-generation anthracycline derivatives, such as epirubicin and idarubicin, provide improved therapeutic indices compared to doxorubicin but have not eliminated the risk of cardiac injury. Similarly, mitoxantrone, an anthracenedione structurally related to anthracyclines, remains capable of inducing myocardial damage and subsequent cardiac dysfunction [18]. Importantly, cardiomyopathy can develop at lower cumulative doses in the presence of risk factors. In particular, hypertension (OR: 1.99, 95% CI: 1.43–2.76), diabetes mellitus (OR: 1.74, 95% CI: 1.11–2.74), and obesity (OR: 1.72, 95% CI: 1.13–2.61) have been significantly associated with an increased risk of anthracycline-induced cardiotoxicity. Other risk factors include arrhythmias, coronary artery disease, concomitant chemotherapy, or genetic predisposition [19]. Recent genetic studies have identified specific single-nucleotide polymorphisms (SNPs) associated with increased susceptibility to anthracycline-induced cardiotoxicity. Over 30 gene variants have been linked to heightened vulnerability to daunorubicin-related myocardial injury across diverse cellular models. Based on these findings, pharmacogenomic testing for RARG rs2229774, SLC28A3 rs7853758, and UGT1A6*4 rs17863783 variants is currently recommended for pediatric patients receiving doxorubicin or daunorubicin, whereas no recommendation exists for adults [20]. Notable variants include those in CBR1 (Carbonyl reductase 1) and CBR3 (Carbonyl reductase 3), ABCC1 (ATP-binding cassette subfamily C member 1) and ABCC5 (ATP-binding cassette subfamily C member 5), related to ATP-binding cassette transporters, and SLC22A (solute carrier gene), which are predominantly implicated in pediatric populations and patients treated for acute lymphoblastic leukemia [6]. Conversely, polymorphisms in the NOS3 gene, responsible for endothelial nitric oxide synthase production, may exert a protective effect in the context of high-dose doxorubicin therapy [21]. Anthracycline-induced acute cardiotoxicity, occurring during or shortly after infusion, is characterized by transient arrhythmias and a temporary decrease in left ventricular ejection fraction (LVEF). Although infrequent (<1%), these effects are generally reversible. Cardiotoxicity has been classified as early (within one year of treatment) or late (commonly emerging approximately seven years post-therapy), with evidence supporting a progressive and potentially irreversible myocardial injury if not promptly identified and managed [20]. Advances in cardiac imaging, particularly cardiac magnetic resonance imaging (MRI), have facilitated earlier detection of subclinical myocardial damage associated with anthracycline exposure [22]. Structural changes may involve dilation of all four cardiac chambers, though less pronounced than in other forms of dilated cardiomyopathy, with variable thinning of the left ventricular wall. Functional impairment commonly encompasses both systolic and diastolic dysfunction of the left ventricle [23]. Multiple molecular pathways have been implicated in anthracycline cardiomyopathy, yet there is consensus regarding characteristic histopathological alterations observable on endomyocardial biopsy of the right ventricle [7]. These include patchy interstitial fibrosis, proliferation of fibroblasts, infiltration by monocyte-derived histiocytes, myofibril loss, distension of the sarcoplasmic reticulum, and vacuolization of cardiomyocytes adjacent to fibrotic areas [23].

Cardiac damage primarily involves alterations in myocardial cell function and pathological cell death mediated by interconnected mechanisms, such as mitochondrial dysfunction, topoisomerase II inhibition, disruptions in iron metabolism, myofibril degradation, and oxidative stress. These molecular disturbances trigger cardiomyocyte loss through multiple regulated and unregulated cell death pathways, including apoptosis, autophagy, necrosis, necroptosis, pyroptosis, and ferroptosis [24].

Anthracyclines induce cellular damage by upregulating NOD-, LRR-, and pyrin domain-containing protein 3 (NLRP3), interleukin-1β (IL-1β), and interleukin-6 (IL-6). Activation of the NLRP3 inflammasome and subsequent release of pro-inflammatory cytokines recruit leukocytes to the myocardium, amplifying inflammation. Mitochondrial dysfunction represents a central mechanism of anthracycline cardiotoxicity. Anthracyclines interact with complex I (NADH dehydrogenase) and mitochondrial DNA (mtDNA), causing excessive production of reactive oxygen species (ROS), lipid peroxidation, and structural mitochondrial damage [25] (Figure 1). Given that mitochondria constitute approximately 36–40% of cardiomyocyte volume and generate about 90% of cellular ATP, their impairment critically compromises cardiac performance [26]. Mitochondrial injury is mediated through multiple interconnected processes:ROS generation and iron-catalyzed free radical formation, impairing enzymes critical for mitochondrial function, such as nitric oxide synthases, NAD(P)H oxidases, catalase, and glutathione peroxidase, leading to widespread DNA, protein, and lipid damage [27,28].Mitochondrial iron overload, resulting from chelation of intracellular iron and interference with iron regulatory proteins, including ferritin and the mitochondrial iron exporter ATP-binding cassette, subfamily B, member 8 (ABCB8), which promotes ferroptosis, a regulated cell death dependent on iron and lipid peroxidation. Doxorubicin also downregulates glutathione peroxidase 4 (GPx4), further facilitating ferroptotic injury [29].Cardiolipin disruption, as doxorubicin binds cardiolipin, destabilizing its association with cytochrome c and impairing mitochondrial membrane integrity and apoptotic signaling [7].Activation of Ca^2+^/calmodulin-dependent protein kinase II (CaMKII) via receptor-interacting protein kinase 3 (RIPK3) signaling, leading to opening of the mitochondrial permeability transition pore (mPTP) and necroptotic cell death, demonstrated in experimental mouse studies [24].

Beyond mitochondrial injury, anthracyclines induce genotoxic stress by targeting DNA topoisomerase II (Top2). The Top2α isoform, predominantly expressed in proliferating tumor cells, mediates anticancer effects, whereas the Top2β isoform, constitutively expressed in adult cardiomyocytes, underlies doxorubicin-induced cardiotoxicity. Formation of doxorubicin–Top2β–DNA complexes results in double-stranded DNA breaks, triggering apoptotic pathways in cardiomyocytes [30,31].

The tumor suppressor protein p53 plays a central role in anthracycline cardiotoxicity. Upon DNA damage, p53 activation initiates apoptosis and mitochondrial dysfunction through outer mitochondrial membrane permeabilization, cytochrome c release, and increased ROS production. Experimental models deficient in p53 demonstrate preserved mitochondrial integrity and reduced cardiac injury following doxorubicin exposure [32]. Furthermore, p53 disrupts mitochondrial quality control by sequestering Parkin (PARK2), a Parkinson’s disease-associated gene and potential tumor suppressor, in the cytosol, thereby inhibiting mitophagy and the clearance of damaged mitochondria [33].

In addition to cardiomyocytes, anthracyclines adversely affect non-myocyte cardiac cells, including fibroblasts, endothelial cells, vascular smooth muscle cells, and immune cells [34]. Experimental studies in cell cultures and mice have shown that doxorubicin compromises endothelial barrier function, thereby increasing vascular permeability and promoting myocardial interstitial edema [35]. It also activates TGF-β signaling via oxidative stress, stimulating fibroblast differentiation and excessive extracellular matrix deposition. These cellular and molecular alterations, combined with activation of innate immune pathways, drive maladaptive remodeling of the left ventricle [36]. In addition to anthracycline-induced cardiotoxicity, radiotherapy remains an indispensable modality in the treatment of thoracic malignancies, such as breast cancer and mediastinal lymphomas. Nevertheless, despite technological advances in radiation planning and delivery, inadvertent exposure of cardiac structures remains a critical concern, particularly when combined with anthracyclines, leading to synergistic increases in long-term cardiac risk [10]. Radiation can promote endothelial damage, inflammation, and fibro-calcific changes, thereby accelerating coronary artery disease, valvular dysfunction, pericardial fibrosis, and myocardial stiffening, all of which may exacerbate the progression of heart failure and arrhythmias in cancer survivors [37,38,39,40,41]. More recently, the gut microbiota has emerged as an additional factor linking cancer therapies and cardiovascular injury. Dysbiosis, induced by both malignancies and chemotherapeutic regimens, may modulate the host response to anthracyclines. Microbiota-derived metabolites, such as Trimethylamine N-oxide (TMAO), have been associated with enhanced atherosclerosis and heart failure risk, whereas short-chain fatty acids (SCFAs) like butyrate display anti-inflammatory and cardioprotective properties. Notably, butyrate has been shown to potentiate the antitumor efficacy of doxorubicin while attenuating its cardiotoxicity, highlighting the microbiome as a promising target for mitigating anthracycline-induced injury within cardio-oncology [42].

#### 2.1.2. Fluoropirimidin

Cardiovascular toxicity induced by chemotherapy is most commonly linked to anthracyclines, with fluoropyrimidines following as the next most frequent cause [43].

Through inhibition of nucleic acid synthesis, 5-fluorouracil (5-FU) and its oral prodrug capecitabine act as cytostatic antimetabolites. They are metabolically converted into active fluoronucleotides that interfere with DNA and RNA synthesis by inhibiting thymidylate synthase, incorporating into RNA, and disrupting DNA replication and repair. These agents are widely used in clinical protocols for a variety of malignancies, including gastrointestinal, genitourinary, head and neck, breast, esophageal, gastric, pancreatic, and colorectal cancers [44]. Although fluoropyrimidines are generally considered tolerable, cardiovascular adverse events are a relevant concern. Reported incidences vary across studies, from 1% to 18% according to Morelli et al. [7], and ranging up to 35% in other cohorts [43,45]. Clinically, fluoropyrimidine-associated cardiotoxicity can manifest as angina, arrhythmias, myocardial ischemia, infarction, or even sudden death. Several pathophysiological mechanisms have been proposed, including coronary vasospasm, endothelial dysfunction, oxidative stress, and mitochondrial injury [7].

The risk is notably heightened in patients with impaired metabolism due to dihydropyrimidine dehydrogenase (DPD) deficiency, the principal clearance pathway for these agents [44]. DPD deficiency may stem from genetic variants in the DPYD gene. Prevalence estimates range from 3 to 15% for partial deficiencies and up to 0.5% for complete loss of function. Four key loss-of-function DPYD alleles, DPYD2A (IVS14+1G>A, c.1905+1G>A), DPYD9B (c.2846A>T), DPYD*13 (c.1679T>G), and HapB3 (c.1129-5923C>G), have consistently demonstrated marked increases in fluoropyrimidine toxicities, with relative risks between 1.7 and over 4 for severe adverse events. In addition, carriers of DPYD variants were found to be significantly correlated with treatment-related mortality (OR = 34.86, 95% CI 13.96–87.05; *p* < 0.05) [46]. Although direct links between DPD deficiency and cardiotoxicity are still being clarified, evidence shows that a substantial proportion (39–61%) of severe toxic reactions to fluoropyrimidines—including cardiac events—occur in patients with reduced DPD activity. Furthermore, data from a prospective study of 487 individuals revealed that around 2% of those carrying DPYD variants developed serious cardiotoxic effects. These observations highlight the importance of DPD testing not only to prevent general fluoropyrimidine toxicity but also as a means to lower cardiovascular risk [47]. In most cases, cardiotoxicity presents as a reversible ischemic event involving coronary vasospasm, yet under certain conditions it may lead to more critical consequences, including cardiomyocyte loss due to thrombotic occlusion or myocardial infarction. Emerging evidence highlights that cardiomyocyte-specific toxicity constitutes an integral component of the overall pathogenic landscape [48]. The pathophysiology of fluoropyrimidine-induced cardiac injury involves several cell types, notably vascular smooth muscle cells, red blood cells, endothelial cells, and cardiomyocytes. Experimental data indicate that 5-FU can trigger vasoconstrictive responses in smooth muscle cells via mechanisms involving protein kinase C activation [43]. Endothelial damage induced by 5-FU appears to be mediated by oxidative stress, leading to senescence and cell loss, thereby facilitating a procoagulant phenotype and possible thrombotic complications. Simultaneously, direct cardiomyocyte injury has been observed, with 5-FU driving ROS accumulation and triggering both apoptosis and autophagic responses [45].

#### 2.1.3. Taxanes

Taxanes, including docetaxel, paclitaxel, and albumin-bound paclitaxel (nab-paclitaxel), are a class of antimitotic agents widely utilized in oncology. They exert their anticancer effects by binding to β-tubulin subunits, stabilizing microtubules, preventing their depolymerization, and thereby disrupting the dynamic microtubule rearrangements required for mitotic spindle formation and cell cycle progression. These drugs are commonly employed in the treatment of various solid tumors, notably breast, lung, and ovarian cancers [49]. However, their clinical use can be limited by notable adverse effects, among which cardiovascular toxicity is significant, occurring in approximately 3% to 20% of patients [7]. Cardiac complications primarily involve electrical conduction disturbances, such as QT interval prolongation, bradycardia, and atrial fibrillation. Most of these events tend to be mild and reversible upon treatment cessation, typically not necessitating targeted cardioprotective therapies [50]. The precise mechanisms driving taxane-related cardiotoxicity remain unclear, but several theories have been proposed. One prominent hypothesis centers on hypersensitivity reactions, characterized by substantial histamine release that may disrupt cardiac conduction and provoke arrhythmias. Consequently, prophylactic administration of corticosteroids and antihistamines is often recommended [7]. Of particular concern is the interaction between paclitaxel and anthracyclines like doxorubicin. Combined treatment has been linked to an increased incidence of heart failure and more severe myocardial damage, including extensive necrosis [51]. This effect is thought to arise from pharmacokinetic interference, whereby paclitaxel impairs the clearance of doxorubicin. In contrast, other taxanes, such as docetaxel, do not appear to potentiate doxorubicin-induced cardiotoxicity to the same extent [52].

#### 2.1.4. Alkylating Drugs

Alkylating agents, such as cisplatin, cyclophosphamide, ifosfamide, and mitomycin, are a class of chemotherapeutics that exert cytotoxic effects primarily by covalently binding to DNA, forming intra- and interstrand crosslinks that distort the DNA helix and block replication and transcription. These DNA lesions activate DNA damage response pathways, including checkpoint kinases and the tumor suppressor p53, ultimately triggering cell cycle arrest and apoptosis. Additionally, alkylating agents can generate reactive oxygen species, leading to oxidative stress, mitochondrial dysfunction, and further amplification of apoptotic signaling in malignant cells [7]. Among these, cisplatin is frequently administered in combination with other agents to enhance therapeutic response and limit resistance. Despite its efficacy, cisplatin has been associated with a considerable incidence of cardiovascular events, including angina and myocardial infarction, with reported prevalence ranging from 7% to 32% [7,53]. Long-term follow-up in survivors of cisplatin-based regimens has revealed sustained elevation of cardiovascular risk, including dysregulated lipid metabolism, hypertension, and insulin resistance persisting over a decade post-treatment [54]. Multiple cellular and molecular mechanisms may underlie these adverse effects. For instance, platelet activation and vascular injury contribute to thrombotic complications, while oxidative damage disrupts endothelial integrity and myocardial homeostasis. These processes are thought to result from cisplatin-mediated activation of pro-thrombotic and inflammatory pathways, including the arachidonic acid cascade and NF-κB signaling, leading to inflammatory cytokine release, cardiac tissue remodeling, and structural deterioration of the myocardium [7].

### 2.2. Cardiotoxicity Induced by Targeted Therapy

#### 2.2.1. Anti-HER2

ERBB kinase receptors (EGFR/ERBB1, ERBB2, ERBB3, and ERBB4), upon binding to their soluble ligands, form homo- or heterodimeric complexes, which activate their tyrosine kinase domains. This activation initiates multiple downstream signaling pathways that influence cell survival, proliferation, migration, and differentiation. Notably, ERBB2 (also known as HER2) receptor is a proto-oncogene frequently amplified and overexpressed in many human cancers [7]. The HER2 receptor is a critical target for therapies like monoclonal antibodies (e.g., trastuzumab, pertuzumab) and TKIs (e.g., lapatinib, tucatinib, afatinib, neratinib, dacomitinib). While these treatments have improved outcomes, they are associated with cardiotoxicity [55]. The cardiotoxicity of ERBB2-targeted therapies arises from inhibition of NRG1 signaling, a paracrine growth factor released by cardiac endothelial cells that plays essential roles in heart function. NRG1, together with its tyrosine kinase receptors ERBB4, ERBB3, and ERBB2, is critical for cardiac development and modulates regenerative, inflammatory, fibrotic, and metabolic processes in the heart. In cardiomyocytes, ERBB4 and ERBB2 are the most prominently expressed receptors, and NRG1 promotes fetal and neonatal cardiomyocyte proliferation, hypertrophy, sarcomerogenesis, and survival. By blocking ERBB2, these therapies interfere with NRG1-mediated protective and regenerative pathways, which can lead to impaired cardiac contractility, reduced stress resilience, and, in some cases, heart failure. Importantly, while ERBB2 expression declines after birth, residual activity in adult cardiomyocytes contributes to maintaining cardiac homeostasis, making the myocardium particularly vulnerable to pharmacologic inhibition of ERBB2 signaling [7]. Trastuzumab-induced cardiotoxicity is generally moderate and reversible, occurring in 2–5% of patients, with heart failure in up to 4%. However, combining trastuzumab with anthracyclines increases heart failure incidence to approximately 28%, attributed to compounded oxidative stress and mitochondrial dysfunction [56]. Recent studies have highlighted that late-onset cardiotoxicity can develop after prolonged HER2-targeted therapy, even in patients with previously stable cardiac function, underscoring the need for ongoing cardiac monitoring [57]. TKIs targeting HER2 also present cardiotoxic risks. Lapatinib and tucatinib have been associated with reductions in LVEF in approximately 2–5% and 1% of patients, respectively, though these effects are typically reversible [58]. Afatinib and neratinib exhibit a more favorable cardiac safety profile, with limited cardiotoxic events reported [59]. Nonetheless, the combination of trastuzumab and lapatinib has not demonstrated a significant increase in cardiotoxicity compared to trastuzumab alone, suggesting that dual HER2 blockade may be safe under careful monitoring [60] (Table 1).

#### 2.2.2. Tyrosine Kinase Inhibitors (TKIs)

Multi-targeted tyrosine kinase inhibitors (TKIs), including sunitinib, sorafenib, axitinib, vandetanib, cabozantinib, lenvatinib, pazopanib, ponatinib, and regorafenib, exert their anticancer effects in part by inhibiting key angiogenic receptors, such as vascular endothelial growth factor receptors (VEGFRs) and platelet-derived growth factor receptors (PDGFRs) [61]. While effective at suppressing tumor growth and angiogenesis, these agents are linked to significant cardiovascular toxicities, including hypertension, left ventricular dysfunction, and heart failure [62]. The underlying mechanisms remain incompletely understood, but VEGFR inhibition is thought to reduce endothelial production of nitric oxide (NO), a critical vasodilator, leading to hypertension [63]. This elevated blood pressure can cause capillary rarefaction, impairing myocardial perfusion and contributing to cardiac dysfunction [64]. Additionally, PDGFR blockade may result in pericyte loss, further compromising coronary microvascular function [63]. Clinical data indicate that up to 47% of patients treated with sunitinib develop hypertension, approximately 28% experience left ventricular dysfunction, and about 8% progress to heart failure [65]. Sorafenib exhibits a similar, though somewhat milder, cardiotoxicity profile [66].

Imatinib has significantly improved treatment outcomes in chronic myeloid leukemia (CML) and gastrointestinal stromal tumors (GIST). It targets the Bcr-Abl fusion protein in CML and c-Kit in GIST, as well as c-Abl and platelet-derived growth factor receptors (PDGFRs). Second-generation TKIs (dasatinib, nilotinib, and bosutinib) and third-generation TKIs (ponatinib) have been developed to overcome BCR-ABL mutations that reduce the efficacy of imatinib. However, these therapies have been associated with an increased risk of occlusive arterial disease [67].

With regard to imatinib, adverse cardiac effects, including severe heart failure, have been reported, particularly in elderly patients with pre-existing cardiac problems [68]. In vivo and in vitro studies suggest that imatinib-induced cardiotoxicity may be related to endoplasmic reticulum stress, mitochondrial damage, and impaired cardiac progenitor cell function [69]. Epidermal growth factor receptor (EGFR) TKIs, including erlotinib, gefitinib, afatinib, and osimertinib, target the EGFR, a transmembrane receptor tyrosine kinase involved in cell proliferation, survival, and differentiation. By binding to the ATP-binding site of EGFR, these agents inhibit autophosphorylation of the receptor and downstream signaling pathways, including RAS-RAF-MEK-ERK and PI3K-AKT, leading to reduced tumor cell proliferation and induction of apoptosis [70]. EGFR-TKIs are widely used in clinical practice, particularly for patients with non-small-cell lung cancer (NSCLC) harboring EGFR mutations. Despite their efficacy in treating these malignancies, EGFR-TKIs are also associated with cardiotoxicity, which may present as QTc interval prolongation, arrhythmias, reduced LVEF, and heart failure. The molecular mechanisms underlying this cardiotoxicity are not yet fully understood; however, several have been identified, including inhibition of the PI3K signaling pathway and effects on ion channels, HER2 inhibition, oxidative stress, autophagy, and apoptosis in cardiomyocytes and mitochondrial dysfunction [71]. Patients with anaplastic lymphoma kinase (ALK)-positive NSCLC can be treated with several lines of ALK inhibitors instead of chemotherapy. Although ALK-TKIs are generally well-tolerated by most patients, long-term toxicities are important to understand, particularly given the impressive increase in life expectancy [72]. Patients treated with ALK inhibitors may develop adverse CV events, including sinus bradycardia, atrioventricular (AV) block, QTc prolongation, hypertension, hyperglycemia, and dyslipidemia [10] (Table 1).

### 2.3. Cardiotoxicity Induced by Hormonal Agents

Tamoxifen is widely used as a selective estrogen receptor modulator in the treatment of hormone-receptor-positive breast cancer. Its cardiotoxic profile is mainly related to dyslipidemia, endothelial dysfunction, and an increased thromboembolic risk [73].

Androgen deprivation therapy (ADT) represents the cornerstone systemic treatment for locally advanced or metastatic prostate cancer (PCa). While it significantly improves survival, ADT is associated with an increased risk of cardiovascular events and cardiotoxicity [74]. The ESC categorizes cancer-therapy-related cardiovascular complications into nine major groups: (1) myocardial dysfunction and HF, (2) coronary artery disease (CAD), (3) valvular disease, (4) arrhythmias (particularly QT prolongation), (5) hypertension, (6) thromboembolic events, (7) peripheral vascular disease and stroke, (8) pulmonary hypertension, and (9) pericardial disorders [75]. Prolonged use of GnRH agonists may contribute to several of these cardiovascular complications [74].

Androgen Receptor Signaling Inhibitors (ARSi) have profoundly reshaped the therapeutic landscape of advanced prostate cancer, offering substantial gains in overall survival and delaying disease progression. However, a growing body of evidence highlights that these agents are not without significant cardiovascular implications. Recent meta-analyses and large pharmacovigilance studies have consistently reported an increased risk of hypertension, arrhythmias, myocardial ischemia, and even heart failure in patients receiving ARSi therapy [76]. Importantly, these concerns are not confined to metastatic disease, as emerging data in neoadjuvant settings also highlight a measurable burden of hypertension and thromboembolic complications, underscoring cardiovascular vulnerability across different stages of prostate cancer. Moreover, pharmacovigilance analyses of FAERS data have suggested differential risk profiles among ARSi, with abiraterone and apalutamide linked to a higher incidence of major adverse cardiovascular events (MACE) compared to enzalutamide [77]. Given that prostate cancer predominantly affects an older population frequently burdened by hypertension, diabetes, or established coronary artery disease, these findings underscore the need for meticulous cardiovascular risk assessment prior to initiating ARSi, regular blood pressure and rhythm monitoring during treatment, and prompt management of any emerging cardiac symptoms. Ideally, such patients should be managed within a collaborative cardio-oncology framework. As ARSi use continues to expand into earlier stages of prostate cancer, it becomes imperative to conduct further prospective studies and establish long-term registries to fully characterize its cardiotoxic profile, thereby informing multidisciplinary strategies that balance oncologic efficacy with cardiovascular safety [78].

Cyclin-dependent kinase 4 and 6 inhibitors (CDK4/6i), namely palbociclib, ribociclib, and abemaciclib, have become a cornerstone in the treatment of hormone-receptor-positive, HER2-negative advanced breast cancer, markedly improving progression-free survival and, in certain settings, overall survival. Despite their generally favorable toxicity profile, accumulating evidence highlights the need to monitor for cardiovascular effects, which, although less pronounced than with some other targeted therapies, are clinically relevant. The most consistently observed concern is QT interval prolongation, particularly with ribociclib, necessitating baseline and periodic electrocardiograms to mitigate the risk of serious arrhythmias. In contrast, palbociclib and abemaciclib have shown a comparatively lower propensity for QT prolongation, though sporadic cases of tachyarrhythmias and hypertension have still been reported. Recent real-world data and pharmacovigilance analyses further suggest that patients with existing cardiovascular comorbidities or those on concomitant QT-prolonging agents may be at increased risk [79]. Given the anticipated long-term use of CDK4/6 inhibitors in early and metastatic disease settings, a prudent approach includes comprehensive cardiovascular risk assessment before treatment initiation, correction of modifiable risk factors, such as electrolyte imbalances, and structured ECG monitoring throughout therapy.

### 2.4. Immuno-Related Cardiovascular Adverse Events (irAEs)

Immunotherapy exerts its effect by reactivating the host immune system. ICIs are among the most commonly used agents in oncology and have significantly improved survival outcomes, particularly in malignancies where previous therapeutic approaches had failed [80]. To date, eleven agents have been approved by the US Food and Drug Administration for use in a multitude of malignancies: two Cytotoxic T-lymphocyte–associated protein 4 (CTLA-4)-blocking antibodies (ipilimumab and tremelimumab); five programmed death-1 (PD-1)-blocking antibodies (nivolumab, pembrolizumab, cemiplimab, dostarlimab, and toripalimab); three programmed death-ligand 1 (PD-L1)-blocking antibodies (atezolizumab, avelumab, and durvalumab); and one lymphocyte activation gene 3 (LAG-3)-blocking antibody (relatlimab) (https://www.fda.gov/, accessed on July 2025). CTLA-4 is located intracellularly in resting T cells and translocates to the surface upon activation. By binding CD28 ligands with high affinity, CTLA-4 antagonizes the costimulatory receptor CD28 and inhibits TCR activity, thereby reducing T cell activation and susceptibility to antigen presentation [81]. The interaction between PD-1 and PD-L1 downregulates the adaptive immune response by suppressing effector T cell activity while enhancing the function of immunosuppressive regulatory T cells (Tregs), thereby maintaining immune homeostasis and preventing excessive or harmful immune reactions. However, cancer cells exploit the PD-1/PD-L1 pathway to evade immune surveillance, promoting tumor development and progression [82]. Nonetheless, the enhanced immune activation induced by ICIs may result in the emergence of immune-related adverse events (irAEs) affecting multiple organs, especially when these agents are administered as part of combination immunotherapy [83]. The immune-related toxicities are generally reversible upon temporary discontinuation of ICI treatment and are usually manageable with glucocorticoid therapy. However, cardiovascular immune-related events (CV irAEs), though uncommon, may carry a high risk of serious illness and death [84]. Late CV events, occurring beyond 90 days, are less thoroughly studied but are typically associated with an increased risk of non-inflammatory heart failure (HF), advancing atherosclerosis, hypertension, and elevated mortality [85]. CV irAEs include myocarditis, pericardial disease, Takotsubo-like cardiomyopathy, myocardial Infarction (MI), arrhythmias, and conduction disorders, with the highest risks observed for myocarditis (OR: 4.42, 95% CI: 1.56–12.50, *p* < 0.01; I^2^ = 0%, *p* = 0.93) and dyslipidemia (OR: 3.68, 95% CI: 1.89–7.19, *p* < 0.01; I^2^ = 0%, *p* = 0.66). The incidence of these CV irAEs ranged from 3.2 (95% CI: 2.0–5.1) to 19.3 (95% CI: 6.7–54.1) per 1000 patients in studies with a median follow-up of 3.2 to 32.8 months [86]. Although the mechanistic basis of ICI-induced cardiotoxicity is not yet fully delineated, accumulating evidence implicates the clonal expansion and activation of T lymphocytes bearing high-avidity, shared antigen-specific T cell receptors (TCRs), recognizing epitopes co-expressed by neoplastic and myocardial tissues [87]. Inhibition of immune checkpoints may consequently render cardiomyocytes more susceptible to immune-mediated injury. As a result, the T-cell-mediated immune response that contributes to tumor eradication may concurrently drive the pathogenesis of myocarditis [9]. The pathophysiological mechanism of ICI-associated Takotsubo cardiomyopathy may involve acute, ICI-induced multivessel coronary vasospasm. Alternatively, myocardial injury could occur indirectly, triggered by an intense adrenergic surge early in the course of ICI treatment characterized by a sudden, excessive release of catecholamines from the adrenal glands or cardiac sympathetic nerve terminals, leading to catecholamine-mediated myocardial stunning [88]. Acute coronary syndrome is a rare but possible CV complication of ICI therapy. The use of ICIs can alter peripheral immune tolerance across different tissues, exposing patients to neuroinflammatory disorders, visceral obesity, atherosclerosis, and leptin resistance. Preclinical models carrying loss-of-function mutations in PD-1, PD-L1, or CTLA-4 genes exhibit a higher propensity for atherosclerotic plaques characterized by elevated levels of VCAM-1, ICAM-1, galectin-3, and oxidized LDL, increased macrophage density, and pro-inflammatory cytokines. In summary, the absence of these immune checkpoint proteins promotes an atherogenic phenotype, mediated by enhanced infiltration of CD3+/CD4+ and CD3+/CD8+ lymphocytes within the atherosclerotic plaque. These findings have led to the hypothesis that ICIs may increase the risk of atherosclerotic cardiovascular disease (ASCVD) in cancer patients [89]. Additionally, ICI-induced inflammation may destabilize the fibrous cap of atherosclerotic plaques, precipitating acute MI [89]. Coronary vasospasm, particularly following treatment with PD-1 inhibitors, such as pembrolizumab, has also been proposed as a contributing mechanism to ICI-associated acute MI [90]. Finally, the primary cause of arrhythmias and conduction disorders is inflammation, which may occur locally, within the ventricular myocardium or the His–Purkinje system, or as part of a systemic inflammatory response [91]. Finally, states of dysbiosis have also been associated not only with reduced efficacy of immunotherapy but also with an increased incidence of immune-related adverse events, likely due to the well-established interplay between the microbiota and the immune system. Pharmacological agents capable of disrupting eubiosis, such as proton pump inhibitors, have been shown to contribute to these effects [92].

### 2.5. Cardiotoxicity Induced by Proteasome Inhibitors

Proteasome inhibitors (PIs), including Carfilzomib and Bortezomib, are pivotal in the treatment of multiple myeloma. However, their use is associated with varying degrees of cardiovascular toxicity, necessitating careful consideration in clinical practice [93].

Carfilzomib is an irreversible PI that inhibits the β2 and β5 subunits of the proteasome. Preclinical studies demonstrated Carfilzomib-induced cardiotoxicity in young adult mice, highlighting the potential for cardiac damage in vivo [94]. A meta-analysis indicated that Carfilzomib significantly increases the risk of cardiotoxicity, with odds ratios of 2.34 for all-grade and 2.69 for high-grade cardiotoxicity [95]. Furthermore, a study analyzing FDA Adverse Event Reporting System (FAERS) data reported an 18.2% rate of any cardiovascular adverse event associated with Carfilzomib, with event rates ranging from 0% to 52% [96]. These findings underscore the importance of monitoring cardiac function in patients receiving Carfilzomib.

Bortezomib, in contrast, is a reversible PI that primarily inhibits the β5 subunit. While it has been associated with a lower incidence of CV adverse events compared to Carfilzomib, preclinical data suggest that Bortezomib can also induce cardiotoxicity. A study by Wesley et al. assessed the effects of Carfilzomib and Bortezomib on cardiac function in animal models, observing decreased fractional shortening and left ventricular ejection fraction, indicating potential cardiotoxic effects of both agents [97]. A real-world study involving 395 patients found that 20.8% experienced any grade of CV adverse events, with hypertension being the most common [98]. Additionally, a systematic review and meta-analysis reported that high-grade cardiotoxicity was more frequent with Bortezomib compared to the control group, with an odds ratio of 1.67. However, the overall cardiovascular risk remains lower than that associated with Carfilzomib [99].

### 2.6. Cardiotoxicity by New Classes of Drugs

ADCs comprise a novel therapeutic approach that merges the targeting precision of monoclonal antibodies with the potent cytotoxicity of chemotherapeutic agents [100]. To date, fifteen ADCs have been approved by the US Food and Drug Administration (FDA) for use across a range of malignancies. These include two anti-HER2 antibodies, trastuzumab emtansine and trastuzumab deruxtecan; two anti-trophoblast antigen 2 (TROP2) antibodies, sacituzumab govitecan and datopotamab deruxtecan; one anti-Nectin-4 antibody, enfortumab vedotin; one anti-Tissue Factor antibody, tisotumab vedotin; one anti-folate receptor alpha (FRα) antibody, mirvetuximab soravtansine; and one anti-c-Met antibody, telisotuzumab vedotin. In addition, other FDA-approved ADCs comprise gemtuzumab ozogamicin targeting CD33, brentuximab vedotin targeting CD30, inotuzumab ozogamicin targeting CD22, polatuzumab vedotin targeting CD79b, belantamab mafodotin targeting BCMA, loncastuximab tesirine targeting CD19, and moxetumomab pasudotox targeting CD22 for hairy cell leukemia, although the latter has since been withdrawn from the US market (https://www.fda.gov/, accessed July 2025) [100]. The cardiotoxicity of ADCs is highly dependent on their molecular target. The most commonly reported cardiovascular adverse events include LVEF reduction, arrhythmias, and, less frequently, hypertension and ischemic events. Among the ADCs for which cardiotoxicity has been most thoroughly investigated are those targeting HER2. In particular, trastuzumab emtansine (T-DM1) has been shown to induce cardiotoxic effects in a notable percentage of patients (exact incidence varies across studies; CHF/LVEF drop grade 3/4 was reported in 0.71%, cardiac ischemia in 0.1%, cardiac arrhythmia in 0.71%, and grade 1/2 LVEF drop in 2.04%) [101]. The underlying mechanism of cardiotoxicity is believed to be multifactorial. It primarily involves the HER2 blockade in cardiac tissue, where HER2 plays a role in cardiomyocyte survival signaling, particularly under stress conditions. The conjugated cytotoxic agent, DM1 (a maytansinoid that disrupts microtubules), may further exacerbate cardiac injury through mitotic spindle inhibition and off-target toxicity, particularly in cells with high metabolic demand like cardiomyocytes. Moreover, the non-cleavable linker used in T-DM1 may contribute to intracellular accumulation of the toxic payload in cardiac cells following nonspecific uptake [102]. Trastuzumab deruxtecan (T-DXd) has demonstrated superior efficacy and a favorable safety profile compared to trastuzumab emtansine in the DESTINY-Breast03 trial. In that study, the incidence of LVEF decline was low, occurring in about 1.1% of patients treated with T-DXd, compared to 1.7% in the T-DM1 group. Symptomatic heart failure was rare, highlighting that while cardiotoxicity is possible, it remains an infrequent event with appropriate monitoring, particularly in patients with prior cardiac risk factors [103]. TROP2-targeted ADCs, such as sacituzumab govitecan and datopotamab deruxtecan, deliver cytotoxic payloads specifically to TROP2-expressing tumor cells. These ADCs typically use topoisomerase I inhibitors as their payload, which induce DNA damage and tumor cell death. Importantly, TROP2 expression is minimal in cardiac tissue, suggesting a low likelihood of direct cardiotoxicity. Clinical trials to date have reported a low incidence of cardiac adverse events, with no significant increase in left ventricular dysfunction or symptomatic heart failure observed [104,105]. However, long-term cardiovascular safety data remain limited, and continued monitoring is recommended, especially in patients with pre-existing cardiac conditions or those receiving combination therapies. Another class of innovative and rapidly evolving therapeutic agents in oncology is T-cell-based immunotherapies, including chimeric antigen receptor (CAR) T cells and bispecific T cell engagers (BiTEs). CAR T cells are patient-derived T lymphocytes that have been genetically modified to express chimeric antigen receptors targeting tumor-associated antigens, such as CD19 or B-cell maturation antigen (BCMA), thereby promoting tumor cell apoptosis. BiTE molecules are engineered fusion proteins composed of two distinct antigen-binding domains: one specific for the CD3 receptor on T cells, facilitating their activation, and the other targeting a tumor-specific antigen on malignant cells [106]. Current knowledge of cardiotoxicity related to T cell therapies is primarily derived from clinical experience with CAR T cell treatments. Available data indicate that cardiotoxic effects predominantly manifest in the setting of cytokine release syndrome (CRS) and correlate with the severity of CRS. Nevertheless, it remains uncertain whether cardiotoxicity represents a direct consequence of CRS or an independent phenomenon. Moreover, certain CAR T cell constructs, such as those targeting melanoma-associated antigen-3, have exhibited cross-reactivity with titin, a cardiac striated muscle protein, resulting in fatal fulminant myocarditis [107].

## 3. Diagnostic Strategies and Risk Stratification

### 3.1. Baseline Risk Stratification

The assessment of baseline CV risk is a critical component in the cardio-oncologic evaluation prior to cancer therapy. This process enables both primary prevention—in patients without known CVD—and secondary prevention in those with established conditions. Key goals include early identification of modifiable risk factors, optimization of current treatments, and avoidance of cardiovascular complications during oncologic care [108].

A comprehensive evaluation involves clinical history, physical examination, blood pressure measurement, lipid and glycemic profiles, renal function assessment, and consideration of lifestyle behaviors, such as smoking and physical activity. Risk stratification is often guided by validated models, such as SCORE2 and SCORE2-OP, which provide 10-year estimates for major atherosclerotic events [109,110]. These models are adapted for age and regional risk but are not specific to oncology settings. In patients with type 2 diabetes, the SCORE2-Diabetes tool offers a more tailored approach [111].

Individuals with prior myocardial infarction, stroke, symptomatic peripheral arterial disease, severe chronic kidney disease, or diabetes with organ damage are automatically considered at very high CV risk and require intensive multifactorial intervention [112]. Older adults represent a unique subset due to their increased baseline risk and often silent subclinical disease. Frailty, defined as a decline in physiological reserves and increased vulnerability to stressors, plays a central role in prognosis and treatment planning. It is recommended that elderly patients undergo structured frailty assessment to determine therapeutic tolerance and adapt both cancer and cardiovascular care accordingly [113].

Crucially, CV risk stratification should not serve as a barrier to timely oncologic treatment. Instead, it facilitates a patient-centered, multidisciplinary approach, ensuring that cardioprotective strategies are introduced proactively and that oncologic decisions are made in alignment with cardiovascular safety.

### 3.2. Cancer-Therapy-Related Risk Stratification

While traditional cardiovascular risk scores, such as SCORE2 or SCORE2-Diabetes, effectively capture long-term atherosclerotic risk in the general population, they fall short in evaluating the acute, therapy-related cardiovascular risk faced by cancer patients. Antineoplastic agents differ significantly in their cardiovascular toxicity potential, and their combination with patient- and tumor-specific factors requires a tailored risk assessment approach [114].

To address this need, the Heart Failure Association (HFA) of the ESC, in collaboration with the International Cardio-Oncology Society (ICOS), developed the HFA-ICOS risk stratification tool, which predicts the likelihood of therapy-related cardiac toxicity (Table 2). This tool is based on expert consensus and organizes patients into four levels of risk—low, moderate, high, and very high—by integrating both oncologic therapy characteristics (e.g., type of agent, cumulative dose, radiation field) and patient-specific factors, such as age, baseline cardiovascular disease, and prior exposure to cardiotoxic agents [10].

Frailty, as an emerging modifier of cardiovascular risk, becomes particularly relevant in elderly oncologic patients. It amplifies vulnerability to therapy-induced toxicity and must be integrated into the global cardiovascular risk assessment. The synergistic effect of aging, comorbidity burden, and cancer therapy underscores the need for individualized risk models [10,113].

### 3.3. Electrocardiogram (ECG)

The 12-lead resting ECG is a low-cost, accessible, and essential tool in cardio-oncology, used both at baseline and during monitoring, especially with therapies known to cause arrhythmias or QTc prolongation [10,114]. Baseline ECG is recommended in all patients, particularly before treatment with TKIs, arsenic trioxide, fluoropyrimidines, or ICIs, which are associated with conduction disturbances and arrhythmias [75]. Key parameters include QTc interval, atrial enlargement or atrial anomaly, axis deviations, and signs of prior infarction.

QTc prolongation increases the risk of torsade de pointes and sudden death, especially in the presence of electrolyte imbalance or QT-prolonging drugs. The Fridericia correction (QTcF) is preferred over Bazett’s. A QTcF > 500 ms or a rise > 60 ms from baseline warrants further evaluation. Serial ECG monitoring is then advised during treatment with electrophysiologically active agents. New arrhythmias, bundle branch blocks, or ST-T abnormalities should prompt further workup with imaging or biomarkers [10,75]. ECG may also aid in detecting silent ischemia during fluoropyrimidine therapy. Continuous or ambulatory monitoring may be useful in symptomatic patients or those with suspected paroxysmal arrhythmias.

In summary, ECG remains a first-line, non-invasive modality for early detection of cardiotoxicity and guiding management in oncology patients.

### 3.4. Echocardiography

Transthoracic echocardiography (TTE) is the cornerstone of cardiovascular imaging in cardio-oncology, crucial both at baseline and during treatment—especially with anthracyclines, HER2-targeted agents, ICIs, multiple myeloma therapies, or thoracic radiotherapy, all associated with left ventricular (LV) dysfunction [10,75]. Baseline TTE is recommended for all patients starting potentially cardiotoxic therapies.

Three-dimensional (3D) LVEF is preferred for its accuracy and reproducibility; the biplane Simpson’s method remains a valid alternative when 3D is unavailable [115]. Global longitudinal strain (GLS) adds sensitivity for early dysfunction; a >15% relative reduction from baseline suggests subclinical cardiotoxicity, even with preserved LVEF [116]. Although the SUCCOUR trial did not show superiority of GLS-guided therapy over LVEF monitoring at 3 years, it led to earlier cardioprotective interventions [117,118]. Newer techniques, such as 3D-GLS and myocardial work indices, show promise but require validation in larger cancer cohorts [119].

Monitoring frequency should be tailored to risk and treatment; high-dose anthracyclines may warrant imaging every two cycles and trastuzumab every 3 months. In low-risk patients, follow-up can be symptom-based [75]. Diastolic function should be assessed, particularly in patients with hypertension or diabetes. Additional findings, such as pericardial effusion, valvular disease, or pulmonary hypertension, must be documented and followed [10].

Combining echocardiographic data with clinical evaluation and biomarkers (e.g., troponins, natriuretic peptides) enhances early cardiotoxicity detection and enables timely intervention. TTE remains an indispensable, non-invasive tool for risk-adapted cardiovascular surveillance in oncology patients.

### 3.5. Serum Biomarkers

Cardiac biomarkers, particularly high-sensitivity cardiac troponins (hs-cTn) and natriuretic peptides (BNP or NT-proBNP), are crucial tools in the early detection and monitoring of cancer-therapy-related cardiovascular toxicity. Their integration into routine cardio-oncology practice enables risk stratification, detection of subclinical injury, and prognostic assessment [10,120].

High-sensitivity troponins are markers of myocardial injury and can identify early cardiotoxic effects, particularly in patients treated with anthracyclines, trastuzumab, and other HER2-targeted therapies. A rise in hs-cTn during or after treatment—even in the absence of symptoms or echocardiographic changes—has been associated with a higher incidence of subsequent left ventricular dysfunction [120,121].

Natriuretic peptides reflect myocardial wall stress and may be elevated in both acute and chronic phases of cardiac dysfunction. Their measurement is especially useful in patients receiving VEGF inhibitors, TKIs, and immunotherapies, where heart failure or hypertension-related complications may emerge [75]. However, interpretation must consider potential confounders, such as renal function, age, and baseline comorbidities. Baseline values of hs-cTn and NT-proBNP should be obtained prior to starting treatment, and serial assessments are advised in patients classified as high or very high CV risk or when using agents with known cardiotoxic profiles [10]. An increase in hs-cTn above the 99th percentile upper reference limit (URL) or an NT-proBNP level > 125 pg/mL may warrant further cardiac imaging or therapy modification [121,122]. Moreover, cTnT and cTnI increase during and shortly after cardiac irradiation. These increases were positively associated with the cardiac radiation doses for the whole heart and the left ventricle, including in chemotherapy-naive cancer patients. Late natriuretic peptide increases years following treatment can identify patients with left ventricle dysfunction.

Emerging biomarkers are increasingly recognized for their potential to predict and monitor cancer-therapy-related cardiotoxicity, particularly in its early or subclinical stages. Among these, myeloperoxidase, indicating oxidative stress, has shown predictive value for cardiotoxicity following anthracyclines or trastuzumab. MicroRNAs, as modulators of inflammation and apoptosis, and growth differentiation factor-15 (GDF-15), a marker of oxidative stress and tissue injury, are being explored for their diagnostic relevance. Additional promising biomarkers include galectin-3 (a marker of fibrosis and remodeling), arginine–nitric oxide metabolites, and heart-type fatty acid-binding protein (H-FABP), all of which reflect distinct pathophysiologic mechanisms [123].

A multimarker strategy integrating these biomarkers may offer enhanced sensitivity in detecting early myocardial injury. However, the clinical utility of biomarkers is currently limited by the lack of standardized assays, defined cut-off values, and evidence-based sampling schedules. Future studies are needed to determine how biomarker-guided interventions can be tailored to specific anticancer therapies.

Furthermore, baseline genetic testing may help identify cancer patients with a predisposition to cardiotoxicity, enabling personalized surveillance strategies based on specific molecular pathways, such as myofilament integrity, mitochondrial function, or apoptosis susceptibility. Importantly, the integration of biomarkers with advanced cardiac imaging and clinical data has been shown to significantly improve the accuracy of cardiotoxicity detection, and this approach is now incorporated into most cardio-oncology surveillance protocols [115]. Finally, there is growing interest in applying omics sciences, particularly radiomics, to the assessment of cancer-therapy-related cardiac dysfunction. Complementary tools, such as genomics, transcriptomics, proteomics, and metabolomics, offer insights into gene expression, RNA transcription, protein dynamics, and metabolic changes during oncologic treatment, opening new frontiers for personalized risk stratification and early detection [124].

### 3.6. Advanced Diagnostic Strategies

Several cancer treatments are associated with an increased risk of stable angina and chronic coronary syndromes (CCS). 5-FU and capecitabine can precipitate effort angina in some patients, while platinum-based chemotherapy-induced ischemia usually occurs during the first 1–3 cycles, particularly in those with underlying CAD. The incidence of cardiac ischemia varies according to the class of agents: 1–5% with antimicrotubule agents, 2–3% with small-molecule VEGF and TKIs, and 0.6–1.5% with VEGF monoclonal antibodies. Moreover, nilotinib, ponatinib, and ICIs accelerate atherosclerosis, which may lead to stable angina [10].

In the cardiovascular evaluation of cancer patients, particularly those undergoing therapies associated with ischemic or structural risk, advanced imaging techniques are essential for diagnosis, risk stratification, and surveillance.

Provocative testing is particularly indicated in symptomatic individuals (e.g., with chest pain or dyspnea), those with a history of cardiovascular disease, and in patients receiving cancer therapies known to promote coronary vasospasm or atherosclerosis, such as fluoropyrimidines, VEGF inhibitors, BCR-ABL inhibitors, and various tyrosine kinase inhibitors [10,75]. It also holds value in asymptomatic long-term survivors exposed to high-dose mediastinal RT (>15 Gy mean heart dose) for whom non-invasive screening for CAD should be considered 5 years post-treatment and repeated every 5–10 years [10].

The choice of stress modality, ranging from exercise ECG (low pre-test probability) and stress echocardiography to coronary CT angiography (CTCA) (anatomic study; intermediate/low pre-test probability), pharmacologic stress cardiac magnetic resonance (CMR), or myocardial perfusion (scintigraphy) imaging (functional image studies; intermediate high pre-test—probability), depends on pre-test probability of coronary artery disease, patient-specific factors, and institutional availability [125]. Stress echocardiography is often preferred in the oncologic population due to its favorable safety profile, broad availability, and diagnostic performance, particularly when physical stress is feasible [126]. In cases requiring pharmacologic stress, agents like dobutamine or adenosine may be used effectively.

Coronary imaging, especially coronary CT angiography (CTCA), has emerged as a valuable non-invasive technique in cancer patients with suspected obstructive coronary disease. CTCA is particularly useful in those with inconclusive stress test results, intermediate pre-test probability, or prior exposure to cardiotoxic treatments associated with vascular toxicity. In patients treated with high-dose mediastinal RT, periodic coronary assessment with either CTCA or functional stress imaging remains an essential component of long-term cardiovascular follow-up [75,127].

CMR complements the diagnostic armamentarium by providing high-resolution, radiation-free assessment of ventricular function, myocardial mass, perfusion, and tissue characteristics. It is especially useful for detecting early signs of cardiotoxicity, including inflammation, edema, and fibrosis, even in asymptomatic patients or those with preserved ejection fraction [128]. Stress CMR further allows for precise evaluation of inducible ischemia and is particularly advantageous in patients with limited echocardiographic windows or complex clinical scenarios. However, its broader use is often limited by availability, cost, and contraindications to gadolinium in patients with severe renal dysfunction [10].

Together, these imaging strategies—provocative testing, coronary imaging, and CMR—offer a synergistic approach for the cardiovascular evaluation of oncology patients. Their appropriate and risk-adapted use enhances early detection of therapy-related ischemic and structural heart disease, guiding timely cardioprotective interventions.

## 4. Management of Cardiotoxicity and Surveillance Programs

CTRCD, as defined in the most recent ESC guidelines, refers to a broad spectrum of cardiac injury caused by oncologic treatments. This includes both asymptomatic and symptomatic forms of cardiac dysfunction, ranging from subclinical left ventricular systolic impairment to overt heart failure. The condition may result from various cancer therapies, including chemotherapy, targeted therapy, immunotherapy, and RT [10].

CTRCD can be broadly categorized into two main clinical forms: asymptomatic and symptomatic. Each of these can be further graded based on severity. The asymptomatic form is subdivided into mild and severe, while the symptomatic form, clinically manifested as heart failure, may range from mild to very severe.

Symptomatic CTRCD presents with typical features of heart failure, such as dyspnea, fatigue, ankle swelling, and sometimes physical signs like jugular venous distension, peripheral edema, or pulmonary rales. Diagnosis in these cases is primarily clinical, although imaging and biomarkers play an important role in refining the assessment and determining severity and etiology.

In contrast, asymptomatic CTRCD is diagnosed entirely through instrumental and laboratory criteria, as patients do not show overt symptoms. Findings, such as a decline in LVEF, a relative drop in GLS of more than 15%, or elevated cardiac biomarkers (troponins or natriuretic peptides), in the absence of symptoms, are diagnostic of early, subclinical cardiac dysfunction. These abnormalities often signal the initial stages of CTRCD and highlight the importance of routine cardiac surveillance during treatment.

A mild asymptomatic CTRCD can even occur with preserved LVEF and may only be detected by (a) elevation of cardiac biomarkers (e.g., cTnI/T > 99th percentile, BNP ≥ 35 pg/mL, NT-proBNP ≥ 125 pg/mL), (b) significant changes from baseline values, and (c) a relative reduction in GLS > 15% (Table 3).

Cardiovascular imaging is central in the diagnosis of CTRCD. TTE is the first-line modality. Accurate and reproducible measurement of LVEF is essential, with 3D echocardiography being preferred where available due to its stronger agreement with CMR, the current gold standard. When 3D is not feasible, the biplane Simpson’s method is recommended. Assessment of myocardial deformation (GLS) is crucial for the early detection of subclinical systolic dysfunction. GLS should be evaluated both at baseline and during therapy to track changes in ventricular performance over time. Equally important is the evaluation of diastolic function, especially in detecting heart failure with preserved ejection fraction (HFpEF), which is commonly observed in patients treated with thoracic RT [47].

Finally, echocardiography allows for the non-invasive identification of other cardiac complications beyond LVEF impairment. These include abnormalities in right ventricular function, atrial size, valvular integrity, pulmonary artery pressure, and pericardial status. Such comprehensive imaging supports a broader evaluation of cancer-therapy-related cardiotoxicity, facilitating timely diagnosis and tailored intervention.

### 4.1. Pharmacological Cardioprotection

The pharmacological prevention and attenuation of CTRCD are cornerstones of cardio-oncology, particularly in patients at increased cardiovascular risk or undergoing potentially cardiotoxic treatments. Pharmacologic cardioprotection encompasses both primary prevention—aimed at averting myocardial injury before clinical dysfunction—and secondary prevention, which focuses on halting or reversing early signs of cardiac damage. The therapeutic strategy should be individualized based on a patient’s baseline cardiovascular profile, type and intensity of oncologic therapy, and early indicators of cardiotoxicity observed through imaging or biomarker monitoring.

The main pharmacological agents employed in cardioprotection include renin–angiotensin system (RAS) inhibitors, beta-adrenergic blockers, dexrazoxane, and, in specific settings, mineralocorticoid receptor antagonists (MRAs), sodium–glucose co-transporter 2 inhibitor (SGLT2i), angiotensin receptor–neprilysin inhibitors (ARNIs), and statins. Each class of agents has a different mechanism of action and evidence base, which guide their application across cardio-oncology protocols (Table 4).

RAS inhibitors, such as angiotensin-converting enzyme inhibitors (ACEIs) and angiotensin receptor blockers (ARBs), confer cardioprotective effects through afterload reduction, anti-remodeling properties, and attenuation of neurohormonal activation. In patients exposed to anthracyclines, the use of enalapril has been shown to prevent LVEF decline in individuals with elevated troponin levels after chemotherapy, as demonstrated in the pivotal trial by Cardinale et al., which established a rationale for biomarker-guided prophylactic therapy [129]. Similarly, the PRADA trial [130] demonstrated that candesartan attenuated the reduction in LVEF in early breast cancer patients receiving anthracycline-based regimens, supporting its use in primary prevention. However, long-term follow-up showed no significant benefit in LVEF preservation at two years. Other studies testing enalapril or ramipril have yielded mixed or negative results, particularly when used in low-risk populations or in short-term follow-up.

Beta-blockers (BBs), particularly carvedilol and bisoprolol, have also demonstrated efficacy in preventing CTRCD. Their mechanism includes sympathetic blockade, reduction of oxidative stress, and improved diastolic relaxation. Carvedilol, due to its additional antioxidant properties and ability to stabilize mitochondrial function, is of particular interest. The CECCY trial [131] evaluated carvedilol in breast cancer patients receiving anthracyclines and showed a reduction in troponin elevation and diastolic dysfunction, although the difference in LVEF decline was not statistically significant compared to placebo. Nonetheless, beta-blockers are often used in combination with RAS inhibitors for additive benefit, particularly in high-risk individuals or those with evidence of early subclinical myocardial injury. However, recent findings from the Cardiac CARE trial [132] challenge this approach, as candesartan plus carvedilol failed to prevent LVEF decline in patients with troponin-guided high-risk features. This has sparked debate, with some authors highlighting the potential limitations of relying solely on LVEF and advocating for more sensitive markers, such as strain imaging or multi-biomarker strategies. A recent meta-analysis further highlighted the lack of high-quality evidence supporting beta-blockers in this setting, reporting neutral effects on LVEF and no reduction in clinical events [133]. In contrast, a factorial trial using ramipril and bisoprolol showed significant reductions in subclinical cardiac damage, suggesting a possible benefit in early myocardial injury prevention [134].

Sodium-glucose cotransporter-2 inhibitors (SGLT2i), originally developed for glycemic control, have demonstrated cardioprotective effects in cancer patients at risk of therapy-induced cardiotoxicity. Their mechanisms include attenuation of mitochondrial stress, reduced intracellular Na^+^/Ca^2+^ overload, AMPK activation, and anti-inflammatory actions [135,136]. Preclinical studies have shown that SGLT2i preserve cardiac function by preventing LVEF and GLS deterioration, limiting fibrosis, and reducing biomarkers of injury. Clinically, a recent meta-analysis of over 100,000 cancer patients and survivors showed that SGLT2i use reduced HF hospitalizations by 51% (RR 0.49, 95% CI 0.36–0.66, I^2^ = 28%, *p* < 0.01) and new HF diagnoses by 71% (RR 0.29, 95% CI 0.10–0.87) and MyD88-dependent signaling, which are critical drivers of cardiac inflammation and fibrosis. Furthermore, SGLT2is has been shown to enhance mitochondrial viability in cardiac cells, promoting improved cellular energy metabolism and function, thus mitigating cardiotoxicity [137]. Recent preclinical and clinical data suggest that SGLT2is exerts cardioprotective effects through multiple mechanisms, including the modulation of inflammasome activity, specifically by reducing NLRP3 (NOD-, LRR-, and Pyrin Domain-Containing Protein 3) inflammasome activation and MyD88-dependent (Myeloid Differentiation Primary Response 88) signaling, which are critical drivers of cardiac inflammation and fibrosis. Furthermore, SGLT2is has been shown to enhance mitochondrial viability in cardiac cells, promoting improved cellular energy metabolism and function, thus mitigating cardiotoxicity [138].

Dexrazoxane is the only agent specifically approved by the FDA and EMA for cardioprotection in patients receiving anthracyclines. It appears to ameliorate the cardiotoxicity seen with anthracyclines by fusing with free and bound iron, thereby decreasing the formation of anthracycline–iron complexes and, eventually, the production of reactive oxygen species, which are harmful to the surrounding cardiac tissue [139]. Clinical trials have consistently demonstrated its efficacy in reducing the incidence of heart failure and preserving LVEF, especially in patients receiving high cumulative doses of anthracyclines [129]. In clinical practice, dexrazoxane infusion (dosage ratio of dexrazoxane/doxorubicin is 10/1, e.g., 500 mg/m2 dexrazoxane per 50 mg/m2 doxorubicin) should be considered (at least 30 min prior to each anthracycline cycle) in adult patients with cancer scheduled to receive a high total cumulative anthracycline dose for curative treatment and in patients with high and very high CTRCD risk (including those with pre-existing HF or low–normal or reduced LVEF) where anthracycline chemotherapy is deemed essential [10].

MRAs, such as spironolactone and eplerenone, have been evaluated in small trials and observational studies. Their benefits include prevention of myocardial fibrosis and remodeling through aldosterone blockade. These agents are generally reserved for patients with reduced ejection fraction or overt heart failure, particularly when there is an indication independent of cancer therapy.

The role of statins in cardioprotection is supported by experimental and observational data suggesting that their pleiotropic effects—including anti-inflammatory action, improvement of endothelial function, and modulation of nitric oxide pathways—may mitigate anthracycline-induced damage. More recently, several trials have studied the use of statins for cardioprotection in patients treated with anthracyclines who do not have conventional indications for statin therapy. A meta-analysis of three observational studies and four RCTs found a lower incidence of cardiotoxicity in patients treated with statins for primary prevention. However, there was no significant change in LVEF between the intervention and control groups. Given the differences in patient populations and statin regimens, the small sizes of the studies, and the lack of hard clinical endpoints or medium-term effects, further research is needed before statin administration can be confidently recommended for this purpose [140].

ARNIs, such as sacubitril/valsartan, are also emerging as potential cardioprotective agents. The PRADA II trial showed that while sacubitril/valsartan did not significantly prevent LVEF decline compared to placebo, it improved global longitudinal strain and attenuated increases in NT-proBNP and troponin, suggesting a protective effect on subclinical myocardial injury during adjuvant breast cancer therapy [141]. The MAINSTREAM trial is also evaluating the role of ARNIs to prevent cardiotoxicity in breast cancer therapy but, at present, ARNI use in CTRCD is still investigational, and no specific guideline recommendations are available [142].

Other agents, such as ivabradine, and non-pharmacologic interventions, like remote ischemic preconditioning, have been tested in early trials, but so far have not demonstrated meaningful benefits in preventing CTRCD [143,144].

Furthermore, combination strategies are being explored. The SAFE-HEaRt study demonstrated that patients with mildly reduced LVEF (40–49%) at baseline could safely receive HER2-targeted therapies without significant cardiac deterioration if managed with cardioprotective drugs and close monitoring [145]. Such findings reinforce the feasibility of continuing life-saving oncologic treatments under cardiologic supervision when pharmacological support is in place.

In patients undergoing cancer therapy who develop severe symptomatic cardiotoxicity (CTRCD), current guidelines recommend discontinuation of chemotherapy unless specific exceptions are discussed within a multidisciplinary team and a robust cardioprotective strategy is implemented, along with close cardiovascular monitoring during each treatment cycle [10].

According to the ESC cardio-oncology guidelines, patients who develop moderate symptomatic or moderate to severe asymptomatic CTRCD while receiving anthracycline therapy should temporarily interrupt treatment. In the case of mild cardiotoxicity, the decision to continue or suspend chemotherapy should be made based on a personalized risk–benefit assessment within the multidisciplinary team. Treatment of left ventricular systolic dysfunction in the oncologic setting should follow the 2021 ESC guidelines for acute and chronic heart failure [47]. Guideline-directed medical therapy (GDMT) with ACEi or ARBs, BBs, SGLT2i, and MRA is recommended, unless contraindicated or not tolerated. These drugs should be titrated to the maximally tolerated dose. In patients with asymptomatic mild CTRCD, chemotherapy may be continued without interruption, but initiation of ACEi, ARB, or BBs should be considered [10]. Additionally, aerobic exercise training is advised both before and during anthracycline-based therapy, based on its demonstrated cardioprotective effects. Furthermore, a recent phase II trial demonstrated that structured cardio-oncology rehabilitation was safe and helped attenuate LVEF decline while also improving body mass index in obese breast cancer patients undergoing cardiotoxic therapy [146].

For patients with mild to moderate symptomatic CTRCD or moderate to severe asymptomatic CTRCD who recover ventricular function, re-challenge with anthracyclines may be considered. This should occur without interruption of ongoing heart failure therapy and after a thorough multidisciplinary discussion. When chemotherapy continuation is deemed necessary, further strategies should be adopted to mitigate cardiotoxic risk, including minimizing the cumulative dose of anthracyclines, using liposomal formulations, and administering dexrazoxane prior to each cycle [147,148].

Chest pain syndromes suspected to be caused by 5-FU-induced or capecitabine-induced coronary vasospasm should be acutely treated with nitrates and/or calcium-channel blockers. After suspected coronary vasospasm, nitrates and/or calcium-channel blockers should be considered. If re-challenging with 5-FU or capecitabine is planned, it should be conducted in a closely monitored setting [140].

Patients treated with HER2-targeted agents may develop symptomatic or asymptomatic CTRCD. In patients with severe CTRCD (LVEF < 40%), HER2-targeted therapy should be interrupted and heart failure treatment initiated per 2021 ESC recommendations, especially if cancer therapy cannot be stopped. Temporary suspension is also warranted in cases of moderate/severe symptomatic or severe asymptomatic CTRCD [10]. For mild symptomatic CTRCD, shared decision making is essential to determine whether to pause or continue treatment. In cases of moderate asymptomatic CTRCD (LVEF 40–49%), continuation of HER2 blockade is permissible, provided that cardioprotective agents (ACEi/ARB and beta-blockers) are introduced and close cardiac monitoring is conducted. For mild asymptomatic dysfunction (LVEF ≥ 50% with GLS reduction and/or biomarker elevation), HER2 treatment may proceed with the optional addition of cardioprotective therapy. All patients continuing or resuming HER2 blockade after CTRCD resolution should undergo frequent cardiovascular assessment—initially, every two cycles, and then less frequently if stability is confirmed [149].

Myocarditis is a rare but severe complication of ICIs often associated with high mortality. It typically presents within the first 12 weeks of therapy, though delayed cases beyond week 20 have been described. Diagnosis is based on clinical symptoms, elevated cardiac troponins, and new ECG abnormalities, such as conduction blocks or arrhythmias. All abnormal findings should prompt urgent cardiac imaging to exclude differential diagnoses, including acute coronary syndrome and viral myocarditis. In hemodynamically unstable patients, empirical initiation of high-dose IV methylprednisolone 1 g/day, 3 days, is recommended even before diagnostic confirmation. CMR, applying modified Lake Louise criteria, and echocardiography are central to the diagnostic workup [150]. Fluorodeoxyglucose Positron Emission Tomography (FDG-PET) can be considered when CMR is contraindicated, though its sensitivity is limited. Endomyocardial biopsy should be reserved for inconclusive cases. Management depends on severity. In fulminant cases (shock, unstable arrhythmias), intensive care unit (ICU) admission and mechanical support may be necessary. Non-fulminant cases include symptomatic patients with stable hemodynamics or incidental findings. ICIs should be suspended in all suspected cases until diagnosis is clarified. Standard heart failure and arrhythmia management per ESC guidelines is indicated [151]. Beyond myocarditis, ICIs have been associated with non-inflammatory left ventricle systolic dysfunction (LVSD), stress cardiomyopathy, and ischemic heart failure phenotypes. Non-inflammatory HF usually presents late and requires the exclusion of myocarditis or acute coronary syndromes. Although vasculitis and coronary artery disease have also been reported in this setting, if myocarditis is excluded, immunosuppressive therapy is not indicated. The decision to continue or discontinue ICIs depends on the severity of heart failure and should be made after comprehensive cardiac evaluation [152].

Despite limited trial data, cardiovascular complications—including heart failure—occur in up to 20% of adults treated with CAR-T therapy [153]. In approximately 14% of cases, this manifests as overt heart failure, which should be managed in accordance with ESC guidelines for acute and chronic HF.

The potential impact of cardioprotective agents on the antitumor efficacy of systemic cancer therapies has been carefully considered. Current evidence does not support any detrimental interaction between commonly used cardioprotective drugs, including RAS inhibitors, beta-blockers, SGLT2 inhibitors, MRAs, dexrazoxane, and statins, and the oncologic effectiveness of chemotherapeutic regimens [154,155]. In fact, several of these agents, such as RAS inhibitors and statins, have demonstrated favorable effects on tumor biology in preclinical models and observational studies, potentially through anti-inflammatory, anti-angiogenic, or pro-apoptotic mechanisms [156,157,158]. Dexrazoxane, previously debated for its safety in pediatric oncology, has now been validated in multiple studies and is FDA/EMA-approved for high-risk patients receiving anthracyclines [159]. While SGLT2 inhibitors and ARNIs remain investigational in cardio-oncology, no evidence thus far suggests that they interfere with cancer therapy, and emerging data highlight a possible role in protecting against subclinical cardiac injury [141,160]. Continued investigation through prospective trials is warranted to further assess long-term oncologic safety and efficacy in this setting.

**Table 4 pharmaceuticals-18-01399-t004:** Pharmacologic cardioprotection in oncology: evidence and recommendations.

Drug Class	Mechanism of Action	Primary Indication	Evidence Level	Guideline Recommendations
RAS Inhibitors (ACEi/ARBs)	Afterload reduction, anti-remodeling, neurohormonal attenuation	Elevated troponin post-anthracycline; high-risk patients	Mixed: Cardinale 2006 [129]; PRADA [130]; SAFE [134]	ESC 2021: consider in patients with LVEF drop ≥ 10% and to a value lower than 50% (Class IIa, LOE B) [47]; AHA 2022: uncertain benefit (Class IIb) [161]
Beta-Blockers	Sympathetic blockade, antioxidant effects, improved diastolic function	High-risk or early subclinical dysfunction; combined with RAS inhibitors	Mixed: CECCY [131]; Cardiac CARE [132]; meta-analyses [133]	ESC 2021: preferably carvedilol if LVEF < 50% [47]; ESMO 2020: may be considered in high-risk patients (Class IIb) [114]; AHA 2022: uncertain benefit (Class IIb) [161]
SGLT2i	Mitochondrial protection, anti-inflammatory, AMPK activation	Pre-existing HFrEF, high cardiotoxic risk therapies, early dysfunction	Growing: preclinical, Bhalraam 2025 [137], Greco 2025 [138]	Recommended in HFrEF and high-risk therapy patients; emerging role in early dysfunction [138]
Dexrazoxane	Topoisomerase II inhibition, iron chelation	High cumulative anthracycline dose	Strong: Swain 2003 [147]; Lipshultz 1995 [148]	ESC 2022: high and very high CV toxicity risk when anthracycline chemotherapy is indicated (Class IIa, LOE B) [10]
MRAs (Spironolactone, Eplerenone)	Aldosterone blockade, anti-fibrotic effects	Reduced LVEF or overt heart failure with concurrent indication	Limited: small trials/observational studies	No formal guideline recommendation; consider in HFrEF per general HF guidelines [47]
Statins	Anti-inflammatory, endothelial function improvement	Under investigation for primary prevention in anthracycline therapy	Weak/modest: meta-analysis; Raisi-Estabragh 2024 [140]	No official recommendation; not routinely recommended pending further evidence
ARNIs (Sacubitril/ Valsartan)	Neprilysin inhibition, natriuretic peptide enhancement, and RAAS inhibition	Under investigation for prevention of cardiotoxicity during breast cancer	Emerging: PRADAII [141]; ongoing MAINSTREAM trial [142]	No formal guideline recommendation; consider per general HF guidelines [47]
Ivabradine/RIC	Heart rate reduction/ischemic preconditioning	Not recommended; insufficient evidence	Negative/experimental: Rizk 2025 [143]; Moreno-Arciniegas 2024 [144]	Not recommended; negative trials and lack of benefit shown

ACEi: angiotensin-converting enzyme inhibitor; ARB: angiotensin receptor blocker; ARNI: angiotensin receptor–neprilysin inhibitor; BB: beta-blocker; CMR: cardiac magnetic resonance; CTRCD: cancer-therapy-related cardiac dysfunction; EF: ejection fraction; ESC: European Society of Cardiology; ESMO: European Society for Medical Oncology; GLS: global longitudinal strain; HF: heart failure; HFrEF: heart failure with reduced ejection fraction; LOE: level of evidence; LVEF: left ventricular ejection fraction; MRA: mineralocorticoid receptor antagonist; NT-proBNP: N-terminal pro–B-type natriuretic peptide; RAAS: renin–angiotensin–aldosterone system; RCT: randomized controlled trial; RIC: remote ischemic conditioning; ROS: reactive oxygen species; SGLT2i: sodium–glucose cotransporter-2 inhibitor.

### 4.2. Cardiac Surveillance Protocols

The following section summarizes current recommendations for cardiac monitoring in patients receiving potentially cardiotoxic cancer therapies, based on the 2022 ESC Guidelines on Cardio-Oncology. Guidelines from the ESC and the International Cardio-Oncology Society (IC-OS) emphasize a risk-adapted and multidisciplinary approach to surveillance [10]. Patients are stratified as low, moderate, high, or very high risk using structured proformas that consider prior cardiovascular disease, age, cardiovascular risk factors, prior cancer therapies, and the type and intensity of current treatment. This stratification directly informs the frequency and intensity of cardiovascular monitoring.

For patients receiving anthracyclines, the frequency of cardiac assessments is dictated by the cumulative dose and individual risk profile. In low-risk patients receiving <250 mg/m^2^ of doxorubicin or equivalent, echocardiography is typically performed at baseline and after treatment completion. In contrast, high-risk patients or those exceeding 250–300 mg/m^2^ require more intensive monitoring, with echocardiographic evaluations repeated after every 100 mg/m^2^ of cumulative dose or every 2–3 cycles of therapy. In all cases, an increase in hs-Tn above the upper reference limit or a >15% relative decline in GLS from baseline should prompt consideration of initiating cardioprotective therapy, even in the absence of symptoms [10].

HER2-targeted therapies, such as trastuzumab, confer a significant risk of CTRCD, especially when administered sequentially or concurrently with anthracyclines. In these patients, echocardiography should be performed every three months during treatment, with additional imaging if symptoms arise or biomarkers change. In low-risk, asymptomatic patients with stable cardiac function, this interval may be extended to every four months following the third-month evaluation [10]. Biomarkers should also be obtained at baseline, especially in patients with prior exposure to anthracyclines, and used during follow-up to detect subclinical dysfunction [162].

In the context of ICIs, the surveillance strategy is less well-established but increasingly recognized as essential due to the risk of fulminant myocarditis. The incidence of ICI-related myocarditis is relatively low (<1%), but the associated mortality may exceed 40% [163]. Baseline ECG, natriuretic peptides, and cTn measurements are recommended in all patients prior to starting treatment. Baseline echocardiography is specifically recommended for high-risk patients. Serial ECG and cTn measurements should be considered before the second, third, and fourth ICI doses, and, if results remain normal, testing can be spaced to every three doses until completion of therapy to detect subclinical ICI-related cardiovascular toxicity. For patients requiring long-term ICI treatment (>12 months), cardiovascular assessment is recommended every 6–12 months in high-risk patients and may be considered in all other patients. Any rise in troponin, new ECG abnormalities, or unexplained symptoms should prompt immediate cardiac workup, including cardiac MRI or endomyocardial biopsy when myocarditis is suspected [10].

For VEGF inhibitors and TKIs, the primary concern is the development of hypertension and arterial thrombotic events, but their use is also associated with a wide array of CV complications, including HF and QTc prolongation. Baseline echocardiography is recommended for high- and very high-risk patients, with cardiology referral if LV function is impaired. Monitoring should include serial ECGs, biomarkers, and echocardiography, with early detection and management of hypertension essential. Home blood pressure monitoring is advised during the first cycle, after dose changes, and every 2–3 weeks thereafter. QTc should be monitored after dose increases, with new QT-prolonging drugs, or if electrolytes are abnormal. Periodic assessment of HF signs and natriuretic peptides can help detect therapy-related cardiac dysfunction. Risk factor assessment, ECG, echocardiogram, lipid profile, HbA1c, and ankle-brachial index are part of the baseline workup, with repeat vascular evaluation at 6–12 months for higher-risk agents, such as nilotinib and ponatinib (multitargeted kinase inhibitors targeting BCR-ABL and generally associated with vascular events) [10]. Left atrial anomaly on baseline ECG before ibrutinib has been shown to be a predictor for the development of atrial fibrillation (AF) during chemotherapy [164,165]. Figure 2 provides an example of VEGFi-related cardiotoxicity. The presence of atrioventricular (AV) conduction delays and premature atrial complexes are associated with the development of atrial arrhythmias in patients undergoing autologous hematopoietic stem cell transplantation (HSCT) [10].

Fluoropyrimidines, known for their coronary toxicity, can cause angina, ischemia-related ECG changes, hypertension, Takotsubo syndrome, myocardial infarction (even in patients with normal coronary arteries), and, less commonly, myocarditis, arrhythmias, or peripheral arterial events, such as Raynaud’s phenomenon and ischemic stroke. Baseline cardiovascular risk assessment, including blood pressure measurement, ECG, lipid profile, HbA1c, and SCORE2/SCORE2-OP or equivalent, is recommended before initiating therapy. A baseline echocardiogram is advised for patients with a history of symptomatic cardiovascular disease, and screening for coronary artery disease may be considered in those at high or very high risk. Close clinical follow-up and management of modifiable risk factors are critical throughout treatment. Figure 3 illustrates an example of fluoropyrimidine-related cardiotoxicity [166].

Although specific guideline-based recommendations are lacking for taxanes and vinca alkaloids, these agents can potentiate LVSD, particularly when used after or in combination with anthracyclines or anti-HER2 therapies. Attention should also be paid to their potential for vasospasm and conduction abnormalities, warranting a cautious approach in high-risk patients. Surveillance protocols for ADCs are emerging, but recommendations generally align with those of their cytotoxic payloads, particularly if anthracycline-based. CAR-T and tumor-infiltrating lymphocyte (TIL) therapies, particularly in the presence of cytokine release syndrome (CRS), pose a significant risk for LV dysfunction. Troponin and NT-proBNP levels, along with echocardiogram, should be assessed at baseline, especially in patients with pre-existing cardiac disease. These patients require close cardiac monitoring during and after CRS, and treatment should be paused in the event of significant clinical deterioration. Therapies, such as EGFRi and ALKi, may cause QTc prolongation and conduction abnormalities, necessitating regular ECG and lipid monitoring, as well as echocardiography in specific cases, such as osimertinib use [167,168]. CDK4/6 inhibitors have also been implicated in QTc prolongation, especially ribociclib, which warrants ECG monitoring at two and four weeks after treatment initiation and with each dose escalation [169]. Hormonal therapies, including androgen deprivation therapy and anti-estrogen agents, are linked to hypertension, ischemia, and LVSD [170,171]. Therefore, baseline and periodic cardiovascular risk stratification using SCORE2/SCORE2-OP is recommended, along with serial ECGs for QTc assessment in at-risk patients [10,172,173]. In experimental models, it has been observed that both tamoxifen and toremifen, when administered at high doses, can induce a prolongation of the QTc interval through direct inhibition of the fast-acting delayed potassium rectifier (IKr) increasing the duration of the action potential and potentially promoting torsade pointes ventricular polymorphic tachycardia. The clinical significance of this adverse effect is limited; however, as in patients treated with the standard dose of 20 mg/day of tamoxifen, this effect has not been detected [174].

PIs, the backbone of combination treatments for patients with multiple myeloma and AL amyloidosis, while also indicated in Waldenström’s macroglobulinemia and other malignancies, particularly carfilzomib, are associated with cardiovascular toxicities, such as heart failure, hypertension, and ischemic events, likely due to endothelial dysfunction and oxidative stress [175]. A baseline cardiovascular assessment is recommended for all patients starting PIs, including blood pressure measurement, ECG, echocardiography (with specific evaluation for amyloidosis when applicable), natriuretic peptides, and troponins, especially in high- or very high-risk patients. Biomarkers should be reassessed every cycle for the first six cycles in patients receiving carfilzomib or bortezomib and every 3–6 months in those with AL-cardiac amyloidosis. Routine echocardiographic surveillance every three cycles is advised in high-risk patients on carfilzomib. Early signs of dysfunction (e.g., GLS decline, biomarker elevation) may warrant treatment modification or cardioprotective strategies. Bortezomib and ixazomib show a lower cardiotoxic profile but still require attention to blood pressure and volume status [10].

Beyond systemic anticancer therapies, thoracic RT represents another major contributor to cardiovascular toxicity in oncology. Thoracic RT is associated with delayed cardiovascular effects, including, notably, accelerated atherosclerosis. Surveillance strategies include long-term follow-up with emphasis on cardiovascular risk factor control and screening beginning five to ten years post-treatment, especially in high-risk individuals. In long-term survivors, particularly those treated in childhood or young adulthood, late-onset cardiotoxicity can emerge years to decades after therapy. Surveillance guidelines recommend periodic reassessment of cardiac function every 5 years or more frequently in high-risk individuals using echocardiography and biomarkers [10]. Survivors exposed to chest irradiation or high cumulative anthracycline doses require life-long follow-up due to the risk of valvular disease, coronary artery disease, and restrictive cardiomyopathy [176].

## 5. The Role of an Integrated Cardio-Oncology Approach

Cardio-oncology has evolved from a reactive, consult-based service into a complex, multidisciplinary discipline requiring dedicated infrastructure, specialized training, and continuous collaboration between cardiologists, oncologists, hematologists, and allied health professionals. At its core, cardio-oncology aims to manage and prevent CVD across the continuum of cancer care—from diagnosis to survivorship—by aligning oncologic efficacy with cardiovascular safety. This objective is achieved through a structured and proactive model of care incorporating personalized risk stratification, early detection of cardiotoxicity, and evidence-based cardioprotection.

A truly integrated cardio-oncology service operates not merely as a series of cardiologic evaluations but as a defined and recognized hospital-based unit. It delivers standardized activities—including ambulatory consultations, ECG and echocardiographic interpretation, bedside assessments, and cardiac imaging reviews—and relies on the collaboration of cardiologists, nurses, trainees, and clinical pharmacists. To support this evolving field, the European Society of Cardiology has issued a Core Curriculum that outlines progressive levels of clinical competency, from observation to full autonomy, based on the CanMEDS framework. This structured training ensures that both specialists and general practitioners involved in cancer care are equipped with the necessary cardiovascular knowledge and skills to manage therapy-related complications effectively [177].

The cardio-oncologist’s role extends beyond surveillance of therapy-related cardiotoxicity. It includes managing patients with pre-existing CVD who require potentially cardiotoxic treatment, addressing the cardiovascular impact of the malignancy itself, and implementing long-term survivorship plans, especially in younger patients at risk of late-onset cardiotoxicity. As such, cardio-oncology must be embedded within a multidisciplinary network, fostering continuous dialogue between oncologists and cardiologists through regular meetings and shared protocols. This collaboration facilitates risk–benefit evaluations, individualized treatment plans, and the development of context-specific surveillance strategies.

Moreover, institutional support is essential for the success of cardio-oncology programs. Hospitals must commit to providing dedicated space, staff, and resources to build sustainable services.

Ultimately, an integrated cardio-oncology approach is not only clinically necessary but structurally imperative. It bridges the gap between two traditionally siloed disciplines, enhances patient outcomes, reduces treatment interruptions, and promotes the development of national and international care models.

## 6. Conclusions

Cardiotoxicity has emerged as a major challenge in modern oncology, with increasing numbers of patients affected by cardiovascular complications from both traditional and novel cancer therapies. This evolving scenario calls for a proactive, structured, and multidisciplinary cardio-oncology approach integrated into every phase of the cancer care continuum. Timely cardiovascular risk assessment, evidence-based surveillance protocols, and close collaboration between cardiologists and oncologists are essential to optimize outcomes, avoid treatment interruptions, and ensure long-term survivorship. The development of dedicated cardio-oncology services, supported by institutional commitment and standardized training pathways, is now a clinical necessity.

To meet the growing demands of this field, cardio-oncology must evolve into a recognized and standardized component of comprehensive cancer care—bridging expertise, improving coordination, and, ultimately, safeguarding both the heart and the cure.

## Figures and Tables

**Figure 1 pharmaceuticals-18-01399-f001:**
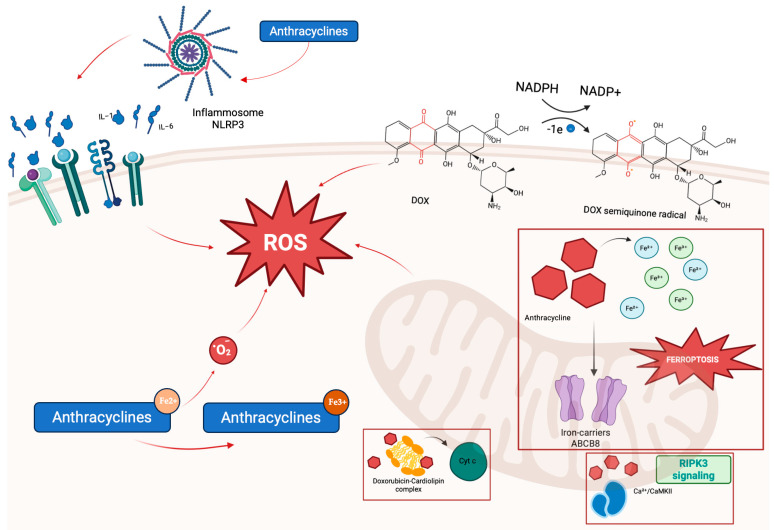
Molecular mechanisms of anthracycline-induced cardiotoxicity. ABCB8, ATP-binding cassette, subfamily B, member 8; CaMKII, calmodulin-dependent protein kinase II; Cyt C, Cytochrome C; DOX, doxorubicin; IL-1, interleukin-1; IL-6, interleukin-6; NLRP3, NOD-, LRR-, and pyrin domain-containing protein 3 (NLRP3), RIPK3, receptor-interacting protein kinase 3; ROS, reactive oxygen species.

**Figure 2 pharmaceuticals-18-01399-f002:**
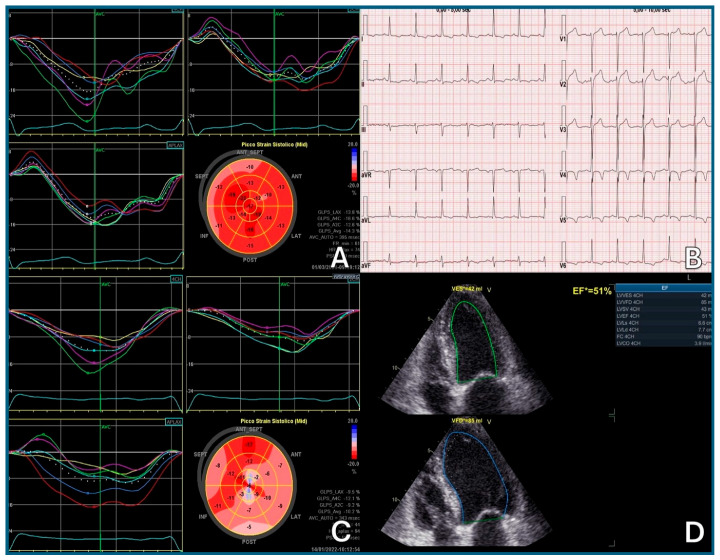
Cardiotoxicity associated with anti-VEGF therapy and taxane-based treatment in a 66-year-old woman with hypertension and a history of recurrent ER−/HER2− breast cancer. (**A**) Baseline strain analysis before starting paclitaxel plus bevacizumab, showing reduced GLS (−14%) with preserved LVEF (55%). (**B**) ECG during an episode of chest pain and dyspnea in February 2022, with elevated troponin and NT-proBNP, showing negative T waves in lateral leads and ST elevation in V1-V3. (**C**) Strain imaging at the time of symptoms showed further GLS impairment (−10%) and reduced LVEF (33%) (not explicit in the image). (**D**) One-year follow-up after heart failure therapy shows full recovery: LVEF 51%, GLS improved to −16%. ECG: electrocardiogram; ER: estrogen receptor; GLS: global longitudinal strain; HER2: human epidermal growth factor receptor 2; LVEF: left ventricular ejection fraction; NT-proBNP: N-terminal pro-B-type natriuretic peptide; VEGF: vascular endothelial growth factor.

**Figure 3 pharmaceuticals-18-01399-f003:**
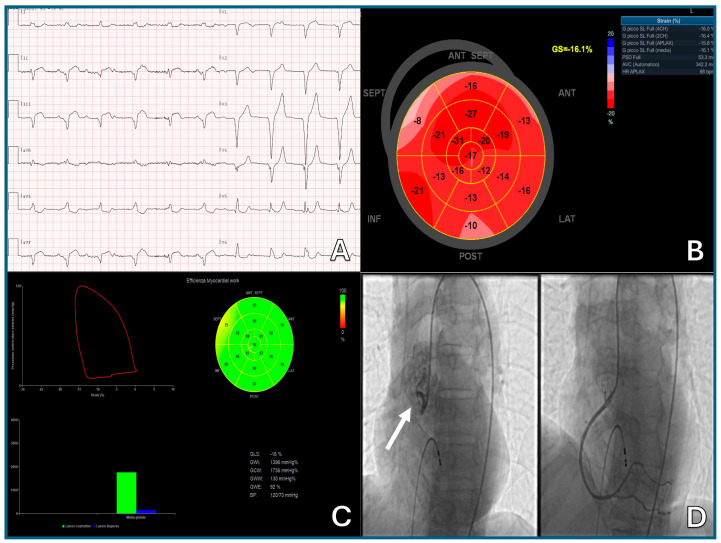
Acute coronary syndrome related to fluoropyrimidine and anti-VEGF therapy in a 75-year-old male with colon cancer, previously treated with colectomy followed by FOLFIRI plus bevacizumab and subsequently capecitabine plus bevacizumab. (**A**) ECG on admission shows inferior STEMI with complete AV block and junctional escape rhythm with Left Bundle Branch Block pattern, which makes it difficult to value ST elevation in inferior leads. (**B**,**C**) Echocardiographic strain and myocardial work analysis reveal basal inferior and septal wall akinesia with preserved LVEF (55%). (**D**) Coronary angiography demonstrates proximal RCA occlusion (arrow), successfully treated with PCI and DES. AV: atrioventricular; DES: drug-eluting stent; ECG: electrocardiogram; FOLFIRI: folinic acid/5-fluorouracil/irinotecan; LVEF: left ventricular ejection fraction; PCI: percutaneous coronary intervention; RCA: right coronary artery; STEMI: ST-elevation myocardial infarction; VEGF: vascular endothelial growth factor.

**Table 1 pharmaceuticals-18-01399-t001:** Cardiovascular toxicities of target therapy: molecular targets, examples, and mechanisms. ALK, anaplastic lymphoma kinase; AV, atrioventricular; CV, cardiovascular; EGFR, Epidermal growth factor receptor; LVEF, left ventricular ePDGFRs, platelet-derived growth factor receptors; VEGFRs, vascular endothelial growth factor receptors; The upward arrow (↑) indicates an increase, the downward arrow (↓) indicates a decrease, while the rightward arrow (→) represents a consequence or result.

Molecular Target	Examples	Cardiovascular Toxicities (Incidence)	Mechanistic Notes
ERBB2 (HER2)	Trastuzumab, Pertuzumab, Lapatinib, Tucatinib, Afatinib, Neratinib, Dacomitinib	Heart failure: 2–5% (trastuzumab), up to 28% with anthracycline combination; LVEF reduction: 1–5% (TKIs)	Inhibition of NRG1-ERBB2/ERBB4 signaling → impaired cardiomyocyte survival, proliferation, and contractility; oxidative stress, mitochondrial dysfunction; reversible in most cases
VEGFRs, PDGFRs	Sunitinib, Sorafenib, Axitinib, Vandetanib, Cabozantinib, Lenvatinib, Pazopanib, Ponatinib, Regorafenib	Hypertension (up to 47%), LV dysfunction (≈28%), heart failure (≈8%)	VEGFR inhibition → ↓ NO → hypertension; capillary rarefaction → impaired myocardial perfusion; PDGFR blockade → pericyte loss → microvascular dysfunction
Bcr-Abl, c-Kit, c-Abl, PDGFRs	Imatinib	Heart failure (especially in elderly)	ER stress, mitochondrial damage, impaired cardiac progenitor cell function
Mutant Bcr-Abl, PDGFRs	Dasatinib, Nilotinib, Bosutinib, Ponatinib	Occlusive arterial disease ↑ risk	Designed to overcome BCR-ABL resistance; CV risk elevated vs. imatinib
EGFR	Erlotinib, Gefitinib, Afatinib, Osimertinib	QTc prolongation, arrhythmias, reduced LVEF, heart failure	PI3K pathway inhibition, HER2 inhibition, oxidative stress, mitochondrial dysfunction, autophagy/apoptosis dysregulation
ALK	Crizotinib, Ceritinib, Alectinib, Brigatinib, Lorlatinib	Sinus bradycardia, AV block, QTc prolongation, hypertension, hyperglycemia, dyslipidemia	Long-term CV effects; generally well-tolerated but important due to extended survival

**Table 2 pharmaceuticals-18-01399-t002:** HFA-ICOS baseline cardiovascular toxicity risk stratification according to 2022 ESC Guidelines.

Baseline CV Toxicity Risk Factors	Anthracycline Chemotherapy	HER2-Targeted Therapies	VEGF Inhibitors	BCR-ABL Inhibitors	Multiple Myeloma Therapies	RAF and MEK Inhibitors
**Previous CVD**						
HF/cardiomyopathy/CTRCD	VH	VH	VH	H	VH	VH
Severe VHD	H	H	-	-	-	H
MI or PCI or CABG	H	H	VH	-	-	H
Stable angina	H	H	VH	-	-	H
Arterial vascular disease	-	-	VH	VH	VH	-
Abnormal ankle-brachial pressure index	-	-	-	H	-	-
PH	-	-	-	H	-	-
Arterial thrombosis with TKI	-	-	-	VH	-	-
Venous thrombosis (DVT/PE)	-	-	H	M2	VH	-
Arrhythmia	-	M2	M2	M2	M2	M1
QTc ≥ 480 ms	-	-	H	H	-	-
450 ≤ QTc, 480 ms (men), 460 ≤ QTc < 480 ms (women)	-	-	M2	M2	-	-
Prior PI CV toxicity	-	-	-	-	VH	-
Prior IMiD CV toxicity	-	-	-	-	H	-
**Cardiac imaging**						
LVEF < 50%	H	H	H	H	H	H
LVEF 50–54%	M2	M2	M2	-	M2	M2
LV hypertrophy	-	-	-	-	M1	-
Cardiac amyloidosis	-	-	-	-	VH	-
**Cardiac biomarkers**						
Elevated baseline cTn	M1	M2	M1	-	M2	M2
Elevated baseline NP	M1	M2	M1	-	H	M2
**Age and CVRF**						
Age ≥ 80 years	H	H	-	-	-	M1
Age 65–79 years	M2	M2	-	-	-	M1
Age ≥ 75 years	-	-	H	H	H	M1
Age 65–74 years	-	-	M1	M2	M1	M1
Age ≥ 60 years	-	-	-	M1	-	-
CVD 10-year risk score > 20%	-	-	-	H	-	-
Hypertension	M1	M1	H	M2	M1	M2
Chronic kidney disease	M1	M1	M1	M1	M1	M1
Proteinuria	-	-	M1	-	-	-
DM	M1	M1	M1	M1	M1	M1
Hyperlipidemia	-	-	M1	M1	M1	-
Family history of thrombophilia	-	-	-	M1	M1	-
**Current cancer treatment**						
Dexamethasone > 160 mg/month	-	-	-	-	M1	-
Includes anthracycline before HER2-targeted therapy	-	M1	-	-	-	-
**Previous exposure to**						
Anthracycline	H	M2	H	-	H	H
Trastuzumab	-	VH	-	-	-	-
RT to left chest or mediastinum	H	M2	M1	-	M1	M2
Non-anthracycline chemotherapy	M1	-	-	-	-	-
**Lifestyle risk factors**						
Current smoker or significant smoking history	M1	M1	M1	H	M1	M1
Obesity (BMI > 30 kg/m^2^)	M1	M1	M1	M1	M1	M1

BCR-ABL, breakpoint cluster region–Abelson oncogene locus; BMI, body mass index; CABG, coronary artery bypass graft; cTn, cardiac troponin; CTRCD, cancer-therapy-related cardiac dysfunction; CV, cardiovascular; CVD, cardiovascular disease; CVRF, cardiovascular risk factors; DM, diabetes mellitus; DVT, deep vein thrombosis; H, high risk; HER2, human epidermal receptor 2; HF, heart failure; IMiD, immunomodulatory drugs; LV, left ventricular; LVEF, left ventricular ejection fraction; M, moderate risk; MEK, mitogen-activated extracellular signal-regulated kinase; MI, myocardial infarction; MM, multiple myeloma; NP, natriuretic peptides (including BNP and NT-proBNP); NT-proBNP, N-terminal pro-B-type natriuretic peptide; PCI, percutaneous coronary intervention; PE, pulmonary embolism; PH, pulmonary hypertension; PI, proteasome inhibitors; QTc, corrected QT interval; RAF, rapidly accelerated fibrosarcoma; RT, radiotherapy; TKI, tyrosine kinase inhibitors; VEGFi, vascular endothelial growth factor inhibitors; VH, very high risk; VHD, valvular heart disease. Risk level: Low risk = no risk factors OR one moderate 1 risk factor; moderate risk (M) = moderate risk factors with a total of 2–4 points (Moderate 1 [M1] = 1 point; Moderate [M2] = 2 points); high risk (H) = moderate risk factors with a total of ≥5 points OR any high-risk factor; very high risk (VH) = any very high risk factor. Points are based on HFA-ICOS risk score. The “-“ symbol indicates either a lack of available data or that no proven or unproven relationship has been established. Taken and adapted from Lyon et al. (2022) [10] and with permission.

**Table 3 pharmaceuticals-18-01399-t003:** Classification of cancer-therapy-related cardiac dysfunction (CTRCD) according to the 2022 ESC Guidelines.

Severity	Asymptomatic CTRCD	Symptomatic CTRCD
Mild	LVEF ≥ 50% AND new relative decline in GLS by >15% from baseline AND/OR new rise in cardiac biomarkers	Mild HF symptoms, no intensification of therapy required
Moderate	New LVEF reduction by ≥10 percentage points to an LVEF of 40–49% OR new LVEF reduction by <10 percentage points to an LVEF of 40–49% AND either new relative decline in GLS by >15% from baseline OR new rise in cardiac biomarkers	Need for outpatient intensification of diuretic and HF therapy
Severe	New LVEF reduction to < 40%	Requires hospitalization for heart failure
Very severe	-----	Heart failure requiring inotropic support, mechanical circulatory assistance, or heart transplantation

Adapted from Lyon AR et al., 2022 [10] ESC Guidelines on cardio-oncology developed in collaboration with the European Hematology Association (EHA), the European Society for Therapeutic Radiology and Oncology (ESTRO), and the International Cardio-Oncology Society (IC-OS). *Eur. Heart J.* 2022. CTRCD, cancer-therapy-related cardiac dysfunction; GLS, global longitudinal strain; HF, heart failure; LVEF, left ventricular ejection fraction.

## Data Availability

No new data were created or analyzed in this study.

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
