# Peer review of "Cardiotoxicity Induced by Anticancer Therapies: A Call for Integrated Cardio-Oncology Practice"

_pharmaceuticals, 2025, doi:10.3390/ph18091399_

Round 1
Reviewer 1 Report
Comments and Suggestions for Authors
A very reliable and detailed manuscript. The authors objectively and thoroughly described various aspects of the cardiotoxicity of oncological drugs and proposed modern ways to monitor these adverse effects. Recommended pharmacological intervention methods in cases of cardiotoxicity were also described. Furthermore, the authors emphasized the necessity for further development of cardio-oncology and the importance of this medical field for the safety of oncological treatment.
Author Response
Comment: A very reliable and detailed manuscript. The authors objectively and thoroughly described various aspects of the cardiotoxicity of oncological drugs and proposed modern ways to monitor these adverse effects. Recommended pharmacological intervention methods in cases of cardiotoxicity were also described. Furthermore, the authors emphasized the necessity for further development of cardio-oncology and the importance of this medical field for the safety of oncological treatment.
Response: We thank you very much for your positive feedback. We are glad that our work on cardiotoxicity, monitoring strategies, and pharmacological interventions was appreciated, and we fully share your view on the importance of further developing the field of cardio-oncology.
Reviewer 2 Report
Comments and Suggestions for Authors
It is known that despite the development and introduction of new cancer treatment methods and a significant improvement in patient survival, the currently used anticancer drugs have an increase in cardiovascular side effects. Such complications have become one of the most common and serious consequences, as they can have a negative impact not only on the health of the cardiovascular system, but also on the continuity and effectiveness of cancer treatment, thereby affecting overall survival.
The article "Cardiotoxicity Induced by Anticancer Therapies: A Call for Integrated Cardio-Oncology Practice" by Giuliana Ciappina et al. expresses the opinion that by summarizing current data, critical knowledge gaps are identified and an interdisciplinary, evidence-based system is proposed to guide the prevention, early detection and optimal treatment of cardiotoxicity associated with antitumor therapy.
The degree of relevance of the article provided is beyond doubt, since this article presents a thorough analysis of various aspects related to such a serious problem as cardiotoxicity in modern oncology. The author has succeeded in achieving the main goal of this manuscript — a comprehensive review of cardiovascular diseases associated with antitumor therapy, emphasizing its growing clinical significance against the background of increasing cancer survival rates.
The introduction contains sufficient information, and the design of the search and data analysis is consistent with the goals set. The main text of the article is well structured and divided into relevant information blocks, and contains an in-depth study of cardiotoxicity associated with cancer therapy. The work supports a coordinated, interdisciplinary treatment model and addresses the problem of early detection, risk assessment, prevention and treatment of cardiotoxicity. The conclusion reflects the main ideas of the review.
The manuscript is written in good scientific language, the authors have conducted a thorough analysis of the literature data and this review will be of interest to the audience of the journal Pharmaceuticals
The manuscript can be published in the journal "Pharmaceuticals" after revision:
- Page 2: The heading "Therapeutic Agents Associated with Cardiotoxicity" should be highlighted as "2. Therapeutic Agents Associated with Cardiotoxicity", so it gets lost in the text, and then there is a numbering of other subheadings related to this heading.
- Subtitle 2.1.2 has a different formatting from the others.
- "2.1. Cardiotoxicity induced by targeted therapy" → "2.2. Cardiotoxicity induced by targeted therapy".
- The headings on pages 11, 16, 25 need to be numbered, as other subheadings related to this heading continue to be numbered.
- The list of references has a strange numbering.
- Introduction: How was the reported CVD mortality related to cancer survivors/cancer patients? Why can't it be, for example, due to genetic or environmental factors? This should be spelled out more clearly.
- Some words/phrases have dashes/long dashes instead of hyphens: for example, "cells—mediates", "o fluoropyrimidines—including cardiac events—occur", that is, it can occur in the text like this: "receptor–positive, HER2-negative".
- To the previous remark: on page 16 there is a sentence "It is especially useful for detecting early signs of cardiotoxicity—including inflammation, edema, and fibrosis—even in asymptomatic patients or those with preserved ejection fraction [113]." To be honest, I don't quite understand why there are dashes between the words... perhaps in the examples mentioned earlier in paragraph 8, neither hyphens nor dashes are needed either... In the sentence "The asymptomatic form is subdued into mild and severe, while the symptomatic form—clinically manifest heart failure—may range from mild to very severe." I can assume that the first is still a dash, but it must be separated by a space on both sides.
- Table 1: the transcriptions in the notes should be divided into two groups: the first is the transcription of the abbreviations that are in the header of the table, the second is the transcription of the abbreviations that are inside the table. And also, what does the digital postscript to M (moderate risk) mean? what is the meaning of the odd number 1 or 2? You can also specify what "-" means. Is this a lack of data at all, or is there no proven/unproven relationship?
Thus, this review, despite some shortcomings, highlights the potential for developing timely cardiovascular risk assessment, evidence-based surveillance protocols, and close collaboration between cardiologists and oncologists, which is crucial to prevent treatment interruptions and ensure long-term survival.
Author Response
Comment 1: Page 2: The heading "Therapeutic Agents Associated with Cardiotoxicity" should be highlighted as "2. Therapeutic Agents Associated with Cardiotoxicity", so it gets lost in the text, and then there is a numbering of other subheadings related to this heading.
Comment 2: Subtitle 2.1.2 has a different formatting from the others.
Comment 3: "2.1. Cardiotoxicity induced by targeted therapy" → "2.2. Cardiotoxicity induced by targeted therapy".
Comment 4: The headings on pages 11, 16, 25 need to be numbered, as other subheadings related to this heading continue to be numbered.
Response to comments 1,2,3,4: Thank you for the suggestions. The issue with the heading numbering was due to a formatting error. We have corrected the formatting and renumbered all headings and subheadings accordingly.
Comment 5: The list of references has a strange numbering.
Response 5: We have corrected the reference list, ensuring that it now follows the journal’s guidelines for numbering and formatting.
Comment 6: Introduction: How was the reported CVD mortality related to cancer survivors/cancer patients? Why can't it be, for example, due to genetic or environmental factors? This should be spelled out more clearly.
Response 6: Thank you for the suggestion. We have revised the text to clarify that CVDs are a leading cause of morbidity and mortality in cancer survivors, while acknowledging that other factors may also contribute.
Comment 7: Some words/phrases have dashes/long dashes instead of hyphens: for example, "cells—mediates", "o fluoropyrimidines—including cardiac events—occur", that is, it can occur in the text like this: "receptor–positive, HER2-negative".
Response 7: We have corrected the text to replace long dashes with proper hyphens where appropriate, ensuring consistent usage throughout the manuscript.
Comment 8: To the previous remark: on page 16 there is a sentence "It is especially useful for detecting early signs of cardiotoxicity—including inflammation, edema, and fibrosis—even in asymptomatic patients or those with preserved ejection fraction [113]." To be honest, I don't quite understand why there are dashes between the words... perhaps in the examples mentioned earlier in paragraph 8, neither hyphens nor dashes are needed either... In the sentence "The asymptomatic form is subdued into mild and severe, while the symptomatic form—clinically manifest heart failure—may range from mild to very severe." I can assume that the first is still a dash, but it must be separated by a space on both sides.
Response 8: We have revised the manuscript to correct the use of dashes and hyphens. In particular, long dashes have been replaced or spaced appropriately, and unnecessary dashes have been removed from examples where they were not needed, ensuring consistency and clarity throughout the text. Thank You.
Comment 9: Table 1: the transcriptions in the notes should be divided into two groups: the first is the transcription of the abbreviations that are in the header of the table, the second is the transcription of the abbreviations that are inside the table. And also, what does the digital postscript to M (moderate risk) mean? what is the meaning of the odd number 1 or 2? You can also specify what "-" means. Is this a lack of data at all, or is there no proven/unproven relationship?
Response 9: Thank you for the suggestion. We have revised Table 1 to separate the transcriptions of abbreviations into two groups: those appearing in the table headers and those within the table content. Additionally, we have clarified all symbols
Reviewer 3 Report
Comments and Suggestions for Authors
Dear Editor of Pharmaceuticals - MDPI
(see the pdf to verify the red marks text)
Thank you for inviting me to be a Referee for this scientific article: ‘Cardiotoxicity Induced by Anticancer Therapies: A Call for Integrated Cardio-Oncology Practice’.
Although I have transcribed my opinion of what is requested in the Pharmaceuticals - MDPI electronic form, I would like to add a few comments.
- There are also some aspects that deserve to be changed, which are referenced in red: (in red proposals to be potentially changed and also to be praised).
Title: Cardiotoxicity Induced by Anticancer Therapies: A Call for Integrated Cardio-Oncology Practice
Abstract – no proposed change
Introduction
Page 2
- Where it reads: “Therapeutic Agents Associated with Cardiotoxicity”; remove.
Page 3
- Cited further ahead – where it reads “…which increases with cumulative dosage and can lead to progressive cardiomyopathy and heart failure [14]. It is worth noting… “The risk of anthracycline-induced heart failure increases as the cumulative dose administered increases: 3–5% with 400 mg/m2 and as high as 18–48% at 700 mg/m2…” as can be read in the citation [14] of this original article (or quote further ahead)
- “the absolute number of affected individuals remains substantial [15].”; published numbers? – noting/objectify.
- objectify with the however of the citation [1 "Hypertension (OR: 1.99; 95% CI: 1.43–2.76), diabetes mellitus (OR: 1.74; 95% CI: 1.11–2.74), and obesity (OR: 1.72; 95% CI: 1.13–2.61) were associated with an increased risk of cardiotoxicity.
- According to the authors of the citation [18] "Currently, pharmacogenomic testing of all childhood cancer patients with an indication for doxorubicin or daunorubicin therapy for RARG rs2229774, SLC28A3 rs7853758, and UGT1A6*4 rs17863783 variants is recommended. There is no recommendation regarding testing in adults."
- “Notable variants include those in CBR1 and CBR3, encoding carbonyl reductases, ABCC1 and ABCC5, related to ATP-binding cassette transporters, and SLC22A, a solute carrier gene, which are predominantly implicated in pediatric populations and patients treated for acute lymphoblastic leukemia.” – Quote?
- The citation [19] highlights “study indicated 6 variants obviously associated with the increased risk for CIC, including CYBA rs4673 (pooled odds ratio, 1.93; 95% CI, 1.13-3.30), RAC2 rs13058338 (2.05; 1.11-3.78), CYP3A5 rs776746 (2.15; 1.00-4.62) ABCC1 rs45511401 (1.46; 1.05-2.01), ABCC2 rs8187710 (2.19; 1.38-3.48), and HER2-Ile655Val rs1136201 (2.48; 1.53-4.02).”, see: not the polymorphisms listed above. - This text lacks a caption for the acronyms used (CBR1 and CBR3, ABCC1 and ABCC5, etc.) and whenever these acronyms for genetic polymorphisms are used (at least once…).
- I believe the authors of the citation [19] make no reference to NOS3 (so this should be changed)...and the citations should be renumbered....
Page 4
- How to reconcile these different data in the text? Perhaps using a citation [28]. Therefore: "Cardiac damage primarily involves alterations in myocardial cell function and pathological cell death, encompassing mitochondrial dysfunction, topoisomerase inhibition, disruptions in iron ion metabolism, myofibril degradation, and oxidative stress." , this is a possible summary of what is currently known about the pathophysiology of myocardial injury caused by anthracyclines…
- NLRP-3: NLR family pyrin domain-containing 3 (NLRP-3)…”
- “Given mitochondria constitute approximately 36–40%…” given (without bold;
- “ABCB8”??
- receptor-interacting protein 3 (RIP3)…” of the mitochondrial permeability transition pore and necroptotic cell death [28].”; demonstrated in mice; in fact, several of the experimental findings on the mechanisms involved in the cardiotoxicity caused by anthracyclines are obtained in animal experiments, and this should be included in the submitted text.
- I don't know if it's possible to use related figures (constructive criticism) in the current article, similar to those used by Xie et al. 2024... sketches of figures 2 and .. or others from the same article... modifying... altering and adapting... obviously altered (since this is a very long article... why not provide a better schematic basis).
- "Parkin"?; Parkin (PARK2), a Parkinson's disease-associated gene, is a potential tumor suppressor whose expression is frequently diminished in tumors;
Page 5
- but the citation [32] does not address "In addition to cardiomyocytes, anthracyclines adversely affect non-myocyte cardiac cells, including fibroblasts, endothelial cells, vascular smooth muscle cells, and immune cells [32]"; but rather "We conclude that extracellular vesicles from breast cancer cells mediate skeletal muscle mitochondrial dysfunction in cancer and may contribute to muscle weakness in some cancer patients." this aspect must be corrected,.
- "Doxorubicin compromises endothelial barrier function, increasing vascular permeability and promoting myocardial interstitial edema [33]."; but in this article, "cell culture... mice...".
- "These cellular and molecular alterations, combined with activation of innate immune pathways, drive maladaptive remodeling of the left ventricle [34].
FROM HERE ON, THE AUTHORS OF THE CURRENT ARTICLE INSERT A LONG TEXT WITHOUT ANY INTERLINK TO THE CARDIOTOXICITY RELATED TO THE USE OF ANTHRACYCLINES; Therefore, this text should be removed (due to the absence of any pathophysiological correlation with anthracyclines, or a logical and sequential interconnection should be made with the use of anthracyclines… and… radiotherapy… and microbiome alterations…
Therefore, all content should be revised or properly integrated… from the text that begins with “Radiotherapy remains an indispensable modality in the treatment of thoracic malignancies such as breast cancer and mediastinal lymphomas. Nevertheless… to “… efficacy of agents like doxorubicin while reducing their cardiotoxic footprint. This dual action highlights a promising avenue for microbiome-targeted strategies within cardio oncology [40].”; this content should be removed or integrated with cardiotoxicity related to anthracyclines… renumber the citations; alternatively, create new chapters (RT…
Page 6:
2.1.2. Fluoropyrimidin
Some inconsistencies or errors:
- Fluoropyrimidin
- Where it reads: "cancers, including gastrointestinal and genitourinary tumors [42]. Even though fluoropyrimidines are commonly regarded as tolerable, they can provoke cardiovascular side effects in a notable proportion of patients, affecting between 1% and 18% [6]." These figures are according to Morelli et al. [6]; but only in this type of cancer? According to Deac et al. [43], "head and neck cancer, breast cancer, esophageal cancer, gastric cancer, pancreatic cancer, and colorectal cancer"; (see, for example, Shiga and Hiraide, 2020 – citation [42] of the current article; 1 and 18%?; The same authors, Jurcyzk [41] Shiga and Hiraide, 2020 – citation [42] in Table 1 of their article indicate an incidence that can reach up to 34.6%. (review this content). - since this is a very extensive article, it was worth briefly describing the clinical aspects potentially linked to cardiotoxicity as well as better describing the cardiotoxic effect and cellular and molecular mechanisms of cardiotoxicity, as can be seen in Table 1 of the cited article [6].
- Where it reads “Four key loss-of-function DPYD alleles—*2A, *13, c.2846A>T, and HapB”; can be read DPYD*2A (IVS14+1G>A, c.1905+1G>A), DPYD*9B (c.2846A>T), DPYD*13 (c.1679T>G), and HapB3 (c.1129-5923C>G) [43].
-Where it reads: “…marked increases in fluoropyrimidine toxicities, with relative risks between 1.7 and over 4 for severe adverse events [44]”; should be added “ “Carriers of DPYD variants were found to be significantly correlated with treatment-related mortality (OR = 34.86, 95% CI 13.96–87.05; p < 0.05)”.
- 2.1.3. taxanes taxanes,
- …”including paclitaxel…”; docetaxel, paclitaxel and albumin-bound paclitaxel (nab-paclitaxel)
- Where it reads: “…occurring in approximately 3% to 20% of patients [48]”; [6] and not [48].
Page 7:
- Where it reads: “Consequently, prophylactic administration of corticosteroids and antihistamines is often recommended [50]”. Quote [50]or [6]
- Where it reads: “…opening of mitochondrial permeability transition pores, and collapse of mitochondrial membrane potential, all contributing to cardiomyocyte injury [51].”; is this content actually in the content of the article cited as [51]?
2.1.4. Alkylating Drugs
- where it reads “…myocardial infarction, with reported prevalence ranging from 7% to 32% [54].”; once again, [54] or Morelli et al. [6] should be cited, which in turn cites Alexandre J, Moslehi JJ, Bersell KR, Funck-Brentano C, Roden DM, Salem J-E. Anticancer drug-induced cardiac rhythm disorders: current knowledge and basic underlying mechanisms. Pharmacol Ther. (2018) 189:89–103. doi: 10.1016/j.pharmthera.2018.04.009
- Where it reads "...including the arachidonic acid cascade and NF-κB signaling, leading to inflammatory cytokine release, cardiac tissue remodeling, and structural deterioration of the myocardium [56]. " This content does not exist in the article cited as [56] Sharifiaghdam, Z.; Amini, S. M.; Dalouchi, F.; Behrooz, A. B.; Azizi, Y. Apigenin-Coated Gold Nanoparticles as a Cardioprotective Strategy against Doxorubicin-Induced Cardiotoxicity in Male Rats via Reducing Apoptosis. Heliyon 2023, 9 (3), e14024. (Review this citation); the reference to NFkB is in fact included in Table 1 of the citation [6] “Expression of proinflammatory chemokines and cytokines driven by increased NFkB activation”-
- Where it reads "...practice has significantly reduced the frequency of such severe outcomes [57]."; but in the cited article [57] there is no reference to heart failure or myocardial infarction. "Clinical benefits of PTCy dose reduction may include faster engraftment and T-cell recovery and less severe BK virus cystitis/urethritis and mucositis."
2.1. Cardiotoxicity induced by targeted therapy; 2.2. and not 2.1.1-
- An introductory explanation is missing "see, for example, Morelli et al. [6] "After the binding of soluble ligands, ERBB kinase receptors arrange in homo- or heterodimer complexes, which activate the tyrosine kinase activity and the consequent signaling events leading to the modulation of cell survival, proliferation, migration, and differentiation [reviewed in (104–107)]. ERBB2 (also known as HER2) receptor is a proto-oncogene frequently amplified and overexpressed in many human cancers.
2.2.1. Anti-HER2
- Where it reads “…neuregulin-1 (NRG1)/ERBB2 signaling pathway essential for cardiac function [58].”; but in what ways is it essential? It would be worthwhile to develop NRG1 signaling a bit, as for example, Moreli et al. do, citing several authors to be cited if this proposal is accepted. “NRG1, together with its tyrosine kinase receptors ERBB4, ERBB3, and ERBB2, is essential for heart development (113–115) [reviewed in (110, 116, 117)] and tunes heart regenerative, inflammatory, fibrotic, and metabolic processes (118, 119) [reviewed in (110, 117, 120–122)]. In cardiomyocytes, the most prominently expressed NRG1 receptors are ERBB4 and ERBB2 (123), and NRG1 stimulates fetal/neonatal cardiomyocyte proliferation, hypertrophy, sarcomerogenesis, and survival (114, 115, 124, 127) [reviewed in (110, 116, 117, 120, 121)].”
Page 8
2.2.2. TKIs
- It may be worth taking advantage of the content of the citation [64] Nine VEGFR-TKIs (Axitinib, Vandetanib, Cabozantinib, Lenvatinib, Pazopanib, Ponatinib, Regorafenib, Sunitinib, Sorafenib)…
- Where it reads “…significant cardiovascular toxicities, including hypertension, left ventricular dysfunction, and heart failure [65]” but the citing article [65] is about “…madecassoside administration significantly mitigated cardiac function decline in MI mice by promoting angiogenesis and inhibiting myocardial cell apoptosis and fibrosis.” Review the adequacy of this citation,
- Where it reads “Clinical data indicate that up to 47% of patients treated with sunitinib develop hypertension, approximately 28% experience left ventricular dysfunction, and about 8% progress to heart failure [68].”; However, article [68] does not contain any reference to sunitinib. Review the adequacy of this citation and cite the authors who cite the content presented.
- Where it reads "Importantly, many of these adverse effects are reversible upon discontinuation of therapy [69]." However, once again, this content is not found in citation [69] that addresses "Sunitinib and Imatinib Display Differential Cardiotoxicity in Adult Rat Cardiac Fibroblasts That Involves a Role for Calcium/Calmodulin-Dependent Protein Kinase II."
- Where it reads “Sorafenib exhibits a similar, although somewhat milder, cardiotoxicity profile [64].”; but the cited article only addresses liver toxicity (“8,619 cases of liver injury were identified. Pazopanib had the highest association with liver injury (reporting odds ratio 3.9). The median onset of liver injury was 21 days. Mortality was 28.5%, with Sorafenib linked to the highest mortality (48.6%). Lenvatinib had the highest hospitalization rate (56%).”
- The introduction of the following text in the TKi chapter “Androgen Receptor Signaling Inhibitors (ARSi) have profoundly reshaped the therapeutic landscape of advanced prostate cancer, … Recent meta-analyses and large pharmacovigilance studies have consistently reported an increased risk of hypertension, arrhythmias, myocardial ischemia, and even heart failure in patients receiving ARSi therapy [70]…toxic profiles, thereby informing multidisciplinary strategies that balance” oncologic efficacy with cardiovascular
- The introduction to the following text in chapter 2.2.2. TKIs is also unclear: "Cyclin-dependent kinase 4 and 6 inhibitors (CDK4/6i), namely palbociclib, ribociclib, and abemaciclib, have become a cornerstone in the treatment of hormone receptor-positive, HER2-negative advanced breast cancer." Cyclin-dependent kinase 4 and 6 inhibitors are not TKi; this text should therefore be removed from this chapter and/or reinserted in another.
- A subchapter is missing here: BCL-ABL tyrosine kinase inhibitors (e.g., imatinib, ponatinib) safety [72]”. This text must be removed or reinserted in a new chapter.
Page 9
2.3. Immune-related cardiovascular adverse events (irAEs)
- The acronyms are missing, meaning: “Cytotoxic T-lymphocyte-associated protein 4 (CTLA4) and Programmed death 1 (PD1) are the main lymphocyte immune checkpoints.” - Perhaps a brief (summarized) explanation is missing about the function of checkpoint factors “CTLA4 is located intracellular in resting T cells, and translocates to the surface upon activation. CTLA4 then antagonizes the costimulatory receptor cluster of differentiation 28 (CD28) by ligation of CD28 ligands withhighaffinitytoinhibit T cell activation. CTLA4 furthermore inhibits. TCR activity to reduce T cell susceptibility to antigen presentation [2,3].PD1 becomes expressed during early antigen-mediated activation of T cells. However, prolonged antigen expression in chronic infections or cancer causes sustained PD1 expression on T cells[1,3]. initiates further down stream signaling (Fig. 1b) thereby antagonizingTCR and CD28signaling [1,3]”
- Where it reads: “five PD1-blocking antibodies (nivolumab, pembrolizumab, cemiplimab, dostarlimab and toripalimab); three PD-L1-blocking antibodies (atezolizumab, avelumab, and durvalumab); and one LAG3 blocking antibody (relatlimab).”; quote?? (see reference of the current article …[77] “ICIs [anti-PD-1 antibodies (nivolumab, pembrolizumab, and cemiplimab), anti-PD-L1 antibodies (atezolizumab, avelumab, and durvalumab), and anti-CTLA-4 antibodies (ipilimumab and tremelimumab)]; what are the differences in relation to the acronyms presented (PD1 and PD-L1??? - anti–programmed death-1/ligand-1 (PD-1/PD-L1)???) ((atezolizumab, avelumab, durvalumab).
- [77] it would be better to aim “ICI use was associated with an increased risk of 6 CV irAEs including myocarditis, pericardial diseases, heart failure, dyslipidemia, myocardial infarction, and cerebral arterial ischemia with higher risks for myocarditis (OR: 4.42, 95% CI: 1.56-12.50, P < 0.01; I2 = 0%, P = 0.93) and dyslipidemia (Peto OR: 3.68, 95% CI: 1.89-7.19, P < 0.01; I2 = 0%, P = 0.66). The incidence of these CVAEs ranged from 3.2 (95% CI 2.0-5.1) to 19.3 (6.7-54.1) per 1000 patients, in studies with a median follow-up ranging from 3.2 to 32.8 months.
- change: to the indications expressed in red “CV irAEs include: myocarditis, pericardial disease, Takotsubo-Like cardiomyopathy, Myocardial Infarction (MI), arrhythmias and conduction disorders.”; should be “CV irAEs include: myocarditis, pericardial disease, Takotsubo-like cardiomyopathy, myocardial Infarction (MI), arrhythmias and conduction disorders.
Page 10
- Where it reads: “PCSK9Inhibitors in Cancer Patients Treated with Immune-Checkpoint Inhibitors to Reduce Cardiovascular Events: New Frontiers in Cardioncology
- the quote [82] addresses “PCSK9 induces peripheral immune tolerance (inhibition of cancer cell- immune recognition), reduces cardiac mitochondrial metabolism, and enhances cancer cell survival”; the relationship with the rupture of the atheroma plaque and ACS should be better explained
2.4. Cardiotoxicity by new classes of drugs
- Where it says “Administration for use in a multitude of malignancies: two anti-HER-2 antibodies (Trastuzumab Emtansine and Trastuzumab Deruxtecan); two anti-TROP2 antibodies (Sacituzumab Govitecan and Datopotamab Deruxtecan); one anti-Nectin-4 antibody (Enfortumab Vedotin); one anti-Tissue Factor antibody (Tisotumab Vedotin); one anti-FRα (folate receptor alpha) antibody (Mirvetuximab Soravtansine) and one anti-c-Met antibody (Telisotuzumab Vedotin) (https://www.fda.gov/ accessed on July 2025).” In this context, also add anti-trophoblast antigen 2 (TROP2); see also Table 2 of Akram et al (2025) and add other ADCs and therapeutic indications and related antigens.
- Where it reads “…ranging from 1% to 4% for symptomatic heart failure, with a higher proportion experiencing asymptomatic LVEF decline) [87]; aim better [87] “CHF/LVEF drop grade 3/4 was reported in 0.71%, cardiac ischemia in 0.1%, cardiac arrhythmia in 0.71% and grade 1/2 LVEF drop in 2.04%”
Page 11
Diagnostic Strategies and Risk Stratification
3.1. Baseline Risk Stratification
It is better and more correct to quote “2021ESCGuidelinesoncardiovascular disease prevention in clinical practice” than those from the 2003 project.
Page 12
- Table 1 was copied entirely from Lyon et al. (2022), so the missing text is: Taken and adapted from Lyon et al. (2022) and with permission (which must be obtained).
- The legend of Table 1 (to understand H, M1…) is missing: “Risk level: Low risk = no risk factors OR one moderate risk factor; moderate risk (M) = moderate risk factors with a total of 2–4 points (Moderate1 [M1] = 1 point; Moderate [M2] = 2 points); high risk (H) = moderate risk factors with a total of ≥5 points OR any high-risk factor; very-high risk (VH) = any very-high risk factor. a AF, atrial flutter, ventricular tachycardia, or ventricular fibrillation. b Elevated above the ULN of the local laboratory reference range. c Systolic BP ≥140 mmHg or Diastolic BP 90mmHg, or on treatment. GFR 60mL/min/1.73m2. HbA1c 7.0% or 53mmol/mol, or on treatment. Non-high-density lipoprotein cholesterol 3.8mmol/L (0.145mg/dL) or on treatment. … (see and transcribe original caption).
Table 1 has some errors:
- 2nd column: Cardiac biomarkers: M2 and not M1
- 2nd column: Proteinuria, not M1; hyperlipidemia, not M1~
- Non-anthracycline?; should be M1
- Review the data in the 3rd column (HER2-targeted therapies) on Proteinuria, DM, Hyperlipidemia
…
- Review all data in Table 1 (compared to the original)
Page 13
3.3. Electrocardiogram (ECG)
- Atrial enlargement or atrial anomaly? (more correct)
Page 14
3.4. Echocardiography
Ok
Page 15
- Why are parts of the text in bold?
- Where it reads "An increase in hs-cTn above the 99th percentile upper reference limit (URL) or an NT-proBNP level >125 pg/mL may warrant further cardiac imaging or therapy modification [108]."; However, in the cited article [108], the authors only evaluated troponin – Review and cite the authors who reference NT-pro BNP > 125 pg/mL.
- “myeloperoxidase” – indicating oxidative stress - see [109]
3.6 Advanced Diagnostic Strategies
- Perhaps a short text on myocardial ischemia and antineoplastic agents is missing to serve as an introduction to the following texts (summarize and modify the content present in citation [9] of the current article and add the indicated citations). “Several cancer treatments are associated with an increased risk of stable angina and chronic coronary syndromes (CCS). 5-FU and capecitabine can precipitate effort angina in some cases. Platinum-containing chemotherapy-induced ischemia usually occurs after one of the first three cycles and in patients with underlying CAD. The incidence of cardiac ischemia is 1–5% with antimicrotubule agents, 2–3% with mall-moleculeVEGF-TKI, and 0.6–1.5% with VEGFi monoclonal antibody therapies. Nilotinib, ponatinib, and ICI also accelerate atherosclerosis, which can lead to stable angina.”
- Where it reads “It also holds value in long-term survivors exposed to high-dose mediastinal radiotherapy, for whom stress testing is recommended 5 years post-treatment and repeated every 5–10 years.”; Citation?; these indications are still maintained?? (see citation [9]) – stress testing has a sensitivity of only 50% (half the time it gives false positive or negative results) to identify obstructive coronary disease (see the most recent ESC guidelines on Acute and Chronic coronary syndromes). - Where it reads “…ranging from exercise ECG (low pre-test probability) and stress echocardiography to Coronary Angio CT (anatomic study; intermediate/low pre-test probability) ), pharmacologic stress cardiac magnetic resonance (CMR) or myocardial perfusion (Scintigraphyh) imaging (functional image studies; intermediate high pre-test - probability) —depends on pretest probability of coronary artery disease…” (see ESC guidelines of Acute and Chronic Coronary Syndromes)
Page 16
- Where it reads “…In cases requiring pharmacologic stress, agents such as dobutamine or vasodilators (adenosine for example) may be used effectively”; adenosine
Page 17
- Table 2 - correct content and the authors correctly refer to it as having been adapted from [9]; Adapted from Lyon AR et al.; remove AR – Lyon et al.
Table 5. Anthracycline equivalence dose (Lyon et al.) one of the strategies to reduce cardiotoxicity due to anthracyclines...
- 4.1 Pharmacological Cardioprotection
Page 18
- Where it reads "The main pharmacological agents employed in cardioprotection include renin–angiotensin system (RAS) inhibitors, beta-adrenergic blockers, dexrazoxane, and, in specific settings, mineralocorticoid receptor antagonists (MRAs), sodium–glucose co-transporter 2 inhibitors (SGLT2i), and statins." There is a lack of text on the potential use of ARNI (Sacubitril/Valsartan)… “Tajstra M. et al. 2023. MAINSTREAM is a randomized, placebo-controlled, double-blind, multicentre, clinical trial” – “Sacubitril/valsartan for cardioprotection in breast cancer (MAINSTREAM): design and rationale of the randomized trial” ;
“SGLT2i use reduced HF hospitalizations by 51% (RR 0.49, 95% CI 0.36–0.66, I² = 28%, P < 0.01) and new HF diagnoses by 71% (RR 0.29, 95% CI 0.10–0.87” [122],”
- Bhalraam, U.; Veerni, R. B.; Paddock, S.; Meng, J.; Piepoli, M.; López-Fernández, T.; Tsampasian, V.; Vassiliou, V. S. Impact of Sodium–Glucose Cotransporter-2 Inhibitors on Heart Failure Outcomes in Cancer Patients and Survivors: A Systematic Review and Meta-Analysis. European Journal of Preventive Cardiology 2025.”… European Journal of Preventive Cardiology (2025) 00, 1–13
- [123] “Recent preclinical and clinical data suggest that SGLT2is exerts cardioprotective effects through multiple mechanisms, including the modulation of inflammasome activity, specifically by reducing NLRP3 inflammasome activation Int. J. Mol. Sci. 2025, 26, 4780 2 of 22 and MyD88-dependent signaling, which are critical drivers of cardiac inflammation and fibrosis. Furthermore, SGLT2is has been shown to enhance mitochondrial viability in cardiac cells, promoting improved cellular energy metabolism and function, thus mitigating cardiotoxicity
Page 19~
- Where it reads “Dexrazoxane is the only agent specifically approved by the FDA and EMA for cardioprotection in patients receiving anthracyclines.”
It might be worth including a summary of this text: “Dexrazoxane appears to ameliorate the cardiotoxicity seen with anthracyclines by fusing with free and bound iron, thereby decreasing the formation of anthracycline-iron complexes and, eventually, the production of reactive oxygen species, which are harmful to the surrounding cardiac tissue…. Dexrazoxane; Chiamaka Eneh; Manidhar Reddy Lekkala; and [9] “In clinical practice, dexrazoxane infusion (dosage ratio dexrazoxane/doxorubicin 10/1; e.g., 500 mg/m² dexrazoxane per 50 mg/m² doxorubicin) should be considered (at least 30 min prior to each anthracycline cycle”).
- [125] – Dexrazoxane ???There is no reference to this cardioprotective drug in this citation.
Page 20
- Why the use of bold?
- Where it reads "...IV methylprednisolone"; it should read "IV methylprednisolone 1 g/day, IV, 3 days [132]"
Page 21
- Table 3 (original?), Taken from?, Permission?
- Guideline Recommendation column – line 1. Where it reads "% to < 50%," it should read "and to a value lower than 50%"
- Dexrazoxane line – instead of Class II b, it should be IIa (Guideline Recommendation column)
Table caption missing...ESC...LOE.
From Page 21 to Page 25 the content is correct from a scientific point of view. However, there are some inaccuracies/omissions that are summarized and important to correct:
- citations are missing to assess the complete accuracy of the text (alternatively, this subchapter should begin with a short text alluding to the fact that the proposed follow-up protocols are found in the 2022 ESC guidelines [9]).
In the current article: "4.2. Cardiac Surveillance Protocols
... For patients receiving anthracyclines, the frequency of cardiac assessments is dictated by the cumulative dose and individual risk profile (citation?). In low-risk patients receiving <250 mg/m² of doxorubicin or equivalent, echocardiography is typically performed at baseline and after treatment completion (citation?) [9]. In contrast, high-risk patients or those exceeding 250–300 mg/m² require more intensive monitoring, with echocardiographic evaluations repeated after every 100 mg/m² of cumulative dose or every 2–3 cycles of therapy. (quote?).
In all cases, an increase in hs-Tn above the upper reference limit or a >15% relative decline in GLS from baseline should prompt consideration of initiating cardioprotective therapy, even in the absence of symptoms (citation?). …HER-2 …In these patients, echocardiography should be performed every three months during treatment, with additional imaging if symptoms arise or biomarkers change (citation?) [9]??.. In low-risk, asymptomatic patients with stable cardiac function, this interval may be extended to every four months following the third-month evaluation (citation?).
A similar text on Fluorpyrimidines and VEGFi-related cardiovascular toxicities is missing
… The incidence of ICI-related myocarditis is relatively low (< 1%) but the associated mortality may exceed 40% [138]? In the study cited cardiovascular death n=6 (n a total of 35, therefore less than 40%; (review). Baseline ECG, troponin, and echocardiography are recommended, with repeat testing every 4–6 weeks for the first 12 weeks of therapy in high-risk patients or those receiving combination regimens (citation?) [9]?
. Any rise in troponin, new ECG abnormalities, or unexplained symptoms should prompt immediate cardiac workup, including cardiac MRI or endomyocardial biopsy when warranted (citation?). For VEGF inhibitors and TKIs, the primary concern is the development of hypertension and arterial thrombotic events. Surveillance focuses on strict blood pressure monitoring and periodic cardiac imaging in high-risk patients. Figure 1??? (you don't understand the introduction here)
…Risk factor assessment, ECG, echocardiogram, lipid profile, HbA1c, and ankle-brachial index are part of the baseline workup, with repeat vascular evaluation at 6–12 months for higher-risk agents such as nilotinib and ponatinib (nilotinib and ponatinib, multitargeted kinase inhibitors targeting BCR-ABL are generally associated with vascular events) [9]. QT prolongation is another concern with several TKIs (endoplasmic reticulum stress is activated by multiple TKIs and leads to cardiotoxicity through promoting expression of pro-inflammatory factors and fetal cardiac genes), and serial ECG monitoring is indicated during the initial weeks of therapy and dose adjustments. Left atrial anomaly (remove enlargement) on baseline ECG before ibrutinib has been shown to be a predictor for the development of atrial fibrillation (AF) during chemotherapy [58,59]?? In none of these 2 citations is there a reference to ibrutinib – review; see Table 12 of [9]). The presence of atrioventricular (AV) conduction delays and premature atrial complexes are associated with the development of atrial arrhythmias in patients undergoing autologous haematopoietic stem cell transplantation (HSCT) [9]?.
…Fluoropyrimidines, known for their coronary toxicity, require rigorous baseline cardiovascular risk stratification. While there are no standardized surveillance intervals for these agents, close clinical follow-up and management of modifiable risk factors are critical throughout the treatment course [139]. (Figure 2)?? Just one example…
…Troponin and NT-proBNP levels, along with ECHO (echocardiography), should be assessed at baseline, especially in patients with pre-existing cardiac disease. These patients require close cardiac monitoring during and after CRS??, and treatment should be paused in the event of significant clinical deterioration. Therapies such as EGFR?? and ALK inhibitors?? may cause QTc prolongation and conduction abnormalities, necessitating regular ECG and lipid monitoring, as well as echocardiography in specific cases such as osimertinib use. CDK4/6 inhibitors have also been implicated in QTc prolongation, especially ribociclib, which warrants ECG monitoring at two and four weeks after treatment initiation and with each dose escalation (citation?).
- Hormonal therapies—including androgen deprivation therapy and anti-estrogen agents (this item is not included in the first part of the text, unless the authors create a new subchapter not including these drugs in the TKI group)—are linked to hypertension, ischemia, and LVSD (citation?).
Therefore, periodic cardiovascular risk stratification using SCORE2/SCORE2-OP is recommended, along with serial ECGs for QTc assessment in at-risk patients (citation?). In experimental models, it has been observed that both tamoxifen and toremifen (not previously addressed for their potential cardiotoxicity), when administered at high doses, can induce a prolongation of the QTc interval by direct inhibition of the fast-acting delayed potassium rectifier (IKr) (increasing the duration of the action potential and potentially promoting torsade pointes ventricular polymorphic tachycardia). The clinical significance of this adverse effect is limited, however, as in patients treated with the standard dose of 20 mg/day tamoxifen this effect has not been detected [140]. Immunomodulatory drugs such as thalidomide and lenalidomide do not have class-specific guideline recommendations but warrant attention to thrombotic risk and symptomatic bradycardia, with ECG monitoring as indicated (citation?).
Proteasome inhibitors (the backbone of combination treatments for patients with multiple myeloma and AL amyloidosis, while also indicated in Waldenström's macroglobulinemia and other malignancies) [141], particularly carfilzomib, are associated with cardiovascular toxicities such as heart failure, hypertension, and ischemic events, likely due to endothelial dysfunction and oxidative stress (also not previously addressed; citation?).. Baseline cardiac evaluation with ECG and echocardiography is recommended, especially in patients with pre-existing cardiovascular disease. Monitoring during therapy should include troponin and natriuretic peptides in high-risk patients. Early signs of dysfunction (e.g., GLS decline, biomarker elevation) may warrant treatment modification or cardioprotective strategies (citation?).
Bortezomib and ixazomib show a lower cardiotoxic profile but still require attention to blood pressure and volume status [141]. Thoracic radiation therapy (RT) is associated with delayed cardio vascular effects, notably accelerated atherosclerosis (previously addressed but RT removed from context; citation?)... Surveillance strategies include long-term follow-up with emphasis on cardiovascular risk factor control and screening beginning five to ten years post-treatment, especially in high-risk individuals. In long-term survivors, particularly those treated in childhood or young adulthood, late-onset cardiotoxicity can emerge years to decades after therapy (citation?). Surveillance guidelines recommend periodic reassessment of cardiac function every 5 years, or more frequently in high-risk individuals, using echocardiography and biomarkers (citation?). Survivors exposed to chest irradiation or high cumulative anthracycline doses require lifelong follow-up due to the risk of valvular disease, coronary artery disease, and restrictive cardiomyopathy [142].
- Contains content not covered in the first part (see Table 1) (anti-androgens...RT...anti-hyeloma...)
- Figure 1 in the text: caption: "Figure 1. Cardiotoxicity associated with anti-VEGF therapy and taxane-based treatment in a 66 year-old woman with hypertension and a history of recurrent ER−/HER2− breast cancer. (A) Base line strain analysis before starting paclitaxel (taxol; remove) plus bevacizumab, showing reduced GLS (−14%) with preserved LVEF (55%). (B) ECG during an episode of chest pain and dyspnea (in February 2022; remove), with elevated troponin and NT-proBNP, showing negative T waves in lateral leads (ST elevation V1-V3 and possible interpretation remain to be addressed) (C) Strain imaging at the time of symptoms showed further GLS impairment (−10%) and reduced LVEF (33%) (not explicit in the image). −16%.
ECG: electrocardiogram; ER: estrogen receptor; GLS: global longitudinal strain; HER2: human epidermal growth factor receptor 2; LVEF: left ventricular ejection fraction; NT-proBNP: N-terminal pro-B-type natriuretic peptide; VEGF: vascular endothelial growth factor.
- Figure 2: “Figure 2. Acute coronary syndrome related to fluoropyrimidine and anti-VEGF therapy in a 75 year-old male with colon cancer, previously treated with colectomy followed by FOLFIRI plus bevacizumab, and subsequently capecitabine plus bevacizumab. (A) ECG on admission shows inferior STEMI with complete AV block and junctional escape rhythm (with Left Bunble Branch Block pattern which makes it difficult to value ST elevation in inferior leads – inferior wall STEMI).
…AV: atrioventricular; DES: drug-eluting stent; ECG: electrocardiogram; FOLFIRI: folinic acid/5-fluorouracil/irinotecan; LVEF: left ventricular ejection fraction; PCI: percutaneous coronary intervention; RCA: right coronary artery; STEMI: ST-elevation myocardial infarction; VEGF: vascular endothelial growth factor.
The chapters "The Role of an Integrated Cardio-Oncology Approach and Conclusions" are well-written, with accurate content and realistic and very interesting proposals for cardio-oncology, and there are no suggestions to make.
References
The citations are correct and perfectly suited to the proposed topic, but contain a number of inaccuracies that must be reviewed and changed:
- Citations in the body of the text in bold...for example, [16]...should be [16] and the same text on page 20 (remove the bold);
- All citations must follow publication guidelines: no ( ) but only the number; 10 authors et al; journal name in diminutive form and in italics, and year in bold; only the authors' names without reference to Departments, Universities, or Institutions. Examples to change (all citations):
- (1) Gao, F.; Xu, T.; Zang, F.; Luo, Y.; Pan, D. Cardiotoxicity of Anticancer Drugs: Molecular Mechanisms, Clinical Management and Innovative Treatment. DDDT 2024, Volume 18, 4089–4116. https://doi.org/10.2147/dddt.s469331.
It should be: 1. Gao, F.; Xu, T.; Zang, F.; Luo, Y.; Pan, D. Cardiotoxicity of Anticancer Drugs: Molecular Mechanisms, Clinical Management and Innovative Treatment. Drug Des Devel Ther 2024, 18: 4089–4116. https://doi.org/10.2147/dddt.s469331.
DDDT = Drug Des Devel Ther ??
- 10 authors et al (multiple citations with many more than 10 authors);
?(9). Lyon, A. R.; López-Fernández, T.; Couch, L. S.; Asteggiano, R.; Aznar, M. C.; Bergler-Klein, J.; Boriani, G.; Cardinale, D.; Cordoba, R.; Cosyns, B.; Cutter, D. J.; De Azambuja, E.; De Boer, R. A.; Dent, S. F.; Farmakis, D.; Gevaert, S. A.; Gorog, D. A.; Herrmann, J.; Lenihan, D.; Moslehi, J.; Moura, B.; Salinger, S. S.; Stephens, R.; Suter, T. M.; Szmit, S.; Tamargo, J.; Thavendiranathan, P.; Tocchetti, C. G.; Van Der Meer, P.; Van Der Pal, H. J. H.; ESC Scientific Document Group; Lancellotti, P.; Thuny, F.; Abdelhamid, M.; Aboyans, V.; Aleman, B.; Alexandre, J.; Barac, A.; Borger, M. A.; Casado-Arroyo, R.; Cautela, J.; ÄŒelutkienÄ—, J.; Cikes, M.; Cohen-Solal, A.; Dhiman, K.; Ederhy, S.; Edvardsen, T.; Fauchier, L.; Fradley, M.; Grapsa, J.; Halvorsen, S.; Heuser, M.; Humbert, M.; Jaarsma, T.; Kahan, T.; Konradi, A.; Koskinas, K. C.; Kotecha, D.; Ky, B.; Landmesser, U.; Lewis, B. S.; Linhart, A.; Lip, G. Y. H.; Løchen, M.-L.; Malaczynska-Rajpold, K.; Metra, M.; Mindham, R.; Moonen, M.; Neilan, T. G.; Nielsen, J. C.; Petronio, A.-S.; Prescott, E.; Rakisheva, A.; Salem, J. E.; Savarese, G.; Sitges, M.; Berg, J. T.; Touyz, R. M.; Tycinska, A.; Wilhelm, M.; Zamorano, J. L.; Laredj, N.; Zelveian, P.; Rainer, P. P.; Samadov, F.; Andrushchuk, U.; Gerber, B. L.; Selimović, M.; Kinova, E.; Samardzic, J.; Economides, E.; Pudil, R.; Nielsen, K. M.; Kafafy, T. A.; Vettus, R.; Tuohinen, S.; Ederhy, S.; Pagava, Z.; Rassaf, T.; Briasoulis, A.; Czuriga, D.; Andersen, K. K.; Smyth, Y.; Iakobishvili, Z.; Parrini, I.; Rakisheva, A.; Pruthi, E. P.; Mirrakhimov, E.; Kalejs, O.; Skouri, H.; Benlamin, H.; ŽaliaduonytÄ—, D.; Iovino, A.; Moore, A. M.; Bursacovschi, D.; Benyass, A.; Manintveld, O.; Bosevski, M.; Gulati, G.; Leszek, P.; Fiuza, M.; Jurcut, R.; Vasyuk, Y.; Foscoli, M.; Simic, D.; Slanina, M.; Lipar, L.; Martin-Garcia, A.; Hübbert, L.; Kurmann, R.; Alayed, A.; Abid, L.; Zorkun, C.; Nesukay, E.; Manisty, C.; Srojidinova, N.; Baigent, C.; Abdelhamid, M.; Aboyans, V.; Antoniou, S.; Arbelo, E.; Asteggiano, R.; Baumbach, A.; Borger, M. A.; ÄŒelutkienÄ—, J.; Cikes, M.; Collet, J.-P.; Falk, V.; Fauchier, L.; Gale, C. P.; Halvorsen, S.; Iung, B.; Jaarsma, T.; Konradi, A.; Koskinas, K. C.; Kotecha, D.; Landmesser, U.; Lewis, B. S.; Linhart, A.; Løchen, M.-L.; Mindham, R.; Nielsen, J. C.; Petersen, S. E.; Prescott, E.; Rakisheva, A.; Sitges, M.; Touyz, R. M. 2022 ESC Guidelines on Cardio-Oncology Developed in Collaboration with the European Hematology Association (EHA), the European Society for Therapeutic Radiology and Oncology (ESTRO) and the International Cardio-Oncology Society (IC-OS). European Heart Journal 2022, 43 (41), 4229 4361. https://doi.org/10.1093/eurheartj/ehac244.
must be 10 authors et al. 2022 ESC Guidelines on Cardio-Oncology Developed in Collaboration with the European Hematology Association (EHA), the European Society for Therapeutic Radiology and Oncology (ESTRO) and the International Cardio-Oncology Society (IC-OS). European Heart Journal 2022, 43 (41), 4229 4361.
(14) Cardinale, D.; Iacopo, F.; Cipolla, C. M. Cardiotoxicity of Anthracyclines. Front. Cardiovasc. Med. 2020, 7. https://doi.org/10.3389/fcvm.2020.00026.
- Cardinale, D.; Iacopo, F.; Cipolla, C. M. Cardiotoxicity of Anthracyclines. Front. Cardiovasc. Med. 2020, 7. https://doi.org/10.3389/fcvm.2020.00026.
- McDonagh, T. A.; Metra, M.; Adamo, M.; Gardner, R. S.; Baumbach, A.; Böhm, M.; Burri, H.; Butler, J.; ÄŒelutkienÄ—, J.; Chioncel, O.; Cleland, J. G. F.; Coats, A. J. S.; Crespo-Leiro, M. G.; Farmakis, D.; Gilard, M.; Heymans, S.; Hoes, A. W.; Jaarsma, T.; Jankowska, E. A.; Lainscak, M.; Lam, C. S. P.; Lyon, A. R.; McMurray, J. J. V.; Mebazaa, A.; Mindham, R.; Muneretto, C.; Francesco Piepoli, M.; Price, S.; Rosano, G. M. C.; Ruschitzka, F.; Kathrine Skibelund, A.; ESC Scientific Document Group. 2021 ESC Guidelines for the Diagnosis and Treatment of Acute and Chronic Heart Failure. Eur Heart J 2021, 42 (36), 3599–3726; 10 authors et al…
https://doi.org/10.1093/eurheartj/ehac244.
48- ??Al-hussaniy, H. A.; Department of Pharmacy, Bilad Alrafidain University College, Diyala, Iraq; Dr. Hany Akeel Institute, Iraqi Medical Research Center, Baghdad, Iraq; Alburghaif, A. H.; Department of Pharmacy, Ashur University College, Baghdad, Iraq; Alkhafaje, Z.; Department of Pharmacy, Alfarahidi University College, Baghdad, Iraq; AL Zobaidy, M. A.-H. J.; Dr. Hany Akeel Institute, Iraqi Medical Research Center, Baghdad, Iraq; Alkuraishy, H. M.; Department of Clinical Pharmacology, College of Medicine, Almustansria University, Baghdad, Iraq; Mostafa-Hedeab, G.; Pharmacology Department & Health Research Unit, Medical College, Jouf University, Jouf, Saudi Arabia; Pharmacology Department, Faculty of Medicine, Beni-Suef University, Beni-Suef, Egypt; Azam, F.; Department of Pharmaceutical Chemistry and Pharmacognosy, Unaizah College of Pharmacy, Qassim University, Uniazah, Saudi Arabia; Al-Samydai, A. M.; Pharmacological and Diagnostic Research Centre, Faculty of Pharmacy, Al-Ahliyya Amman University, Amman, Jordan; Al-tameemi, Z. S.; Department of Pharmacy, Bilad Alrafidain University College, Diyala, Iraq; Dr. Hany Akeel Institute, Iraqi Medical Research Center, Baghdad, Iraq; Naji, M. A.; Dr. Hany Akeel Institute, Iraqi Medical Research Center, Baghdad, Iraq. Chemotherapy-Induced Cardiotoxicity: A New Perspective on the Role of Digoxin, ATG7 Activators, Resveratrol, and Herbal Drugs. JMedLife 2023, 16 (4), 491–500. https://doi.org/10.25122/jml 2022-0322.”; apenas os nomes dos autores sem referência a Departamentos, Universidades….Instituições
??53. Avagimyan, A.; Pogosova, N.; Kakturskiy, L.; Sheibani, M.; Challa, A.; Kogan, E.; Fogacci, F.; Mikhaleva, L.; Vandysheva, R.; Yakubovskaya, M.; Faggiano, A.; Carugo, S.; Urazova, O.; Jahanbin, B.; Lesovaya, E.; Polana, S.; Pharmaceuticals 2025, 18, x FOR PEER REVIEW 31 of 40 Kirsanov, K.; Sattar, Y.; Trofimenko, A.; Demura, T.; Saghazadeh, A.; Koliakos, G.; Shafie, D.; Alizadehasl, A.; Cicero, A.; Costabel, J. P.; Biondi-Zoccai, G.; Ottaviani, G.; Sarrafzadegan, N. Doxorubicin-Related Cardiotoxicity: Review of Fundamental Pathways of Cardiovascular System Injury. Cardiovasc Pathol 2024, 73, 107683.; 10 autores et al ….
????94. Visseren, F. L. J.; Mach, F.; Smulders, Y. M.; Carballo, D.; Koskinas, K. C.; Bäck, M.; Benetos, A.; Biffi, A.; Boavida, J.-M.; Capodanno, D.; Cosyns, B.; Crawford, C.; Davos, C. H.; Desormais, I.; Di Angelantonio, E.; Franco, O. H.; Halvorsen, S.; Hobbs, F. D. R.; Hollander, M.; Jankowska, E. A.; Michal, M.; Sacco, S.; Sattar, N.; Tokgozoglu, L.; Tonstad, S.; Tsioufis, K. P.; Van Dis, I.; Van Gelder, I. C.; Wanner, C.; Williams, B.; ESC Scientific Document Group; De Backer, G.; Regitz-Zagrosek, V.; Aamodt, A. H.; Abdelhamid, M.; Aboyans, V.; Albus, C.; Asteggiano, R.; Bäck, M.; Borger, M. A.; Brotons, C.; ÄŒelutkienÄ—, J.; Cifkova, R.; Cikes, M.; Cosentino, F.; Dagres, N.; De Backer, T.; De Bacquer, D.; Delgado, V.; Den Ruijter, H.; Dendale, P.; Drexel, H.; Falk, V.; Fauchier, L.; Ference, B. A.; Ferrières, J.; Ferrini, M.; Fisher, M.; Fliser, D.; Fras, Z.; Gaita, D.; Giampaoli, S.; Gielen, S.; Graham, I.; Jennings, C.; Jorgensen, T.; Kautzky-Willer, A.; Kavousi, M.; Koenig, W.; Konradi, A.; Kotecha, D.; Landmesser, U.; Lettino, M.; Lewis, B. S.; Linhart, A.; Løchen, M.-L.; Makrilakis, K.; Mancia, G.; Marques-Vidal, P.; McEvoy, J. W.; McGreavy, P.; Merkely, B.; Neubeck, L.; Nielsen, J. C.; Perk, J.; Petersen, S. E.; Petronio, A. S.; Piepoli, M.; Pogosova, N. G.; Prescott, E. I. B.; Ray, K. K.; Reiner, Z.; Richter, D. J.; Rydén, L.; Shlyakhto, E.; Sitges, M.; Sousa-Uva, M.; Sudano, I.; Tiberi, M.; Touyz, R. M.; Ungar, A.; Verschuren, W. M. M.; Wiklund, O.; Wood, D.; Zamorano, J. L.; Smulders, Y. M.; Carballo, D.; Koskinas, K. C.; Bäck, M.; Benetos, A.; Biffi, A.; Boavida, J.-M.; Capodanno, D.; Cosyns, B.; Crawford, C. A.; Davos, C. H.; Desormais, I.; Di Angelantonio, E.; Franco Duran, O. H.; Halvorsen, S.; Richard Hobbs, F. D.; Hollander, M.; Jankowska, E. A.; Michal, M.; Sacco, S.; Sattar, N.; Tokgozoglu, L.; Tonstad, S.; Tsioufis, K. P.; Dis, I. V.; Van Gelder, I. C.; Wanner, C.; Williams, B. 2021 ESC Guidelines on Cardiovascular Disease Prevention in Clinical Practice. European Heart Journal 2021, 42 (34), 3227–3337. https://doi.org/10.1093/eurheartj/ehab484.; 10 authors et al ….
96-98??? 10 authors et al ….
Review all citations...including the Abbreviated Journal Name in italics and the year of publication in bold.
In conclusion: This article under analysis is of great scientific interest, generally well-written, and scientifically correct, but some aspects need to be corrected as duly indicated.
This article requires important changes to be made for publication: inconsistencies/omissions; erroneous or inappropriate citations; chapters with inconsistent text; the references chapter needs to be modified...the proposed changes are duly marked (in red).
Author Response
Abstract – no proposed change
Introduction
Page 2
- Where it reads: “Therapeutic Agents Associated with Cardiotoxicity”; remove.
Response: We would like to thank the reviewer for this valuable suggestion. The text was intended to indicate the title of Chapter 2 (1. Introduction, 2. Therapeutic agents associated with cardiotoxicity, etc.). However, the absence of the correct sequential numbering in the manuscript made this unclear. We have carefully revised and amended the text, and we trust that the revised version will now appear clearer and more readable.
Page 3
- Cited further ahead – where it reads “…which increases with cumulative dosage and can lead to progressive cardiomyopathy and heart failure [14]. It is worth noting… “The risk of anthracycline-induced heart failure increases as the cumulative dose administered increases: 3–5% with 400 mg/m2 and as high as 18–48% at 700 mg/m2…” as can be read in the citation [14] of this original article (or quote further ahead)
Response: We thank the reviewer for the suggestion. We have revised the text to include the quantitative risk estimates (3–5% at 400 mg/m² and 18–48% at 700 mg/m²), in order to provide greater precision.
- “the absolute number of affected individuals remains substantial [15].”; published numbers? – noting/objectify.
Response: We thank the reviewer for this comment. We have revised the text to provide a quantitative estimate, noting that severe cardiac complications may occur in up to 5% of high‑risk patients, thereby objectifying the statement regarding the absolute number of affected individuals.
- objectify with the however of the citation [1 "Hypertension (OR: 1.99; 95% CI: 1.43–2.76), diabetes mellitus (OR: 1.74; 95% CI: 1.11–2.74), and obesity (OR: 1.72; 95% CI: 1.13–2.61) were associated with an increased risk of cardiotoxicity.
Response: We thank the reviewer for this comment. We have revised the text to include the reported odds ratios and 95% confidence intervals for hypertension, diabetes, and obesity, thereby objectifying their association with anthracycline-induced cardiotoxicity
- According to the authors of the citation [18] "Currently, pharmacogenomic testing of all childhood cancer patients with an indication for doxorubicin or daunorubicin therapy for RARG rs2229774, SLC28A3 rs7853758, and UGT1A6*4 rs17863783 variants is recommended. There is no recommendation regarding testing in adults."
Response: We thank the reviewer for the suggestion. The text has been revised to specify that pharmacogenomic testing for the mentioned variants is recommended for pediatric patients, with no current recommendation for adults.
- “Notable variants include those in CBR1 and CBR3, encoding carbonyl reductases, ABCC1 and ABCC5, related to ATP-binding cassette transporters, and SLC22A, a solute carrier gene, which are predominantly implicated in pediatric populations and patients treated for acute lymphoblastic leukemia.” – Quote?
Response: We appreciate the reviewer’s suggestion. The text has been revised to include a direct quotation.
- The citation [19] highlights “study indicated 6 variants obviously associated with the increased risk for CIC, including CYBA rs4673 (pooled odds ratio, 1.93; 95% CI, 1.13-3.30), RAC2 rs13058338 (2.05; 1.11-3.78), CYP3A5 rs776746 (2.15; 1.00-4.62) ABCC1 rs45511401 (1.46; 1.05-2.01), ABCC2 rs8187710 (2.19; 1.38-3.48), and HER2-Ile655Val rs1136201 (2.48; 1.53-4.02).”, see: not the polymorphisms listed above.
Response: We thank the reviewer for pointing this out. We have corrected the citation in the text.
- This text lacks a caption for the acronyms used (CBR1 and CBR3, ABCC1 and ABCC5, etc.) and whenever these acronyms for genetic polymorphisms are used (at least once…).
Response: We thank the reviewer for this comment. We have revised the text to include explanations for all genetic acronyms (CBR1, CBR3, ABCC1, ABCC5, etc.) at their first occurrence.
- I believe the authors of the citation [19] make no reference to NOS3 (so this should be changed)...and the citations should be renumbered....
Response: We thank the reviewer for this comment. We have replaced citation [19] with a more appropriate reference supporting the information on NOS3, and have renumbered the citations accordingly.
Page 4
- How to reconcile these different data in the text? Perhaps using a citation [28]. Therefore: "Cardiac damage primarily involves alterations in myocardial cell function and pathological cell death, encompassing mitochondrial dysfunction, topoisomerase inhibition, disruptions in iron ion metabolism, myofibril degradation, and oxidative stress." , this is a possible summary of what is currently known about the pathophysiology of myocardial injury caused by anthracyclines…
Response: We thank the reviewer for the suggestion. The text has been revised to better reconcile the different mechanisms of anthracycline-induced cardiac injury, integrating molecular pathways with cardiomyocyte death processes. The reference numbering has been updated accordingly.
- NLRP-3: NLR family pyrin domain-containing 3 (NLRP-3)…”
Response: We thank the reviewer for the suggestion. The text has been modified accordingly to clarify the description of NLRP-3.
- “Given mitochondria constitute approximately 36–40%…” given (without bold;
Response: Thank you for the comment. The text has been revised.
- “ABCB8”??
Response: We thank the reviewer for the comment. In accordance with the suggestion, the text has been revised to include the full name of the ABCB8 protein, in order to improve clarity for the reader.
- receptor-interacting protein 3 (RIP3)…” of the mitochondrial permeability transition pore and necroptotic cell death [28].”; demonstrated in mice; in fact, several of the experimental findings on the mechanisms involved in the cardiotoxicity caused by anthracyclines are obtained in animal experiments, and this should be included in the submitted text.
Response: We thank the reviewer for the suggestion. The text has been revised to include the full names of the acronyms (CaMKII, RIPK3, and mPTP) and to clarify that the described mechanisms have been demonstrated in experimental studies in mice. We believe these changes improve both clarity and accuracy.
- I don't know if it's possible to use related figures (constructive criticism) in the current article, similar to those used by Xie et al. 2024... sketches of figures 2 and .. or others from the same article... modifying... altering and adapting... obviously altered (since this is a very long article... why not provide a better schematic basis).
Response: Thank you for your suggestion. We have prepared a figure to better illustrate the molecular mechanisms of anthracycline-induced damage. In addition, we have included a new schematic figure explaining cardiotoxicity induced by emerging therapies. We hope these additions provide a clearer and more comprehensive visual basis for the concepts discussed in the article.
- "Parkin"?; Parkin (PARK2), a Parkinson's disease-associated gene, is a potential tumor suppressor whose expression is frequently diminished in tumors;
Response: We thank the reviewer for the suggestion. The text has been revisedas suggested.
Page 5
- but the citation [32] does not address "In addition to cardiomyocytes, anthracyclines adversely affect non-myocyte cardiac cells, including fibroblasts, endothelial cells, vascular smooth muscle cells, and immune cells [32]"; but rather "We conclude that extracellular vesicles from breast cancer cells mediate skeletal muscle mitochondrial dysfunction in cancer and may contribute to muscle weakness in some cancer patients." this aspect must be corrected,
Response: We thank the reviewer for pointing out this mistake. The citation has been corrected, and an appropriate reference has been inserted to support the statement regarding the effects of anthracyclines on non-myocyte cardiac cells.
- "Doxorubicin compromises endothelial barrier function, increasing vascular permeability and promoting myocardial interstitial edema [33]."; but in this article, "cell culture... mice...".
Response: We thank the reviewer for the observation. The text has been revised to clarify that the reported effects of doxorubicin on endothelial barrier function were demonstrated in experimental studies in cell cultures and mice.
- "These cellular and molecular alterations, combined with activation of innate immune pathways, drive maladaptive remodeling of the left ventricle [34].
FROM HERE ON, THE AUTHORS OF THE CURRENT ARTICLE INSERT A LONG TEXT WITHOUT ANY INTERLINK TO THE CARDIOTOXICITY RELATED TO THE USE OF ANTHRACYCLINES; Therefore, this text should be removed (due to the absence of any pathophysiological correlation with anthracyclines, or a logical and sequential interconnection should be made with the use of anthracyclines… and… radiotherapy… and microbiome alterations…
Therefore, all content should be revised or properly integrated… from the text that begins with “Radiotherapy remains an indispensable modality in the treatment of thoracic malignancies such as breast cancer and mediastinal lymphomas. Nevertheless… to “… efficacy of agents like doxorubicin while reducing their cardiotoxic footprint. This dual action highlights a promising avenue for microbiome-targeted strategies within cardio oncology [40].”; this content should be removed or integrated with cardiotoxicity related to anthracyclines… renumber the citations; alternatively, create new chapters (RT…
Response: We thank the reviewer for the suggestion. The section on radiotherapy and microbiome has been revised to establish a clear and logical connection with anthracycline-induced cardiotoxicity.References have been updated and renumbered accordingly.
Page 6:
2.1.2. Fluoropyrimidin
Some inconsistencies or errors:
- Fluoropyrimidin
- Where it reads: "cancers, including gastrointestinal and genitourinary tumors [42]. Even though fluoropyrimidines are commonly regarded as tolerable, they can provoke cardiovascular side effects in a notable proportion of patients, affecting between 1% and 18% [6]." These figures are according to Morelli et al. [6]; but only in this type of cancer? According to Deac et al. [43], "head and neck cancer, breast cancer, esophageal cancer, gastric cancer, pancreatic cancer, and colorectal cancer"; (see, for example, Shiga and Hiraide, 2020 – citation [42] of the current article; 1 and 18%?; The same authors, Jurcyzk [41] Shiga and Hiraide, 2020 – citation [42] in Table 1 of their article indicate an incidence that can reach up to 34.6%. (review this content). - since this is a very extensive article, it was worth briefly describing the clinical aspects potentially linked to cardiotoxicity as well as better describing the cardiotoxic effect and cellular and molecular mechanisms of cardiotoxicity, as can be seen in Table 1 of the cited article [6].
Response: We thank the reviewer for this comment. We have clarified that the reported incidence (1–18%) applies across several cancer types, not only gastrointestinal or genitourinary, and added that higher rates (up to 34.6%) have been reported. We also included a brief mention of the main clinical manifestations.
- Where it reads “Four key loss-of-function DPYD alleles—*2A, *13, c.2846A>T, and HapB”; can be read DPYD*2A (IVS14+1G>A, c.1905+1G>A), DPYD*9B (c.2846A>T), DPYD*13 (c.1679T>G), and HapB3 (c.1129-5923C>G) [43].
Response: We thank the reviewer for pointing this out. The nomenclature of the DPYD variants has been corrected accordingly.
-Where it reads: “…marked increases in fluoropyrimidine toxicities, with relative risks between 1.7 and over 4 for severe adverse events [44]”; should be added “ “Carriers of DPYD variants were found to be significantly correlated with treatment-related mortality (OR = 34.86, 95% CI 13.96–87.05; p < 0.05)”.
Response: We thank the reviewer for this important remark. We have added the requested information.
- 2.1.3. taxanes taxanes,
- …”including paclitaxel…”; docetaxel, paclitaxel and albumin-bound paclitaxel (nab-paclitaxel)
Response: We thank the reviewer for this suggestion. The text has been updated to improve accuracy.
- Where it reads: “…occurring in approximately 3% to 20% of patients [48]”; [6] and not [48].
Response: We thank the reviewer for pointing out this error. The reference has been corrected.
Page 7:
- Where it reads: “Consequently, prophylactic administration of corticosteroids and antihistamines is often recommended [50]”. Quote [50]or [6]
Response: We thank the reviewer for this comment. The reference has been corrected to accurately reflect the source supporting prophylactic administration of corticosteroids and antihistamines.
- Where it reads: “…opening of mitochondrial permeability transition pores, and collapse of mitochondrial membrane potential, all contributing to cardiomyocyte injury [51].”; is this content actually in the content of the article cited as [51]?
Response: We thank the reviewer for this observation. Since the cited article does not discuss these mechanisms, we have removed this part of the text to ensure accuracy.
2.1.4. Alkylating Drugs
- where it reads “…myocardial infarction, with reported prevalence ranging from 7% to 32% [54].”; once again, [54] or Morelli et al. [6] should be cited, which in turn cites Alexandre J, Moslehi JJ, Bersell KR, Funck-Brentano C, Roden DM, Salem J-E. Anticancer drug-induced cardiac rhythm disorders: current knowledge and basic underlying mechanisms. Pharmacol Ther. (2018) 189:89–103. doi: 10.1016/j.pharmthera.2018.04.009
Response: We thank the reviewer for this observation. We have updated the text to include both Morelli et al. and Alexandre et al., 2018, to accurately reflect the reported prevalence of myocardial infarction.
- Where it reads "...including the arachidonic acid cascade and NF-κB signaling, leading to inflammatory cytokine release, cardiac tissue remodeling, and structural deterioration of the myocardium [56]. " This content does not exist in the article cited as [56] Sharifiaghdam, Z.; Amini, S. M.; Dalouchi, F.; Behrooz, A. B.; Azizi, Y. Apigenin-Coated Gold Nanoparticles as a Cardioprotective Strategy against Doxorubicin-Induced Cardiotoxicity in Male Rats via Reducing Apoptosis. Heliyon 2023, 9 (3), e14024. (Review this citation); the reference to NFkB is in fact included in Table 1 of the citation [6] “Expression of proinflammatory chemokines and cytokines driven by increased NFkB activation”-
Response: We thank the reviewer for pointing this out. The reference has been corrected to Morelli et al., which accurately describes NF-κB activation and its role in proinflammatory chemokine and cytokine expression contributing to cardiotoxicity.
- Where it reads "...practice has significantly reduced the frequency of such severe outcomes [57]."; but in the cited article [57] there is no reference to heart failure or myocardial infarction. "Clinical benefits of PTCy dose reduction may include faster engraftment and T-cell recovery and less severe BK virus cystitis/urethritis and mucositis."
Response: We thank the reviewer for this comment. We have removed this part of the text to ensure accuracy.
2.1. Cardiotoxicity induced by targeted therapy; 2.2. and not 2.1.1-
- An introductory explanation is missing "see, for example, Morelli et al. [6] "After the binding of soluble ligands, ERBB kinase receptors arrange in homo- or heterodimer complexes, which activate the tyrosine kinase activity and the consequent signaling events leading to the modulation of cell survival, proliferation, migration, and differentiation [reviewed in (104–107)]. ERBB2 (also known as HER2) receptor is a proto-oncogene frequently amplified and overexpressed in many human cancers.
Response: We thank the reviewer for this suggestion. We have added an introductory explanation to the section.
2.2.1. Anti-HER2
- Where it reads “…neuregulin-1 (NRG1)/ERBB2 signaling pathway essential for cardiac function [58].”; but in what ways is it essential? It would be worthwhile to develop NRG1 signaling a bit, as for example, Moreli et al. do, citing several authors to be cited if this proposal is accepted. “NRG1, together with its tyrosine kinase receptors ERBB4, ERBB3, and ERBB2, is essential for heart development (113–115) [reviewed in (110, 116, 117)] and tunes heart regenerative, inflammatory, fibrotic, and metabolic processes (118, 119) [reviewed in (110, 117, 120–122)]. In cardiomyocytes, the most prominently expressed NRG1 receptors are ERBB4 and ERBB2 (123), and NRG1 stimulates fetal/neonatal cardiomyocyte proliferation, hypertrophy, sarcomerogenesis, and survival (114, 115, 124, 127) [reviewed in (110, 116, 117, 120, 121)].”
Response: We thank the reviewer for this suggestion. We have expanded the text to describe NRG1 signaling in the heart and its role in cardiomyocyte function, and we have highlighted the link to ERBB2-targeted therapy–induced cardiotoxicity.
Page 8
2.2.2. TKIs
- It may be worth taking advantage of the content of the citation [64] Nine VEGFR-TKIs (Axitinib, Vandetanib, Cabozantinib, Lenvatinib, Pazopanib, Ponatinib, Regorafenib, Sunitinib, Sorafenib)…
Response: We thank the reviewer for this suggestion. The text has been updated to include all VEGFR-TKIs listed in citation, providing a complete overview of these agents in the context of angiogenic receptor inhibition.
- Where it reads “…significant cardiovascular toxicities, including hypertension, left ventricular dysfunction, and heart failure [65]” but the citing article [65] is about “…madecassoside administration significantly mitigated cardiac function decline in MI mice by promoting angiogenesis and inhibiting myocardial cell apoptosis and fibrosis.” Review the adequacy of this citation.
Response: We thank the reviewer for this observation. The reference has been corrected to cite an appropriate source that accurately reports cardiovascular toxicities, including hypertension, left ventricular dysfunction, and heart failure, associated with VEGFR-TKIs. The previous citation has been removed.
- Where it reads “Clinical data indicate that up to 47% of patients treated with sunitinib develop hypertension, approximately 28% experience left ventricular dysfunction, and about 8% progress to heart failure [68].”; However, article [68] does not contain any reference to sunitinib. Review the adequacy of this citation and cite the authors who cite the content presented.
Response: We thank the reviewer for this observation. The reference has been corrected to cite an appropriate source.
- Where it reads "Importantly, many of these adverse effects are reversible upon discontinuation of therapy [69]." However, once again, this content is not found in citation [69] that addresses "Sunitinib and Imatinib Display Differential Cardiotoxicity in Adult Rat Cardiac Fibroblasts That Involves a Role for Calcium/Calmodulin-Dependent Protein Kinase II."
Response: We thank the reviewer for this observation. The statement regarding reversibility of adverse effects has been removed.
- Where it reads “Sorafenib exhibits a similar, although somewhat milder, cardiotoxicity profile [64].”; but the cited article only addresses liver toxicity (“8,619 cases of liver injury were identified. Pazopanib had the highest association with liver injury (reporting odds ratio 3.9). The median onset of liver injury was 21 days. Mortality was 28.5%, with Sorafenib linked to the highest mortality (48.6%). Lenvatinib had the highest hospitalization rate (56%).”
Response: We thank the reviewer for this observation. The reference has been corrected to cite an appropriate source
- The introduction of the following text in the TKi chapter “Androgen Receptor Signaling Inhibitors (ARSi) have profoundly reshaped the therapeutic landscape of advanced prostate cancer, … Recent meta-analyses and large pharmacovigilance studies have consistently reported an increased risk of hypertension, arrhythmias, myocardial ischemia, and even heart failure in patients receiving ARSi therapy [70]…toxic profiles, thereby informing multidisciplinary strategies that balance” oncologic efficacy with cardiovascular
- The introduction to the following text in chapter 2.2.2. TKIs is also unclear: "Cyclin-dependent kinase 4 and 6 inhibitors (CDK4/6i), namely palbociclib, ribociclib, and abemaciclib, have become a cornerstone in the treatment of hormone receptor-positive, HER2-negative advanced breast cancer." Cyclin-dependent kinase 4 and 6 inhibitors are not TKi; this text should therefore be removed from this chapter and/or reinserted in another.
Response: We thank the reviewer for the valuable suggestion. In the revised manuscript, we have created two dedicated subchapters: one focused on Androgen Receptor Signaling Inhibitors (ARSi) and the other on cyclin-dependent kinase (CDK) inhibitors, in order to provide a clearer overview of their oncologic relevance and associated cardiovascular toxicities.
- A subchapter is missing here: BCL-ABL tyrosine kinase inhibitors (e.g., imatinib, ponatinib) safety [72]”. This text must be removed or reinserted in a new chapter.
Response: Thank you for your suggestion. We have now integrated this section into the paragraph specifically dedicated to TKIs, to ensure a more coherent and comprehensive discussion.
Page 9
2.3. Immune-related cardiovascular adverse events (irAEs)
- The acronyms are missing, meaning: “Cytotoxic T-lymphocyte-associated protein 4 (CTLA4) and Programmed death 1 (PD1) are the main lymphocyte immune checkpoints.”
Response: We have revised the text by adding the full terminology of the acronyms (CTLA-4, PD-1, PD-L1, LAG-3) at first mention, as suggested.
- Perhaps a brief (summarized) explanation is missing about the function of checkpoint factors “CTLA4 is located intracellular in resting T cells, and translocates to the surface upon activation. CTLA4 then antagonizes the costimulatory receptor cluster of differentiation 28 (CD28) by ligation of CD28 ligands withhighaffinitytoinhibit T cell activation. CTLA4 furthermore inhibits. TCR activity to reduce T cell susceptibility to antigen presentation [2,3].PD1 becomes expressed during early antigen-mediated activation of T cells. However, prolonged antigen expression in chronic infections or cancer causes sustained PD1 expression on T cells[1,3]. initiates further down stream signaling (Fig. 1b) thereby antagonizingTCR and CD28signaling [1,3]”
Response: Thank you for your suggestion. We have now included a brief explanation of checkpoint functions in the revised manuscript.
- Where it reads: “five PD1-blocking antibodies (nivolumab, pembrolizumab, cemiplimab, dostarlimab and toripalimab); three PD-L1-blocking antibodies (atezolizumab, avelumab, and durvalumab); and one LAG3 blocking antibody (relatlimab).”; quote?? (see reference of the current article …[77] “ICIs [anti-PD-1 antibodies (nivolumab, pembrolizumab, and cemiplimab), anti-PD-L1 antibodies (atezolizumab, avelumab, and durvalumab), and anti-CTLA-4 antibodies (ipilimumab and tremelimumab)]; what are the differences in relation to the acronyms presented (PD1 and PD-L1??? - anti–programmed death-1/ligand-1 (PD-1/PD-L1)???) ((atezolizumab, avelumab, durvalumab).
Response: Thank you for this observation. We have clarified the terminology and harmonized the acronyms throughout the manuscript.
- [77] it would be better to aim “ICI use was associated with an increased risk of 6 CV irAEs including myocarditis, pericardial diseases, heart failure, dyslipidemia, myocardial infarction, and cerebral arterial ischemia with higher risks for myocarditis (OR: 4.42, 95% CI: 1.56-12.50, P < 0.01; I2 = 0%, P = 0.93) and dyslipidemia (Peto OR: 3.68, 95% CI: 1.89-7.19, P < 0.01; I2 = 0%, P = 0.66). The incidence of these CVAEs ranged from 3.2 (95% CI 2.0-5.1) to 19.3 (6.7-54.1) per 1000 patients, in studies with a median follow-up ranging from 3.2 to 32.8 months.
Response: Thank you for your comment. We have updated the manuscript to include a more detailed summary of cardiovascular immune-related adverse events associated with ICI use, specifying the six main CV irAEs, their relative risks, and incidence rates, as suggested.
- change: to the indications expressed in red “CV irAEs include: myocarditis, pericardial disease, Takotsubo-Like cardiomyopathy, Myocardial Infarction (MI), arrhythmias and conduction disorders.”; should be “CV irAEs include: myocarditis, pericardial disease, Takotsubo-like cardiomyopathy, myocardial Infarction (MI), arrhythmias and conduction disorders.
Response: Thank you for your comment. We have corrected the text as suggested.
Page 10
- Where it reads: “PCSK9Inhibitors in Cancer Patients Treated with Immune-Checkpoint Inhibitors to Reduce Cardiovascular Events: New Frontiers in Cardioncology
- the quote [82] addresses “PCSK9 induces peripheral immune tolerance (inhibition of cancer cell- immune recognition), reduces cardiac mitochondrial metabolism, and enhances cancer cell survival”; the relationship with the rupture of the atheroma plaque and ACS should be better explained
Response: Thank you for your comment. We have revised the manuscript in accordance with the suggestion.
2.4. Cardiotoxicity by new classes of drugs
- Where it says “Administration for use in a multitude of malignancies: two anti-HER-2 antibodies (Trastuzumab Emtansine and Trastuzumab Deruxtecan); two anti-TROP2 antibodies (Sacituzumab Govitecan and Datopotamab Deruxtecan); one anti-Nectin-4 antibody (Enfortumab Vedotin); one anti-Tissue Factor antibody (Tisotumab Vedotin); one anti-FRα (folate receptor alpha) antibody (Mirvetuximab Soravtansine) and one anti-c-Met antibody (Telisotuzumab Vedotin) (https://www.fda.gov/ accessed on July 2025).” In this context, also add anti-trophoblast antigen 2 (TROP2); see also Table 2 of Akram et al (2025) and add other ADCs and therapeutic indications and related antigens.
Response: We thank the Reviewer for this valuable suggestion. The text has been revised accordingly, and now includes all FDA-approved ADCs up to July 2025, with their full names, acronyms, and target antigens.
- Where it reads “…ranging from 1% to 4% for symptomatic heart failure, with a higher proportion experiencing asymptomatic LVEF decline) [87]; aim better [87] “CHF/LVEF drop grade 3/4 was reported in 0.71%, cardiac ischemia in 0.1%, cardiac arrhythmia in 0.71% and grade 1/2 LVEF drop in 2.04%”
Response: We thank the Reviewer for this observation. In response, we have modified the text to replace the generic range with the precise incidence rates.
Page 11
Diagnostic Strategies and Risk Stratification
3.1. Baseline Risk Stratification
It is better and more correct to quote “2021ESCGuidelinesoncardiovascular disease prevention in clinical practice” than those from the 2003 project.
Response: We thank the reviewer for the helpful comment. Citation 94 already refers to the “2021 ESC Guidelines on cardiovascular disease prevention in clinical practice,” ensuring the manuscript reflects the most up-to-date recommendations. The concern may arise because the reference in the bibliography is not correctly formatted, making it difficult to read. We correct the formatting to make it clear and easily accessible.
Page 12
- Table 1 was copied entirely from Lyon et al. (2022), so the missing text is: Taken and adapted from Lyon et al. (2022) and with permission (which must be obtained).
- The legend of Table 1 (to understand H, M1…) is missing: “Risk level: Low risk = no risk factors OR one moderate risk factor; moderate risk (M) = moderate risk factors with a total of 2–4 points (Moderate1 [M1] = 1 point; Moderate [M2] = 2 points); high risk (H) = moderate risk factors with a total of ≥5 points OR any high-risk factor; very-high risk (VH) = any very-high risk factor. a AF, atrial flutter, ventricular tachycardia, or ventricular fibrillation. b Elevated above the ULN of the local laboratory reference range. c Systolic BP ≥140 mmHg or Diastolic BP 90mmHg, or on treatment. GFR 60mL/min/1.73m2. HbA1c 7.0% or 53mmol/mol, or on treatment. Non-high-density lipoprotein cholesterol 3.8mmol/L (0.145mg/dL) or on treatment. … (see and transcribe original caption)
Table 1 has some errors:
- 2nd column: Cardiac biomarkers: M2 and not M1
- 2nd column: Proteinuria, not M1; hyperlipidemia, not M1~
- Non-anthracycline?; should be M1
- Review the data in the 3rd column (HER2-targeted therapies) on Proteinuria, DM, Hyperlipidemia
…
- Review all data in Table 1 (compared to the original)
Response: We thank the reviewer for the helpful comments. We add the following sentence to Table 1: “Taken and adapted from Lyon et al. (2022) and with permission.” Permission has been requested and will be confirmed once obtained. We also correct the errors in Table 1, revise all data according to the original source, and add the risk levels in the legend to clarify H, M1, M2, VH, and all other descriptors.
Page 13
3.3. Electrocardiogram (ECG)
- Atrial enlargement or atrial anomaly? (more correct)
We revise the text to use “atrial enlargement or atrial anomaly,” which is more accurate.
Page 14
3.4. Echocardiography
Ok
Page 15
- Why are parts of the text in bold?
Response: The bold text was a formatting error and has been corrected.
- Where it reads "An increase in hs-cTn above the 99th percentile upper reference limit (URL) or an NT-proBNP level >125 pg/mL may warrant further cardiac imaging or therapy modification [108]."; However, in the cited article [108], the authors only evaluated troponin – Review and cite the authors who reference NT-pro BNP > 125 pg/mL.
Response: We thank the reviewer for the comment. We add a citation that specifically references NT-proBNP >125 pg/mL to support the statement, in addition to the original reference.
- “myeloperoxidase” – indicating oxidative stress - see [109]
Response: We correct the text to include “myeloperoxidase”.
3.6 Advanced Diagnostic Strategies
- Perhaps a short text on myocardial ischemia and antineoplastic agents is missing to serve as an introduction to the following texts (summarize and modify the content present in citation [9] of the current article and add the indicated citations). “Several cancer treatments are associated with an increased risk of stable angina and chronic coronary syndromes (CCS). 5-FU and capecitabine can precipitate effort angina in some cases. Platinum-containing chemotherapy-induced ischemia usually occurs after one of the first three cycles and in patients with underlying CAD. The incidence of cardiac ischemia is 1–5% with antimicrotubule agents, 2–3% with mall-moleculeVEGF-TKI, and 0.6–1.5% with VEGFi monoclonal antibody therapies. Nilotinib, ponatinib, and ICI also accelerate atherosclerosis, which can lead to stable angina.”
Response: We thank the reviewer for the suggestion. We add a short introductory paragraph as suggested.
- Where it reads “It also holds value in long-term survivors exposed to high-dose mediastinal radiotherapy, for whom stress testing is recommended 5 years post-treatment and repeated every 5–10 years.”; Citation?; these indications are still maintained?? (see citation [9]) – stress testing has a sensitivity of only 50% (half the time it gives false positive or negative results) to identify obstructive coronary disease (see the most recent ESC guidelines on Acute and Chronic coronary syndromes). - Where it reads “…ranging from exercise ECG (low pre-test probability) and stress echocardiography to Coronary Angio CT (anatomic study; intermediate/low pre-test probability) ), pharmacologic stress cardiac magnetic resonance (CMR) or myocardial perfusion (Scintigraphyh) imaging (functional image studies; intermediate high pre-test - probability) —depends on pretest probability of coronary artery disease…” (see ESC guidelines of Acute and Chronic Coronary Syndromes)
Response: We thank the reviewer for the comment. We improve the wording on radiotherapy, noting that stress testing in long-term survivors has a Class IIa, Level C recommendation according to the 2022 ESC Cardio-Oncology Guidelines, and we integrate the text with the 2024 ESC Guidelines on Acute and Chronic Coronary Syndromes.
Page 16
- Where it reads “…In cases requiring pharmacologic stress, agents such as dobutamine or vasodilators (adenosine for example) may be used effectively”; adenosine
Response: We correct the text
Page 17
- Table 2 - correct content and the authors correctly refer to it as having been adapted from [9]; Adapted from Lyon AR et al.; remove AR – Lyon et al.
Response: We correct the text
Table 5. Anthracycline equivalence dose (Lyon et al.) one of the strategies to reduce cardiotoxicity due to anthracyclines...
- 4.1 Pharmacological Cardioprotection
Page 18
- Where it reads "The main pharmacological agents employed in cardioprotection include renin–angiotensin system (RAS) inhibitors, beta-adrenergic blockers, dexrazoxane, and, in specific settings, mineralocorticoid receptor antagonists (MRAs), sodium–glucose co-transporter 2 inhibitors (SGLT2i), and statins." There is a lack of text on the potential use of ARNI (Sacubitril/Valsartan)… “Tajstra M. et al. 2023. MAINSTREAM is a randomized, placebo-controlled, double-blind, multicentre, clinical trial” – “Sacubitril/valsartan for cardioprotection in breast cancer (MAINSTREAM): design and rationale of the randomized trial” ;
Response: Thank you for the suggestion. We have integrated the text with the latest data on ARNI (sacubitril/valsartan) from the MAINSTREAM and PRADA II trials, highlighting their potential cardioprotective effects in patients receiving anthracycline-based breast cancer therapy, while noting that their use remains investigational and no specific guideline recommendations are currently available. The ARNI class has also been added to Table 3.
“SGLT2i use reduced HF hospitalizations by 51% (RR 0.49, 95% CI 0.36–0.66, I² = 28%, P < 0.01) and new HF diagnoses by 71% (RR 0.29, 95% CI 0.10–0.87” [122],”
- Bhalraam, U.; Veerni, R. B.; Paddock, S.; Meng, J.; Piepoli, M.; López-Fernández, T.; Tsampasian, V.; Vassiliou, V. S. Impact of Sodium–Glucose Cotransporter-2 Inhibitors on Heart Failure Outcomes in Cancer Patients and Survivors: A Systematic Review and Meta-Analysis. European Journal of Preventive Cardiology 2025.”… European Journal of Preventive Cardiology (2025) 00, 1–13
- [123] “Recent preclinical and clinical data suggest that SGLT2is exerts cardioprotective effects through multiple mechanisms, including the modulation of inflammasome activity, specifically by reducing NLRP3 inflammasome activation Int. J. Mol. Sci. 2025, 26, 4780 2 of 22 and MyD88-dependent signaling, which are critical drivers of cardiac inflammation and fibrosis. Furthermore, SGLT2is has been shown to enhance mitochondrial viability in cardiac cells, promoting improved cellular energy metabolism and function, thus mitigating cardiotoxicity
Response: Thank you for the suggestion. We have integrated the text as recommended, including the updated data on SGLT2i showing reductions in heart failure hospitalizations and new-onset heart failure, as well as the mechanistic insights on inflammasome modulation and mitochondrial protection.
Page 19~
- Where it reads “Dexrazoxane is the only agent specifically approved by the FDA and EMA for cardioprotection in patients receiving anthracyclines.”
It might be worth including a summary of this text: “Dexrazoxane appears to ameliorate the cardiotoxicity seen with anthracyclines by fusing with free and bound iron, thereby decreasing the formation of anthracycline-iron complexes and, eventually, the production of reactive oxygen species, which are harmful to the surrounding cardiac tissue…. Dexrazoxane; Chiamaka Eneh; Manidhar Reddy Lekkala; and [9] “In clinical practice, dexrazoxane infusion (dosage ratio dexrazoxane/doxorubicin 10/1; e.g., 500 mg/m² dexrazoxane per 50 mg/m² doxorubicin) should be considered (at least 30 min prior to each anthracycline cycle”).
- Response: Thank you for the suggestion. We have added the information on the mechanism of action of dexrazoxane and its clinical use, along with the appropriate citation.
- [125] – Dexrazoxane ???There is no reference to this cardioprotective drug in this citation.
Response: Thank you for pointing this out. We have removed the incorrect citation for dexrazoxane.
Page 20
- Why the use of bold?
Response: It was a formatting error and has now been corrected.
- Where it reads "...IV methylprednisolone"; it should read "IV methylprednisolone 1 g/day, IV, 3 days [132]"
Response: We have updated the text to include the dosage as suggested. Thank you for the suggestion.
Page 21
- Table 3 (original?), Taken from?, Permission?
- Guideline Recommendation column – line 1. Where it reads "% to < 50%," it should read "and to a value lower than 50%"
- Dexrazoxane line – instead of Class II b, it should be IIa (Guideline Recommendation column)
Table caption missing...ESC...LOE.
Response: Thank you for your helpful comments. Table 3 is original. We have made the suggested corrections and included all relevant abbreviations
From Page 21 to Page 25 the content is correct from a scientific point of view. However, there are some inaccuracies/omissions that are summarized and important to correct:
- citations are missing to assess the complete accuracy of the text (alternatively, this subchapter should begin with a short text alluding to the fact that the proposed follow-up protocols are found in the 2022 ESC guidelines [9]).
Response: We have added an introductory sentence to this paragraph, referencing the 2022 ESC Cardio-Oncology Guidelines.
In the current article: "4.2. Cardiac Surveillance Protocols
... For patients receiving anthracyclines, the frequency of cardiac assessments is dictated by the cumulative dose and individual risk profile (citation?). In low-risk patients receiving <250 mg/m² of doxorubicin or equivalent, echocardiography is typically performed at baseline and after treatment completion (citation?) [9]. In contrast, high-risk patients or those exceeding 250–300 mg/m² require more intensive monitoring, with echocardiographic evaluations repeated after every 100 mg/m² of cumulative dose or every 2–3 cycles of therapy. (quote?).
In all cases, an increase in hs-Tn above the upper reference limit or a >15% relative decline in GLS from baseline should prompt consideration of initiating cardioprotective therapy, even in the absence of symptoms (citation?). …HER-2 …In these patients, echocardiography should be performed every three months during treatment, with additional imaging if symptoms arise or biomarkers change (citation?) [9]??.. In low-risk, asymptomatic patients with stable cardiac function, this interval may be extended to every four months following the third-month evaluation (citation?).
Response: We have added the relevant citations to support the statements regarding cardiac assessment frequency for patients receiving anthracyclines and HER2-targeted therapies, as suggested.
A similar text on Fluorpyrimidines and VEGFi-related cardiovascular toxicities is missing
Response: We have added text addressing cardiovascular toxicities related to fluoropyrimidines and VEGF inhibitors.
… The incidence of ICI-related myocarditis is relatively low (< 1%) but the associated mortality may exceed 40% [138]? In the study cited cardiovascular death n=6 (n a total of 35, therefore less than 40%; (review). Baseline ECG, troponin, and echocardiography are recommended, with repeat testing every 4–6 weeks for the first 12 weeks of therapy in high-risk patients or those receiving combination regimens (citation?) [9]?
Response: Thank you for your valuable suggestion. We have updated the citation on the epidemiology of ICI-related myocarditis to ensure it accurately reflects the reported data. Additionally, we have added the subsequent citation supporting baseline ECG, troponin, and echocardiography monitoring.
. Any rise in troponin, new ECG abnormalities, or unexplained symptoms should prompt immediate cardiac workup, including cardiac MRI or endomyocardial biopsy when warranted (citation?). For VEGF inhibitors and TKIs, the primary concern is the development of hypertension and arterial thrombotic events. Surveillance focuses on strict blood pressure monitoring and periodic cardiac imaging in high-risk patients. Figure 1??? (you don't understand the introduction here)
Response: Thank you for the comment. We have added the appropriate citation to support the recommendations. Additionally, we included a sentence clarifying that Figure 1 provides an example of VEGF inhibitor-related cardiotoxicity.
…Risk factor assessment, ECG, echocardiogram, lipid profile, HbA1c, and ankle-brachial index are part of the baseline workup, with repeat vascular evaluation at 6–12 months for higher-risk agents such as nilotinib and ponatinib (nilotinib and ponatinib, multitargeted kinase inhibitors targeting BCR-ABL are generally associated with vascular events) [9]. QT prolongation is another concern with several TKIs (endoplasmic reticulum stress is activated by multiple TKIs and leads to cardiotoxicity through promoting expression of pro-inflammatory factors and fetal cardiac genes), and serial ECG monitoring is indicated during the initial weeks of therapy and dose adjustments. Left atrial anomaly (remove enlargement) on baseline ECG before ibrutinib has been shown to be a predictor for the development of atrial fibrillation (AF) during chemotherapy [58,59]?? In none of these 2 citations is there a reference to ibrutinib – review; see Table 12 of [9]). The presence of atrioventricular (AV) conduction delays and premature atrial complexes are associated with the development of atrial arrhythmias in patients undergoing autologous haematopoietic stem cell transplantation (HSCT) [9]?.
Response: We have added quotes as suggested.
…Fluoropyrimidines, known for their coronary toxicity, require rigorous baseline cardiovascular risk stratification. While there are no standardized surveillance intervals for these agents, close clinical follow-up and management of modifiable risk factors are critical throughout the treatment course [139]. (Figure 2)?? Just one example…
Response: We included a sentence clarifying that Figure 2 provides a clinical example.
…Troponin and NT-proBNP levels, along with ECHO (echocardiography), should be assessed at baseline, especially in patients with pre-existing cardiac disease. These patients require close cardiac monitoring during and after CRS??, and treatment should be paused in the event of significant clinical deterioration. Therapies such as EGFR?? and ALK inhibitors?? may cause QTc prolongation and conduction abnormalities, necessitating regular ECG and lipid monitoring, as well as echocardiography in specific cases such as osimertinib use. CDK4/6 inhibitors have also been implicated in QTc prolongation, especially ribociclib, which warrants ECG monitoring at two and four weeks after treatment initiation and with each dose escalation (citation?).
- Hormonal therapies—including androgen deprivation therapy and anti-estrogen agents (this item is not included in the first part of the text, unless the authors create a new subchapter not including these drugs in the TKI group)—are linked to hypertension, ischemia, and LVSD (citation?).
Therefore, periodic cardiovascular risk stratification using SCORE2/SCORE2-OP is recommended, along with serial ECGs for QTc assessment in at-risk patients (citation?). In experimental models, it has been observed that both tamoxifen and toremifen (not previously addressed for their potential cardiotoxicity), when administered at high doses, can induce a prolongation of the QTc interval by direct inhibition of the fast-acting delayed potassium rectifier (IKr) (increasing the duration of the action potential and potentially promoting torsade pointes ventricular polymorphic tachycardia). The clinical significance of this adverse effect is limited, however, as in patients treated with the standard dose of 20 mg/day tamoxifen this effect has not been detected [140]. Immunomodulatory drugs such as thalidomide and lenalidomide do not have class-specific guideline recommendations but warrant attention to thrombotic risk and symptomatic bradycardia, with ECG monitoring as indicated (citation?).
Response: Thank you for your valuable comments. We have added the appropriate citations throughout this section to support the statements regarding cardiovascular monitoring for TKIs, CDK4/6 inhibitors and hormonal therapies. Additionally, we have clarified all abbreviations to improve comprehension. We have included a discussion of tamoxifen in the pharmacological section, and we have removed the part on thalidomide, as it was not previously addressed in the manuscript.
Proteasome inhibitors (the backbone of combination treatments for patients with multiple myeloma and AL amyloidosis, while also indicated in Waldenström's macroglobulinemia and other malignancies) [141], particularly carfilzomib, are associated with cardiovascular toxicities such as heart failure, hypertension, and ischemic events, likely due to endothelial dysfunction and oxidative stress (also not previously addressed; citation?).. Baseline cardiac evaluation with ECG and echocardiography is recommended, especially in patients with pre-existing cardiovascular disease. Monitoring during therapy should include troponin and natriuretic peptides in high-risk patients. Early signs of dysfunction (e.g., GLS decline, biomarker elevation) may warrant treatment modification or cardioprotective strategies (citation?).
Bortezomib and ixazomib show a lower cardiotoxic profile but still require attention to blood pressure and volume status [141]. Thoracic radiation therapy (RT) is associated with delayed cardio vascular effects, notably accelerated atherosclerosis (previously addressed but RT removed from context; citation?)... Surveillance strategies include long-term follow-up with emphasis on cardiovascular risk factor control and screening beginning five to ten years post-treatment, especially in high-risk individuals. In long-term survivors, particularly those treated in childhood or young adulthood, late-onset cardiotoxicity can emerge years to decades after therapy (citation?). Surveillance guidelines recommend periodic reassessment of cardiac function every 5 years, or more frequently in high-risk individuals, using echocardiography and biomarkers (citation?). Survivors exposed to chest irradiation or high cumulative anthracycline doses require lifelong follow-up due to the risk of valvular disease, coronary artery disease, and restrictive cardiomyopathy [142].
- Contains content not covered in the first part (see Table 1) (anti-androgens...RT...anti-hyeloma...)
Response: Thank you for your comments. We have removed the sections that were not previously contextualized in the manuscript, including thalidomide and anti-myeloma agents not previously discussed. We have also added the relevant references to support the remaining statements.
- Figure 1 in the text: caption: "Figure 1. Cardiotoxicity associated with anti-VEGF therapy and taxane-based treatment in a 66 year-old woman with hypertension and a history of recurrent ER−/HER2− breast cancer. (A) Base line strain analysis before starting paclitaxel (taxol; remove) plus bevacizumab, showing reduced GLS (−14%) with preserved LVEF (55%). (B) ECG during an episode of chest pain and dyspnea (in February 2022; remove), with elevated troponin and NT-proBNP, showing negative T waves in lateral leads (ST elevation V1-V3 and possible interpretation remain to be addressed) (C) Strain imaging at the time of symptoms showed further GLS impairment (−10%) and reduced LVEF (33%) (not explicit in the image). −16%.
ECG: electrocardiogram; ER: estrogen receptor; GLS: global longitudinal strain; HER2: human epidermal growth factor receptor 2; LVEF: left ventricular ejection fraction; NT-proBNP: N-terminal pro-B-type natriuretic peptide; VEGF: vascular endothelial growth factor.
- Figure 2: “Figure 2. Acute coronary syndrome related to fluoropyrimidine and anti-VEGF therapy in a 75 year-old male with colon cancer, previously treated with colectomy followed by FOLFIRI plus bevacizumab, and subsequently capecitabine plus bevacizumab. (A) ECG on admission shows inferior STEMI with complete AV block and junctional escape rhythm (with Left Bunble Branch Block pattern which makes it difficult to value ST elevation in inferior leads – inferior wall STEMI).
…AV: atrioventricular; DES: drug-eluting stent; ECG: electrocardiogram; FOLFIRI: folinic acid/5-fluorouracil/irinotecan; LVEF: left ventricular ejection fraction; PCI: percutaneous coronary intervention; RCA: right coronary artery; STEMI: ST-elevation myocardial infarction; VEGF: vascular endothelial growth factor.
Response: Thank you for your suggestions. We have revised the figure captions as indicated, clarifying the clinical details for both Figures 1 and 2.
The chapters "The Role of an Integrated Cardio-Oncology Approach and Conclusions" are well-written, with accurate content and realistic and very interesting proposals for cardio-oncology, and there are no suggestions to make.
References
The citations are correct and perfectly suited to the proposed topic, but contain a number of inaccuracies that must be reviewed and changed:
- Citations in the body of the text in bold...for example, [16]...should be [16] and the same text on page 20 (remove the bold);
- All citations must follow publication guidelines: no ( ) but only the number; 10 authors et al; journal name in diminutive form and in italics, and year in bold; only the authors' names without reference to Departments, Universities, or Institutions. Examples to change (all citations):
- (1) Gao, F.; Xu, T.; Zang, F.; Luo, Y.; Pan, D. Cardiotoxicity of Anticancer Drugs: Molecular Mechanisms, Clinical Management and Innovative Treatment. DDDT 2024, Volume 18, 4089–4116. https://doi.org/10.2147/dddt.s469331.
It should be: 1. Gao, F.; Xu, T.; Zang, F.; Luo, Y.; Pan, D. Cardiotoxicity of Anticancer Drugs: Molecular Mechanisms, Clinical Management and Innovative Treatment. Drug Des Devel Ther 2024, 18: 4089–4116. https://doi.org/10.2147/dddt.s469331.
DDDT = Drug Des Devel Ther ??
- 10 authors et al (multiple citations with many more than 10 authors);
?(9). Lyon, A. R.; López-Fernández, T.; Couch, L. S.; Asteggiano, R.; Aznar, M. C.; Bergler-Klein, J.; Boriani, G.; Cardinale, D.; Cordoba, R.; Cosyns, B.; Cutter, D. J.; De Azambuja, E.; De Boer, R. A.; Dent, S. F.; Farmakis, D.; Gevaert, S. A.; Gorog, D. A.; Herrmann, J.; Lenihan, D.; Moslehi, J.; Moura, B.; Salinger, S. S.; Stephens, R.; Suter, T. M.; Szmit, S.; Tamargo, J.; Thavendiranathan, P.; Tocchetti, C. G.; Van Der Meer, P.; Van Der Pal, H. J. H.; ESC Scientific Document Group; Lancellotti, P.; Thuny, F.; Abdelhamid, M.; Aboyans, V.; Aleman, B.; Alexandre, J.; Barac, A.; Borger, M. A.; Casado-Arroyo, R.; Cautela, J.; ÄŒelutkienÄ—, J.; Cikes, M.; Cohen-Solal, A.; Dhiman, K.; Ederhy, S.; Edvardsen, T.; Fauchier, L.; Fradley, M.; Grapsa, J.; Halvorsen, S.; Heuser, M.; Humbert, M.; Jaarsma, T.; Kahan, T.; Konradi, A.; Koskinas, K. C.; Kotecha, D.; Ky, B.; Landmesser, U.; Lewis, B. S.; Linhart, A.; Lip, G. Y. H.; Løchen, M.-L.; Malaczynska-Rajpold, K.; Metra, M.; Mindham, R.; Moonen, M.; Neilan, T. G.; Nielsen, J. C.; Petronio, A.-S.; Prescott, E.; Rakisheva, A.; Salem, J. E.; Savarese, G.; Sitges, M.; Berg, J. T.; Touyz, R. M.; Tycinska, A.; Wilhelm, M.; Zamorano, J. L.; Laredj, N.; Zelveian, P.; Rainer, P. P.; Samadov, F.; Andrushchuk, U.; Gerber, B. L.; Selimović, M.; Kinova, E.; Samardzic, J.; Economides, E.; Pudil, R.; Nielsen, K. M.; Kafafy, T. A.; Vettus, R.; Tuohinen, S.; Ederhy, S.; Pagava, Z.; Rassaf, T.; Briasoulis, A.; Czuriga, D.; Andersen, K. K.; Smyth, Y.; Iakobishvili, Z.; Parrini, I.; Rakisheva, A.; Pruthi, E. P.; Mirrakhimov, E.; Kalejs, O.; Skouri, H.; Benlamin, H.; ŽaliaduonytÄ—, D.; Iovino, A.; Moore, A. M.; Bursacovschi, D.; Benyass, A.; Manintveld, O.; Bosevski, M.; Gulati, G.; Leszek, P.; Fiuza, M.; Jurcut, R.; Vasyuk, Y.; Foscoli, M.; Simic, D.; Slanina, M.; Lipar, L.; Martin-Garcia, A.; Hübbert, L.; Kurmann, R.; Alayed, A.; Abid, L.; Zorkun, C.; Nesukay, E.; Manisty, C.; Srojidinova, N.; Baigent, C.; Abdelhamid, M.; Aboyans, V.; Antoniou, S.; Arbelo, E.; Asteggiano, R.; Baumbach, A.; Borger, M. A.; ÄŒelutkienÄ—, J.; Cikes, M.; Collet, J.-P.; Falk, V.; Fauchier, L.; Gale, C. P.; Halvorsen, S.; Iung, B.; Jaarsma, T.; Konradi, A.; Koskinas, K. C.; Kotecha, D.; Landmesser, U.; Lewis, B. S.; Linhart, A.; Løchen, M.-L.; Mindham, R.; Nielsen, J. C.; Petersen, S. E.; Prescott, E.; Rakisheva, A.; Sitges, M.; Touyz, R. M. 2022 ESC Guidelines on Cardio-Oncology Developed in Collaboration with the European Hematology Association (EHA), the European Society for Therapeutic Radiology and Oncology (ESTRO) and the International Cardio-Oncology Society (IC-OS). European Heart Journal 2022, 43 (41), 4229 4361. https://doi.org/10.1093/eurheartj/ehac244.
must be 10 authors et al. 2022 ESC Guidelines on Cardio-Oncology Developed in Collaboration with the European Hematology Association (EHA), the European Society for Therapeutic Radiology and Oncology (ESTRO) and the International Cardio-Oncology Society (IC-OS). European Heart Journal 2022, 43 (41), 4229 4361.
(14) Cardinale, D.; Iacopo, F.; Cipolla, C. M. Cardiotoxicity of Anthracyclines. Front. Cardiovasc. Med. 2020, 7. https://doi.org/10.3389/fcvm.2020.00026.
- Cardinale, D.; Iacopo, F.; Cipolla, C. M. Cardiotoxicity of Anthracyclines. Front. Cardiovasc. Med. 2020, 7. https://doi.org/10.3389/fcvm.2020.00026.
- McDonagh, T. A.; Metra, M.; Adamo, M.; Gardner, R. S.; Baumbach, A.; Böhm, M.; Burri, H.; Butler, J.; ÄŒelutkienÄ—, J.; Chioncel, O.; Cleland, J. G. F.; Coats, A. J. S.; Crespo-Leiro, M. G.; Farmakis, D.; Gilard, M.; Heymans, S.; Hoes, A. W.; Jaarsma, T.; Jankowska, E. A.; Lainscak, M.; Lam, C. S. P.; Lyon, A. R.; McMurray, J. J. V.; Mebazaa, A.; Mindham, R.; Muneretto, C.; Francesco Piepoli, M.; Price, S.; Rosano, G. M. C.; Ruschitzka, F.; Kathrine Skibelund, A.; ESC Scientific Document Group. 2021 ESC Guidelines for the Diagnosis and Treatment of Acute and Chronic Heart Failure. Eur Heart J 2021, 42 (36), 3599–3726; 10 authors et al…
https://doi.org/10.1093/eurheartj/ehac244.
48- ??Al-hussaniy, H. A.; Department of Pharmacy, Bilad Alrafidain University College, Diyala, Iraq; Dr. Hany Akeel Institute, Iraqi Medical Research Center, Baghdad, Iraq; Alburghaif, A. H.; Department of Pharmacy, Ashur University College, Baghdad, Iraq; Alkhafaje, Z.; Department of Pharmacy, Alfarahidi University College, Baghdad, Iraq; AL Zobaidy, M. A.-H. J.; Dr. Hany Akeel Institute, Iraqi Medical Research Center, Baghdad, Iraq; Alkuraishy, H. M.; Department of Clinical Pharmacology, College of Medicine, Almustansria University, Baghdad, Iraq; Mostafa-Hedeab, G.; Pharmacology Department & Health Research Unit, Medical College, Jouf University, Jouf, Saudi Arabia; Pharmacology Department, Faculty of Medicine, Beni-Suef University, Beni-Suef, Egypt; Azam, F.; Department of Pharmaceutical Chemistry and Pharmacognosy, Unaizah College of Pharmacy, Qassim University, Uniazah, Saudi Arabia; Al-Samydai, A. M.; Pharmacological and Diagnostic Research Centre, Faculty of Pharmacy, Al-Ahliyya Amman University, Amman, Jordan; Al-tameemi, Z. S.; Department of Pharmacy, Bilad Alrafidain University College, Diyala, Iraq; Dr. Hany Akeel Institute, Iraqi Medical Research Center, Baghdad, Iraq; Naji, M. A.; Dr. Hany Akeel Institute, Iraqi Medical Research Center, Baghdad, Iraq. Chemotherapy-Induced Cardiotoxicity: A New Perspective on the Role of Digoxin, ATG7 Activators, Resveratrol, and Herbal Drugs. JMedLife 2023, 16 (4), 491–500. https://doi.org/10.25122/jml 2022-0322.”; apenas os nomes dos autores sem referência a Departamentos, Universidades….Instituições
??53. Avagimyan, A.; Pogosova, N.; Kakturskiy, L.; Sheibani, M.; Challa, A.; Kogan, E.; Fogacci, F.; Mikhaleva, L.; Vandysheva, R.; Yakubovskaya, M.; Faggiano, A.; Carugo, S.; Urazova, O.; Jahanbin, B.; Lesovaya, E.; Polana, S.; Pharmaceuticals 2025, 18, x FOR PEER REVIEW 31 of 40 Kirsanov, K.; Sattar, Y.; Trofimenko, A.; Demura, T.; Saghazadeh, A.; Koliakos, G.; Shafie, D.; Alizadehasl, A.; Cicero, A.; Costabel, J. P.; Biondi-Zoccai, G.; Ottaviani, G.; Sarrafzadegan, N. Doxorubicin-Related Cardiotoxicity: Review of Fundamental Pathways of Cardiovascular System Injury. Cardiovasc Pathol 2024, 73, 107683.; 10 autores et al ….
????94. Visseren, F. L. J.; Mach, F.; Smulders, Y. M.; Carballo, D.; Koskinas, K. C.; Bäck, M.; Benetos, A.; Biffi, A.; Boavida, J.-M.; Capodanno, D.; Cosyns, B.; Crawford, C.; Davos, C. H.; Desormais, I.; Di Angelantonio, E.; Franco, O. H.; Halvorsen, S.; Hobbs, F. D. R.; Hollander, M.; Jankowska, E. A.; Michal, M.; Sacco, S.; Sattar, N.; Tokgozoglu, L.; Tonstad, S.; Tsioufis, K. P.; Van Dis, I.; Van Gelder, I. C.; Wanner, C.; Williams, B.; ESC Scientific Document Group; De Backer, G.; Regitz-Zagrosek, V.; Aamodt, A. H.; Abdelhamid, M.; Aboyans, V.; Albus, C.; Asteggiano, R.; Bäck, M.; Borger, M. A.; Brotons, C.; ÄŒelutkienÄ—, J.; Cifkova, R.; Cikes, M.; Cosentino, F.; Dagres, N.; De Backer, T.; De Bacquer, D.; Delgado, V.; Den Ruijter, H.; Dendale, P.; Drexel, H.; Falk, V.; Fauchier, L.; Ference, B. A.; Ferrières, J.; Ferrini, M.; Fisher, M.; Fliser, D.; Fras, Z.; Gaita, D.; Giampaoli, S.; Gielen, S.; Graham, I.; Jennings, C.; Jorgensen, T.; Kautzky-Willer, A.; Kavousi, M.; Koenig, W.; Konradi, A.; Kotecha, D.; Landmesser, U.; Lettino, M.; Lewis, B. S.; Linhart, A.; Løchen, M.-L.; Makrilakis, K.; Mancia, G.; Marques-Vidal, P.; McEvoy, J. W.; McGreavy, P.; Merkely, B.; Neubeck, L.; Nielsen, J. C.; Perk, J.; Petersen, S. E.; Petronio, A. S.; Piepoli, M.; Pogosova, N. G.; Prescott, E. I. B.; Ray, K. K.; Reiner, Z.; Richter, D. J.; Rydén, L.; Shlyakhto, E.; Sitges, M.; Sousa-Uva, M.; Sudano, I.; Tiberi, M.; Touyz, R. M.; Ungar, A.; Verschuren, W. M. M.; Wiklund, O.; Wood, D.; Zamorano, J. L.; Smulders, Y. M.; Carballo, D.; Koskinas, K. C.; Bäck, M.; Benetos, A.; Biffi, A.; Boavida, J.-M.; Capodanno, D.; Cosyns, B.; Crawford, C. A.; Davos, C. H.; Desormais, I.; Di Angelantonio, E.; Franco Duran, O. H.; Halvorsen, S.; Richard Hobbs, F. D.; Hollander, M.; Jankowska, E. A.; Michal, M.; Sacco, S.; Sattar, N.; Tokgozoglu, L.; Tonstad, S.; Tsioufis, K. P.; Dis, I. V.; Van Gelder, I. C.; Wanner, C.; Williams, B. 2021 ESC Guidelines on Cardiovascular Disease Prevention in Clinical Practice. European Heart Journal 2021, 42 (34), 3227–3337. https://doi.org/10.1093/eurheartj/ehab484.; 10 authors et al ….
96-98??? 10 authors et al ….
Review all citations...including the Abbreviated Journal Name in italics and the year of publication in bold.
Response: Thank you for the comment. We have thoroughly reviewed and modified the entire reference list to ensure full compliance with the journal’s guidelines, including the use of italicized abbreviated journal names and bold formatting for the year of publication.
Reviewer 4 Report
Comments and Suggestions for Authors
Major points:
1- In the sentence, "Anthracyclines exert cellular damage through increases in NLRP-3, interleukin-1, and interleukin-6 that reduce mitochondrial biogenesis", how can increased NLRP3 and interleukins reduce mitochondrial biogenesis?
2- The molecular mechanisms listed for anthracycline-induced cardiotoxicity need to be rearranged and presented as subtitles.
3- The molecular pathways implicated in each chemotherapy category (anthracyclines, fluoropirimidin, alkylating agents, taxanes) need to be explained in detail.
4- A table must be added summarizing the main findings of the cardiotoxicity induced by targeted therapies.
5- Importantly, the effect of the pharmacologic agents mentioned (i.e. RAS inhibitors, SGLT-2 inhibitors, and others) on the anticancer activity of the chemotherapeutic agents used must be discussed.
Minor points:
1- Some phrases need to be clarified (i.e. "Central to this process is mitochondrial dysfunction" - 2- Abbreviations, when first mentioned, need to be written in full.
3- A figure can be added to clarify the "cardiotoxicity-induced by new classes of drugs".
Comments on the Quality of English Language1- Some language mistakes have been detected and need to be improved.
Author Response
Major points:
1- In the sentence, "Anthracyclines exert cellular damage through increases in NLRP-3, interleukin-1, and interleukin-6 that reduce mitochondrial biogenesis", how can increased NLRP3 and interleukins reduce mitochondrial biogenesis?
Response 1: We thank the reviewer for the comment. We have modified the text to make the mechanism of anthracycline-induced mitochondrial damage clearer.
2- The molecular mechanisms listed for anthracycline-induced cardiotoxicity need to be rearranged and presented as subtitles.
Response 2: We thank the reviewer for the suggestion. We have revised the manuscript to present the molecular mechanisms of anthracycline-induced cardiotoxicity in a more structured and schematic manner. Specifically, we have organized the content under clear subtitles and provided an ordered list of the main mechanisms, while removing redundant concepts. We hope that this reorganization improves clarity and facilitates reader comprehension.
3- The molecular pathways implicated in each chemotherapy category (anthracyclines, fluoropirimidin, alkylating agents, taxanes) need to be explained in detail.
Response 3: We thank the reviewer for the comment. In response, we have revised the manuscript to include a detailed description of the molecular mechanisms of action for each chemotherapy category. Specifically, the molecular pathways underlying the cytotoxic effects of anthracyclines, fluoropyrimidines, alkylating agents, and taxanes are now explicitly described within each corresponding paragraph. We believe that these additions provide a clearer and more comprehensive understanding of the cellular effects of these agents.
4- A table must be added summarizing the main findings of the cardiotoxicity induced by targeted therapies.
Response 4: Thank you for the suggestion. We have added a table summarizing the main findings on cardiotoxicity induced by targeted therapies, including molecular targets, representative drugs, associated cardiovascular toxicities, and underlying mechanistic notes.
5- Importantly, the effect of the pharmacologic agents mentioned (i.e. RAS inhibitors, SGLT-2 inhibitors, and others) on the anticancer activity of the chemotherapeutic agents used must be discussed.
Response 5: Thank you for your valuable comment. We have added a section in the text discussing the effects of these pharmacologic agents (e.g., RAS inhibitors, SGLT2 inhibitors) on the anticancer activity of the chemotherapeutic treatments.
Minor points:
1.Some phrases need to be clarified (i.e. "Central to this process is mitochondrial dysfunction")
Response 1: We have revised the text to clarify ambiguous phrases, including the specified example, to ensure accurate and precise expression of the mechanisms described.
2- Abbreviations, when first mentioned, need to be written in full.
Response 2: All abbreviations are now written in full upon first mention throughout the manuscript, in accordance with journal guidelines.
3- A figure can be added to clarify the "cardiotoxicity-induced by new classes of drugs".
Response 3: We sincerely thank the Reviewer for this valuable suggestion. At present, the molecular mechanisms underlying cardiotoxicity induced by new classes of drugs remain only partially understood, and the available information is still limited. For this reason, we felt that including a figure on this topic might risk oversimplifying or providing a misleading representation. To enhance clarity, however, we have added a figure illustrating in detail the molecular mechanisms of anthracycline-induced cardiotoxicity, which are better established and supported by the current literature.
4-Some language mistakes have been detected and need to be improved.
Response 4: All detected language errors have been corrected to improve readability, grammar, and overall clarity of the manuscript.
Round 2
Reviewer 4 Report
Comments and Suggestions for Authors
The authors can add a section about the role of proteasome inhibitors in cardio-oncology (i.e. Carfilzomib and Bortezomib), including both preclinical and clinical data. I think this will enrich the manuscript.
Author Response
Comment: The authors can add a section about the role of proteasome inhibitors in cardio-oncology (i.e. Carfilzomib and Bortezomib), including both preclinical and clinical data. I think this will enrich the manuscript.
Response: We thank the reviewer for this valuable suggestion. We agree that including a discussion on the role of proteasome inhibitors in cardio-oncology would enrich the manuscript. Accordingly, we have added a new section (paragraph 2.5) dedicated to this topic, highlighting both preclinical and clinical evidence regarding the cardiotoxicity and cardiovascular effects of proteasome inhibitors such as Carfilzomib and Bortezomib. In addition, we have expanded paragraph 4.2. Cardiac Surveillance Protocols with a specific discussion on surveillance strategies related to proteasome inhibitors.